# Regret Bounds for Episodic Risk-Sensitive Linear Quadratic Regulator

**Wenhao Xu, Xuefeng Gao,**[*] **& Xuedong He**
Department of Systems Engineering and Engineering Management
The Chinese University of Hong Kong, Hong Kong, China
{whxu, xfgao, xdhe}@se.cuhk.edu.hk

## Abstract

Risk-sensitive linear quadratic regulator is one of the most fundamental problems in risk-sensitive optimal control. In this paper, we study online adaptive control of risk-sensitive linear quadratic regulator in the finite horizon episodic setting. We propose a simple least-squares greedy algorithm and show that it achieves $\widetilde{\mathcal{O}}(\log N)$ regret under a specific identifiability assumption, where $N$ is the total number of episodes. If the identifiability assumption is not satisfied, we propose incorporating exploration noise into the least-squares-based algorithm, resulting in an algorithm with $\widetilde{\mathcal{O}}(\sqrt{N})$ regret. To our best knowledge, this is the first set of regret bounds for episodic risk-sensitive linear quadratic regulator. Our proof relies on perturbation analysis of less-standard Riccati equations for risk-sensitive linear quadratic control, and a delicate analysis of the loss in the risk-sensitive performance criterion due to applying the suboptimal controller in the online learning process.

## 1 Introduction

In classical reinforcement learning (RL), one optimizes the *expected cumulative rewards* in an unknown environment modeled by a Markov decision process (MDP, Sutton & Barto (2018)). However, this risk-neutral performance criterion may not be the most suitable one in applications such as finance, robotics and healthcare. Hence, a large body of literature has studied *risk-sensitive* RL, incorporating the notion of risk into the decision criteria, see, e.g., Mihatsch & Neuneier (2002); Shen et al. (2014); Chow et al. (2017); Prashanth L & Fu (2018).

In this paper, we study online learning and adaptive control for a *risk-sensitive* linear quadratic control problem, referred to as the Linear Exponential-of-Quadratic Regulator (LEQR) problem. The LEQR problem is one of the most fundamental problems in risk-sensitive optimal control, and there is extensive literature on this topic (Jacobson, 1973; Whittle, 1990; Zhang et al., 2021a). In this control problem, the system dynamics is linear in the state and control variables, and it is disturbed with additive Gaussian noise. The cost in each period is convex quadratic in both the state and the control/action variables, and the performance criteria is the logarithm of the expectation of the exponential functions of the cumulative costs. When the system parameters are known, the optimal control at each stage is linear in state with the coefficient determined by certain Riccati equation. Different from the risk-neutral setting, the solution to the Riccati equation for LEQR explicitly depends on the risk parameter and the covariance matrix of the additve Gaussian noise in the system dynamics (Jacobson, 1973). For general risk-sensitive nonlinear control, one does not have such closed-form solutions. However, one can use LEQR as a local approximation model and solve risk-sensitive control problems by iteratively solving LEQR problems, see e.g. Roulet et al. (2020).

We consider the standard *finite-horizon episodic* RL setting, where the system matrices of LEQR are *unknown* to the agent. The learning agent repeatedly interacts with the unknown system over $N$ episodes, the time horizon of each episode is fixed, and the system resets to a fixed initial state distribution at the beginning of each episode. We focus on the finite horizon LEQR model because it is widely used as a model of locally linear dynamics. The performance of the agent or the online

---

[*]Corresponding author.

algorithm is often quantified by the total regret, which measures the cumulative suboptimality of the algorithm accrued over time as compared to the optimal policy. We seek algorithms with (finite-time) regret that is sublinear in $N$, which means the per episode regret converges to zero and the agent can act near optimally as $N$ grows.

Regret bounds for the *risk-neutral* linear quadratic regulator (LQR) in the *infinite-horizon average reward setting* have been extensively studied in the literature, see e.g. Abbasi-Yadkori & Szepesvári (2011); Mania et al. (2019); Cohen et al. (2019); Simchowitz & Foster (2020). It has been shown that in this *average reward setting*, the *certainty-equivalent controller* where the agent selects control inputs according to the optimal controller for her estimate of the system, together with a simple random-search type exploration strategy, is (rate-)optimal for the online adaptive control of *risk-neutral* LQR (Simchowitz & Foster, 2020). However, non-asymptotic regret analysis of the finite-horizon episodic LQR has received much less attention, though some applications, especially in finance, naturally fall into the episodic setting. For example, a common task faced by a financial institution is to liquidate a large position of assets, e.g., a stock, in a finite amount of time, e.g., in one day. With a linear price impact, such problems can be formulated as a stochastic control problem with linear dynamics and quadratic cost functions; see Section 1.5 of Almgren & Chriss (2001). One can consider optimizing the expected utility of the total cost of trading and with an exponential utility function (see e.g. Schied et al. (2010)), the problem becomes an episodic LEQR problem. In different days, the institution may need to liquidate different assets, so the initial state of this problem, which represents the initial position of the asset that the institution needs to liquidate during the day, resets at the beginning of each day, resembling the episodic setting. Basei et al. (2022) is among the first to establish regret bounds for the *risk-neutral continuous time finite-horizon* LQR in the episodic setting. They proposed a greedy least-squares algorithm and established a regret bound that is logarithmic in the number of episodes $N$ under a specific identifiability condition. By contrast, we study finite-horizon LEQR, which is a risk-sensitive model, in a discrete-time setting.

On the other hand, there is a surge of interest recently on studying finite-time regret bounds for risk-sensitive RL. The first regret bound for risk-sensitive tabular MDP is due to Fei et al. (2020), who study episodic RL with the goal of optimizing the exponential utility of the cumulative rewards. There is now a rapidly growing body of literature on this topic, see, e.g. (Fei et al., 2020; 2021; Du et al., 2022; Bastani et al., 2022; Liang & Luo, 2022; Xu et al., 2023; Wang et al., 2023; Wu & Xu, 2023; Chen et al., 2024). Most of the studies consider learning in risk-sensitive MDPs with finite state and action spaces.

Inspired by these studies, in this paper we study regret bounds for online adaptive control of the (discrete-time) risk-sensitive LEQR in the finite-horizon episodic setting, where both the state and the action spaces are *continuous*. In particular, we obtain two main results:

- First, we propose a simple least-squares greedy algorithm without exploration noise (Algorithm 1), and show that it achieves a regret of order $\log N$ under a certain identifiability condition (Assumption 1) on the LEQR model.
- Second, without Assumption 1, we propose another algorithm with actively injected exploration noise (Algorithm 2), and show that it achieves a regret of order $\sqrt{N}$.

To the best of our knowledge, this is the first set of regret bounds for finite-horizon episodic LEQR. In the learning theory community, there has been a significant interest in the questions of whether logarithmic regret is possible for what type of linear systems and under what assumptions. See e.g. Agarwal et al. (2019); Cassel et al. (2020); Faradonbeh et al. (2020); Foster & Simchowitz (2020); Lale et al. (2020). Our first result provides an answer to these questions in the setting of risk-sensitive LEQR models. In addition, our second result complements the first result by showing that $\sqrt{N}$-regret bounds can be established for risk-sensitive LEQR models without the identifiability assumption.

We briefly discuss the technical challenges and highlight the novelty of our regret analysis. Even though our proposed algorithms are fairly simple, the analysis is nontrivial and it builds on two new components: (a) perturbation analysis of Riccati equation for LEQR; and (b) analysis of risk-sensitive performance loss due to the suboptimal controller applied in the online control process. For the perturbation analysis in (a), we cannot use the existing techniques from the literature on online learning in risk-neutral LQR (Mania et al., 2019; Simchowitz & Foster, 2020; Basei et al., 2022). This is because the Riccati equation for LEQR is less standard and more complicated: there are some

extra parameters (see $\widetilde{P}_t, t = 0, \cdots, T - 1$ in (5)) involved in the equation, and the risk-sensitive parameter impacts the solution to the Riccati equation. To overcome this challenge, we first analyze one-step perturbation bound for the solution to Riccati equation, and then leverage the recursive structure of Riccati equation from our finite-horizon LEQR problem to establish a bound on the controller mismatch in terms of the error in the estimated system matrices. For the performance loss in (b), we can not employ the existing approach in online control of risk-neutral LQR as well. This is because the performance objective in LEQR is nonlinear in terms of the random cumulative costs (unlike the expectation which is a linear operator). Indeed, this type of non-linearity has been one of the key challenges in regret analysis for risk-sensitive tabular MDPs (Fei et al., 2021). To address this challenge, we leverage results from Jacobson (1973) for LEQR to express the performance loss in terms of the controller mismatch (i.e. the gap between the executed controller and the optimal controller). Due to these two new technical components, our analysis is substantially different from the proof in the closely related work Basei et al. (2022). In addition, Basei et al. (2022) did not analyze the case when the identifiability condition does not hold and provide $\sqrt{N}$-regret bound.

There are several recent studies on RL for LEQR. Zhang et al. (2021a) proposes model-free policy gradient methods for solving the finite-horizon LEQR problem and provides a sample complexity result. Sample complexity is another popular performance metric for RL algorithms in addition to the regret. Note that the controller in Zhang et al. (2021a) is assumed to have simulation access to the model, i.e., the controller can execute multiple policies within each episode. By contrast, our work considers online control of LEQG with regret guarantees, where we do not assume access to a simulator and the agent can only execute one policy within each episode. Other related works include Zhang et al. (2021b), which proposes a nested natural actor-critic algorithm for LEQR with the average reward criteria, and Cui et al. (2023), which proposes a robust policy optimization algorithm for solving the LEQR problem to handle model disturbances and mismatches. These studies do not consider regret bounds for LEQR, and hence are different from our work.

Finally, we comment that an alternative approach to considering risk sensitive LQR is $H_\infty$-optimal adaptive control (Hassibi et al., 1999). This approach takes a different perspective from LEQR: it considers deterministic, unknown noise and, instead of taking expectation with respect to random noise as in LEQR, it considers the $H_\infty$ norm of the cost with respect to the deterministic, unknown noise. Thus, in the presence of system noise, $H_\infty$-optimal adaptive control takes the robust control approach to consider the worst case performance while LEQR assumes a probabilistic model for the noise and a degree of risk aversion. Regret bounds for $H_\infty$ control have been studied in e.g. Karapetyan et al. (2022). Because of different settings and objective functions in LEQG and $H_\infty$-optimal control, the regret bounds in these two problems are not directly comparable.

## 2 PROBLEM FORMULATION

### 2.1 THE LEQR PROBLEM

We first provide a brief review of the LEQR problem (Jacobson, 1973). We consider the following linear discrete-time dynamic system:

$$x_{t+1} = Ax_t + Bu_t + w_t, \quad t = 0, 1, \cdots, T - 1, \tag{1}$$

where the state vector $x_t \in \mathbb{R}^n$, the control vector $u_t \in \mathbb{R}^m$, the matrices $A \in \mathbb{R}^{n \times n}$, $B \in \mathbb{R}^{n \times m}$, and the process noise $w_t \in \mathbb{R}^n$ form a sequence of i.i.d. Gaussian random vectors. For the simplicity of presentation, we assume the noise $w_t \sim \mathcal{N}(0, I)$ where $I$ is the identity matrix. The goal in the finite-horizon LEQR problem is to choose a control policy $\pi = \{u_0, u_1, \cdots, u_{T-1}\}$ so as to minimize the exponential risk-sensitive cost given by

$$J^\pi(x_0) = \frac{1}{\gamma} \log \mathbb{E} \exp\left(\frac{\gamma}{2}\left(\sum_{t=0}^{T-1} c_t(x_t, u_t) + c_T(x_T)\right)\right), \tag{2}$$

where $c_t(x_t, u_t) = x_t^\top Q x_t + u_t^\top R u_t$, $c_T(x_T) = x_T^\top Q_T x_T$, $Q \succeq 0, Q_T \succeq 0$ (i.e. positive semidefinite), $R \succ 0$ (i.e. positive definite), and $\gamma$ is the risk-sensitivity parameter.

Note that when $\gamma$ is small, we have by Taylor expansion:

$$\frac{1}{\gamma} \log \mathbb{E} \exp(\gamma Z) = \mathbb{E}[Z] + \frac{\gamma}{2} Var(Z) + O(\gamma^2),$$

for a random variable $Z$ with a finite moment generating function. It is well understood in the economics literature that $\gamma$ measures the risk aversion degree, and a positive (negatively, respectively) $\gamma$ stands for risk-averse (risk-seeking, respectively) attitude; see for instance Pratt (1964). When $\gamma \to 0$, the LEQR problem reduces to the conventional risk-neutral linear quadratic control where one minimizes the expected total quadratic cost and the controller becomes risk-neutral. For concreteness, we focus on the case where $\gamma > 0$ (our analysis extends to $\gamma \leq 0$). The optimal performance is denoted by

$$J^\star(x_0) = \inf_\pi J^\pi(x_0). \tag{3}$$

When the system parameters are all *known*, Jacobson (1973) shows that under the assumption that $I - \gamma P_{t+1} \succ 0$ for all $t = 0, 1, \cdots, T - 1$ (Note that if $\gamma$ is too large, we have $J^\pi(x_0) = \infty$ for all policies), the optimal feedback control for (3) is a linear function of the system state

$$u_t^\star = K_t x_t, \quad t = 0, 1, \cdots, T - 1, \tag{4}$$

where $(K_t)$ can be solved from the following discrete-time (modified) Riccati equation:

$$
\begin{aligned}
P_T &= Q_T, \\
\widetilde{P}_{t+1} &= P_{t+1} + \gamma P_{t+1} \left(I_n - \gamma P_{t+1}\right)^{-1} P_{t+1}, \\
K_t &= -(B^\top \widetilde{P}_{t+1} B + R)^{-1} B^\top \widetilde{P}_{t+1} A, \\
P_t &= Q + K_t^\top R K_t + (A + BK_t)^\top \widetilde{P}_{t+1}(A + BK_t), \\
& t = 0, 1, \cdots, T - 1.
\end{aligned} \tag{5}
$$

One can see that scaling all the cost matrices $Q, Q_T$, and $R$ does not change the optimal controller, and hence we assume $R \succeq I_m$ without loss of generality. Note that in the risk-neutral setting where $\gamma = 0$, we have $\widetilde{P}_t = P_t$ in the Riccati equation (5). However, in the risk-sensitive setting, we have extra matrices $(\widetilde{P}_t)$ in the Riccati equation. This is one of the difficulties we need to overcome when we study perturbation analysis of Riccati equations for the LEQR problem.

## 2.2 Finite-horizon Episodic RL in LEQR

In this paper, we consider the online learning/control setting for LEQR, where the system matrices $(A, B)$ are *unknown* to the agent. The learning agent repeatedly interacts with the linear system (1) over $N$ episodes, where the time horizon of each episode is $T$. In each episode $i = 1, 2, \cdots, N$, an arbitrary fixed initial state $x_0^k = x_0 \in \mathbb{R}^n$ is picked.[1] An online learning algorithm executes policy $\pi^i$ throughout episode $i$ based on the observed past data (states, actions and costs) up to the end of episode $i-1$. The performance of an online algorithm over $N$ episodes of interaction with the linear system (1) is the (total) regret:

$$\text{Regret}(N) = \sum_{i=1}^N \left(J^{\pi^i}(x_0^i) - J^\star(x_0^i)\right),$$

where the term $J^{\pi^i}(x_0^i) - J^\star(x_0^i)$ (see (2) and (3)) measures the performance loss when the agent executes the suboptimal policy $\pi^i$ in episode $i$.

# 3 A LOGARITHMIC REGRET BOUND

In this section, we propose a simple least-squares greedy algorithm and show that it achieves a regret that is logarithmic in $N$, under a specific identifiability assumption.

## 3.1 A Least-Squares Greedy Algorithm

We now present the details of the least-squares greedy algorithm, which combines least-squares estimation for the unknown system matrices $(A, B)$ with a greedy strategy.

---

[1]The results of the paper can also be extended to the case where the initial states are drawn from a fixed distribution over $\mathbb{R}^n$.

We divide the $N$ episodes into $L$ epochs. The $l$-th epoch has $m_l$ episodes, thus $\sum_{l=1}^{L} m_l = N$. At the beginning of the $l$-th epoch, we estimate the system matrices $(A, B)$ by using the data from the $(l-1)$-th epoch, and the obtained estimator is denoted by $(A^l, B^l)$. Then we select the control inputs according to the optimal controller for the estimate $(A^l, B^l)$ of the system, and execute such a policy throughout epoch $l$. The feedback control $K_t^l$ is obtained by replacing $(A, B)$ in (5) with the estimate $(A^l, B^l)$. Then, in the $k$-th episode of epoch $l$, we play the greedy policy $u_t^{l,k}$ by taking $K_t^l$ into (4).

It remains to discuss the estimation procedure for $(A^l, B^l)$ which is conducted at the beginning of epoch $l$. Within the $l$-th epoch, we note that the same policy is executed in each episode. Because we consider the episodic setting where the system state reset to the same state at $t = 0$, we obtain that the state-action trajectories across different episodes are i.i.d within the same epoch. Note that the random linear dynamical system in epoch $l$ is given by

$$x_{t+1}^l = Ax_t^l + Bu_t^l + w_t^l, \quad t = 0, 1, \cdots, T-1, \tag{6}$$

where $u_t^l = K_t^l x_t^l$. For simplicity of notation, we denote by $z_t^l = \begin{bmatrix} x_t^{l\top} & u_t^{l\top} \end{bmatrix}^\top$, which is the state-action random vector at step $t$ in epoch $l$. We also denote by $\theta = [A\ B]^\top$ for the system matrices. Taking the transpose of (6) and multiplying $z_t^l$ on both sides of (6), we can get $z_t^l x_{t+1}^{l\top} = z_t^l z_t^{l\top} \theta + z_t^l w_t^{l\top}$. Summing over $T$ steps and taking the expectation, we obtain $\mathbb{E}\left[Y^l\right] = \mathbb{E}\left[V^l\right] \theta$, where $V^l = \sum_{t=0}^{T-1} z_t^l z_t^{l\top}$ and $Y^l = \sum_{t=0}^{T-1} z_t^l x_{t+1}^{l\top}$. It follows that

$$\theta = [A\ B]^\top = \left(\mathbb{E}\left[V^l\right]\right)^{-1} \left(\mathbb{E}\left[Y^l\right]\right), \tag{7}$$

provided that the matrix $\mathbb{E}[V^l]$ is invertible. The formula (7) and the fact that state-action trajectories across different episodes are i.i.d. within the same epoch provide the basis for our estimation procedure. Given the data in epoch $l$, we now discuss the construction of the estimator $\theta^{l+1} := \begin{bmatrix} A^{l+1}, B^{l+1} \end{bmatrix}^\top$.

Consider the sample state process in the $k$-th episode of epoch $l$:

$$x_{t+1}^{l,k} = Ax_t^{l,k} + Bu_t^{l,k} + w_t^{l,k}, t = 0, 1, \cdots, T-1. \tag{8}$$

Denote the sample state-action vector by $z_t^{l,k} = \begin{bmatrix} x_t^{l,k\top} & u_t^{l,k\top} \end{bmatrix}^\top$. Then, we can design the $l_2$-regularized least-squares estimation for $\theta$ by replacing the expectation in (7) with the sample average over the $m_l$ episodes in epoch $l$ and adding the regularized term $\frac{1}{m_l} I_{n+m}$:

$$\theta^{l+1} = \left(\bar{V}^l + \frac{1}{m_l} I_{n+m}\right)^{-1} \bar{Y}^l, \tag{9}$$

where $\bar{V}^l = \frac{1}{m_l} \sum_{k=1}^{m_l} \sum_{t=0}^{T-1} z_t^{l,k} z_t^{l,k\top}$ and $\bar{Y}^l = \frac{1}{m_l} \sum_{k=1}^{m_l} \sum_{t=0}^{T-1} z_t^{l,k} x_{t+1}^{l,k\top}$.

We now summarize the details of the least-squares greedy algorithm in Algorithm 1. Note that the input parameter $\theta^1$ denotes the initial guess of the true system matrices $(A, B)$.

---

**Algorithm 1** The Least-Squares Greedy Algorithm

---

  **Input:** Parameters $L, T, m_1, \theta^1, Q, Q_T, R$
  **for** $l = 1, \cdots, L$ **do**
    $m_l = 2^{l-1} m_1$
    Compute $(K_t^l)$ for all $t$ by (5) using $\theta^l$.
    **for** $k = 1, \cdots, m_l$ **do**
      **for** $t = 0, \cdots, T-1$ **do**
        Play $u_t^{l,k} \leftarrow K_t^l x_t^{l,k}$.
      **end for**
    **end for**
    Obtain $\theta^{l+1}$ from the $l_2$-regularized least-squares estimation (9).
  **end for**

---

## 3.2 Logarithmic Regret

In this section, we state our first main result. We first introduce the following assumption.

**Assumption 1.** *For the sequence of the controller* $(K_t)$ *defined in (5), we assume that*

$$\left\{ v \in \mathbb{R}^{n+m} \,\middle|\, \left[ I_n \ K_t^\top \right] v = 0, \forall t = 0, \cdots, T-1 \right\} = \{0\}. \tag{10}$$

Assumption 1 is essentially Assumption H.1(2) in Basei et al. (2022) for learning finite-horizon continuous-time risk-netural LQR, and it is referred to as the self-exploration property therein (i.e., exploration is 'automatic' due to the system noise and the *time-dependent* optimal feedback matrix $(K_t)_{t=0,\ldots,T-1}$ ). One can show that Assumption 1 is equivalent to the condition (see Lemma 7)

$$\mathbb{E} \left[ \sum_{t=0}^{T-1} z_t z_t^\top \right] = \sum_{t=0}^{T-1} \left[ \begin{array}{c} I_n \\ K_t \end{array} \right] \mathbb{E} \left[ x_t x_t^\top \right] \left[ I_n \ K_t^\top \right] \succ 0, \tag{11}$$

which resembles the persistence of excitation assumption in adaptive control (Aström & Wittenmark, 2008, Definition 2.1, Chapter 2).

In view of (7) and (11), Assumption 1 essentially guarantees the identifiability of the true system matrices when the time-dependent optimal control in (5) is executed. This is important for the proposed greedy least-squares algorithm to achieve a logarithmic regret bound. Assumption 1 can be satisfied under various sufficient conditions. We provide one set of sufficient conditions in Proposition 4 in the appendix.

We now present our first main result, which provides a logarithmic regret bound of Algorithm 1. We denote $\| \cdot \|$ as the spectral norm for matrices.

**Theorem 1.** *Suppose Assumption 1 holds and assume the optimal controller for the initial estimate* $\theta^1$ *also satisfy (10). Fix* $\delta \in (0, \frac{3}{\pi^2})$. *Then we can choose* $m_1 = \mathcal{C}_0(-\log \delta)$ *for some positive constant* $\mathcal{C}_0$ *such that with probability at least* $1 - \frac{\pi^2 \delta}{3}$, *the regret of Algorithm 1 satisfies*

$$\mathrm{Regret}(N) \leq \mathcal{C} \left( \sum_{t=0}^{T-1} \psi_t \right) \left[ \log \left( \frac{m+n}{\sqrt{\delta}} \right) L + L \log L \right], \tag{12}$$

*where* $\mathcal{C}$ *is a constant independent of* $N$ *and* $(\psi_t)$ *is a sequence recursively defined by*

$$\psi_{T-1} = 2\widetilde{\Gamma}^3, \quad \psi_t = 2\widetilde{\Gamma}^3 (10 \mathcal{V}^2 \mathcal{L} \widetilde{\Gamma}^4)^{2(T-t-1)} + 12 \widetilde{\Gamma}^4 \psi_{t+1}, \ t \in [T-2],$$

*with*

$$\Gamma_t = \max \left\{ \|A\|, \|B\|, \|Q\|, \|Q_T\|, \|R\|, \|P_t\|, \|\widetilde{P}_t\|, \|K_{t-1}\| \right\}, \quad \widetilde{\Gamma} = 1 + \max_t \Gamma_t,$$

$$\mathcal{V} = 2(\mathcal{L}+1)\widetilde{\Gamma}^3, \quad \mathcal{L} = \frac{1}{(1 - \gamma \sigma^2 \widetilde{\Gamma})^2}. \tag{13}$$

Because $\sum_{l=1}^{L} m_l = N$ and $m_l = 2^{l-1} m_1$, we infer that $L = \left\lceil \log_2 \left( \frac{N}{m_1} + 1 \right) \right\rceil \lesssim \log N$, where $\lesssim$ means the inequality holds with a multiplicative constant. Hence, Theorem 1 implies that the regret of Algorithm 1 satisfies $\mathrm{Regret}(N) = O\left( \log N \cdot \log \log(N) \right)$, where $O$ hides dependency on other constants. In Appendix C, we provide some further discussions on the dependency of the regret bound on other problem parameters, including the horizon length $T$, and the risk parameter $\gamma$ of the LEQR model. Note that Algorithm 1 requires $L$, or equivalently $N$ (the total number of episodes) as input. For unknown $N$, one can use the doubling trick (Besson & Kaufmann, 2018). Specifically, consider an increasing sequence $\{n_k\}_{k=0}^{\infty}$ where $n_k = 2^{2^k}$ for $k \geq 1$ and $n_0 = 0$. For each $k$, one restarts Algorithm 1 at the beginning of episode $n_k + 1$, and run the algorithm until episode $n_{k+1}$ with the input $N = n_{k+1} - n_k$. One can readily verify that this leads to an anytime algorithm which still achieves a logarithmic regret bound.

## 3.3 Proof Sketch of Theorem 1

In this section, we provide the proof sketch of Theorem 1. The full proof is given in Appendix A.

**Step 1:** We adapt the analysis in Basei et al. (2022) and use Bernstein inequality for the sub-exponential random variables to derive the following bound on estimation errors of system matrices.

**Proposition 1** (Informal). *Fix $\delta \in (0, \frac{3}{\pi^2})$. Let $\delta_l = \delta/l^2$. For $m_l \gtrsim \log\left(\frac{(m+n)^2}{\delta_l}\right)$, we have with probability at least $1 - 2\delta_l$,*

$$\left\|\theta^{l+1} - \theta\right\| \lesssim \sqrt{\frac{\log\left(\frac{(m+n)^2}{\delta_l}\right)}{m_l}} + \frac{\log\left(\frac{(m+n)^2}{\delta_l}\right)}{m_l} + \frac{\log^2\left(\frac{(m+n)^2}{\delta_l}\right)}{m_l^2}.$$

For a complete rigorous statement, see Proposition 3 in Appendix.

**Step 2:** We recursively carry out the perturbation analysis of less-standard Riccati equation (5) and prove that the perturbation of the controller $\Delta K_t^l := K_t^l - K_t$ is on the order of $O(\epsilon_l)$, where $\epsilon_l = \max\left\{\|A^l - A\|, \|B^l - B\|\right\}$. The formal statement is presented in Lemma 8.

**Step 3:** We use a result of Jacobson (1973) (see Lemma 10) and the proof technique in Fazel et al. (2018) to prove that

$$J^{\pi^{l,k}}(x_0^{l,k}) - J^\star(x_0^{l,k}) = -\frac{1}{2\gamma}\sum_{t=1}^{T-1}\log\left(\det\left(I_n - \gamma D_t^l\right)\right) + \frac{1}{2}x_0^{l,k\top}D_0^l x_0^{l,k},$$

where $D_t^l$ is a function of $\Delta K_t^l, \cdots, \Delta K_{T-1}^l$, with $\|D_t^l\| \leq \psi_t \mathcal{V}^2 \epsilon_l^2 + o(\epsilon_l^2)$. See Proposition 5. Here, $\pi^{l,k}$ is the sub-optimal controller $\pi^{l,k}$ executed in the $k$-th episode of the $l$-th epoch.

**Step 4:** We can then bound the regret: $\text{Regret}(N) = \sum_{l=1}^{L}\sum_{k=1}^{m_l}\left(J^{\pi^{l,k}}(x_0^{l,k}) - J^\star(x_0^{l,k})\right) \lesssim \sum_{l=1}^{L}m_l\epsilon_l^2 \lesssim \sum_{l=1}^{L}\log(l) \lesssim O(\log N \cdot \log\log(N))$.

# 4 A SQUARE-ROOT REGRET BOUND

Theorem 1 shows that the logarithmic regret bound is achievable for episodic LEQR under Assumption 1. One may wonder how does the regret bound changes after removing Assumption 1. In particular, is $\sqrt{N}$ regret achievable without Assumption 1? This section provides an affirmative answer to this question, by proposing and analyzing a least-squares-based algorithm with actively injected exploration noise.

## 4.1 A LEAST-SQUARES-BASED ALGORITHM WITH EXPLORATION NOISE

We now introduce the algorithm (Algorithm 2), which is is different from Algorithm 1. We no longer divide the $N$ episodes into epochs of increasing lengths to estimate the system matrices. Instead, in the $k$-th episode, the algorithm updates the estimation of the system matrices $(A, B)$ by using the data from the previous $k - 1$ episodes, which is denoted by $(A^k, B^k)$. Similar to $K_t^l$ in Section 3.1, we can obtain the feedback control $K_t^k$ by replacing the true system matrices in (5) with $(A^k, B^k)$. Then, we execute the control with exploration noise $(g_t^k)$ that follows a Gaussian distribution in the $k$-th episode. The design of Algorithm 2 is inspired by (Mania et al., 2019; Simchowitz & Foster, 2020) that establish $\sqrt{T}$ regret bounds for risk-neutral LQR in the infinite-horizon average reward setting, where $T$ is the number of time steps.

The estimation of system matrices $(A, B)$ in Algorithm 2 is different from that in Algorithm 1. In Algorithm 2, the estimator $(A^{k+1}, B^{k+1})$ is obtained by solving the following $l_2$-regularized least-squares problem (based on the linear dynamics (1)):

$$\theta^{k+1} \in \arg\min_y\left\{\lambda\|y\|^2 + \sum_{i=1}^{k}\sum_{t=0}^{T-1}\|x_{t+1}^i - y^\top z_t^i\|^2\right\}, \tag{14}$$

where $\theta^{k+1} = \left[A^{k+1}, B^{k+1}\right]^\top$, $z_t^i = [x_t^{i\top}, u_t^{i\top}]^\top$ and $\lambda > 0$ is the regularization parameter. By solving (14), we can get

$$\theta^{k+1} = \left(\bar{V}^k\right)^{-1}\left(\sum_{i=1}^{k}\sum_{t=0}^{T-1}z_t^i x_{t+1}^{i\top}\right), \tag{15}$$

where $\bar{V}^k = \lambda I + \sum_{i=1}^{k}\sum_{t=0}^{T-1}z_t^i z_t^{i\top}$.

---

**Algorithm 2** The Least-Squares-Based Algorithm with Exploration Noise

---

**Input:** Parameters $T, N, \theta^1, Q, Q_T, R, \lambda$
**for** $k = 1, \cdots, N$ **do**
    Compute $(K_t^k)$ for all $t$ by (5) using $\theta^k$.
    **for** $t = 0, \cdots, T-1$ **do**
        Play $u_t^k \leftarrow K_t^k x_t^k + g_t^k, g_t^k \sim \mathcal{N}(0, \frac{1}{\sqrt{k}} I_m)$.
    **end for**
    Obtain $\theta^{k+1}$ from (15).
**end for**

---

### 4.2 SQUARE-ROOT REGRET

In this section, we present the second main result of our paper, which demonstrates that Algorithm 2 can attain $\widetilde{\mathcal{O}}(\sqrt{N})$-regret (ignoring logarithmic factors). This is the first $\widetilde{\mathcal{O}}(\sqrt{N})$-regret for online learning in risk-sensitive LEQR models.

**Theorem 2.** *Fix $\delta \in (0,1)$. When $N \geq 200 \left(3(n+m) + \log\left(\frac{4N}{\delta}\right)\right)$, with probability at least $1-\delta$, the regret of Algorithm 2 satisfies*

$$\text{Regret}(N) \leq \widetilde{\mathcal{C}} \sum_{t=0}^{T-1} (\alpha_t C_N + \beta_t) \sqrt{N},$$

*where $\widetilde{\mathcal{C}}$ is a constant independent of $N$, $C_N$ exhibits a logarithmic dependence on $N$ and depends on $\lambda$, and $(\alpha_t), (\beta_t)$ are two sequences recursively defined by*

$$\alpha_{T-1} = 2\widetilde{\Gamma}^3, \quad \alpha_t = 2\widetilde{\Gamma} \left(10\mathcal{V}^2 \mathcal{L}\widetilde{\Gamma}^4\right)^{2(T-t-1)} + 12\widetilde{\Gamma}^4 \alpha_{t+1},$$

$$\beta_{T-1} = 0, \quad \beta_t = 12\widetilde{\Gamma}^4 + 12\widetilde{\Gamma}^4 \beta_{t+1},$$

*with $\widetilde{\Gamma}, \mathcal{V}, \mathcal{L}$ defined in (13).*

### 4.3 PROOF SKETCH OF THEOREM 2

We provide a proof outline for Theorem 2. The complete proof is given in Appendix B.

**Step 1:** We adapt the self-normalized martingale analysis framework (Abbasi-Yadkori et al., 2011; Cohen et al., 2019; Simchowitz & Foster, 2020) to derive the following high probability bound for the estimation error. See Proposition 6 for the complete statement.

**Proposition 2** (informal). *When $k$ is large enough, with probability at least $1 - \delta$,*

$$\left\|\theta^{k+1} - \theta\right\| \lesssim k^{-\frac{1}{4}} \sqrt{\log\left(1 + k \log\left(\frac{N}{\delta}\right)\right)}. \tag{16}$$

**Step 2:** We conduct perturbation analysis of the Riccati equation (5) and show that $\Delta K_t^k := K_t^k - K_t$ is on the order of $O(\epsilon_k)$, where $\epsilon_k$ denotes the estimation error of system matrices, i.e. right-hand-side of (16). This step is essentially the same as Step 2 in Section 3.3.

**Step 3:** Because of the additional exploration noise added to the online control, we show that the loss in the risk-sensitive performance becomes

$$J^{\pi^k}(x_0^k) - J^\star(x_0^k) = -\frac{1}{2\gamma} \sum_{t=0}^{T-1} \log\det\left(I_n - \gamma F_t^k\right) - \frac{1}{2\gamma} \sum_{t=1}^{T-1} \log\det\left(I_m - \gamma U_t^k\right) + \frac{1}{2} x_0^{k\top} U_0^k x_0^k,$$

where $F_t^k$ and $U_t^k$ are functions of $\Delta K_i^k = K_i^k - K_i, i = t, \cdots, T-1$, with $F_t^k \leq \frac{2\widetilde{\Gamma}^3}{\sqrt{k}} + o(\epsilon_k^2)$ and $U_t^k \leq \alpha_t \mathcal{V}^2 \epsilon_k^2 + \frac{5\widetilde{\Gamma}^5(1+\beta_t)}{\sqrt{k}} + o(\epsilon_k^2)$. See Proposition 7.

**Step 4:** Finally we can bound the regret: $\text{Regret}(N) = \sum_{k=1}^{N} \left(J^{\pi^k}(x_0^k) - J^\star(x_0^k)\right) \lesssim \sum_{k=1}^{N} \epsilon_k^2 \lesssim \sum_{k=1}^{N} \frac{1}{\sqrt{k}} \log\left(1 + N \log\left(\frac{N}{\delta}\right)\right) \lesssim \widetilde{\mathcal{O}}(\sqrt{N})$.

## 5    SIMULATION STUDIES

We perform simulation studies to illustrate the regret performances of Algorithms 1 and 2. Note that our paper is the first to obtain regret bounds for episodic risk-sensitive LEQR and there are currently no other existing algorithms with sublinear regret for this problem. Our experiments are conducted on a PC with 2.10 GHz Intel Processor and 16 GB of RAM. We consider the following three LEQR systems for illustrations:

**System 1.** We use the system matrices and cost matrices in Section 6.1 of Dean et al. (2020) with the following minor change: we set $Q_T = Q = \frac{1}{2}I$ instead of $Q = 10^{-3}I$ as in their paper because the effect of risk parameters is difficult to visualize when $Q$ has small eigenvalues.

**System 2.** We generate non-positive-semidefinite, non-symmetric random system matrices $A \in \mathbb{R}^{7 \times 7}$ and $B \in \mathbb{R}^{7 \times 4}$ with all entries sampled from the uniform distribution $U(0, 1)$. The cost matrices $Q$ and $R$ are randomly generated positive definite matrices, and we set $Q_T = 0$.

**System 3.** We consider a flying robot problem in Section IV of Tsiamis et al. (2020) with the following minor change: we set $Q_T = 0$ instead of $Q_T = \text{diag}\{1, 0.1, 2, 0.1\}$ as in their paper so that the effect of risk parameters is easier to visualize.

We implement Algorithms 1 and 2 in all the systems and compute the expectation and 95% confidence interval of the regret of each algorithm using 150 independent runs. In both algorithms, we randomly generate the initial guess $\theta^1 = (A^1, B^1)$ with all entries of $A^1$ and $B^1$ sampled from the uniform distribution. In Algorithm 1, we set $m_1 = 500$ and $L = \left\lceil \log_2 \left( \frac{N}{m_1} + 1 \right) \right\rceil$. In Algorithm 2, we set $\lambda = 0.8$.

Because the simulation results for Systems 1, 2 and 3 are similar, we only present those for System 1 here and make the results for System 2 and system 3 available in Appendix D. Figures 1a, 1b, and 1c show the average regret of Algorithm 1 in System 1 using 150 independent runs and Figures 1d, 1e and 1f show the average regret of Algorithm 2 in the same system. The two blue dotted lines in Figures 1a and 1d depict the 95% confidence interval of the regret when $\gamma = 0.1$ and $T = 3$. We observe that Algorithm 1 incurs a large regret in the initial learning process, and as a result, the actual performance of Algorithm 1 is worse than that of Algorithm 2 on this instance. This is because Algorithm 1 updates parameter estimations less frequently compared with Algorithm 2, and the inaccurate estimations in the initial learning process lead to a large regret for Algorithm 1. With $T = 3$, Figure 1b illustrates the effect of the risk aversion on the regret. The true value of the learning agent's risk aversion parameter $\gamma$ is 0.1. Figure 1b plots the regret of the algorithm when the true value of $\gamma$ is used and when a wrong value, e.g., 0.05, 0.2, and 0, is used. The plot shows that if one runs Algorithm 1 with a misspecified risk aversion degree $\gamma$, particularly with $\gamma = 0$, which corresponds to the algorithm in Basei et al. (2022) with the assumption of risk-neutral agents, the regret performance is much worse compared with the case of a correctly specified risk aversion parameter. Setting $\gamma = 0.01$, Figure 1c illustrates the effect of the time horizon $T$.[2] Consistent with our theoretical results, the regret is increasing in $T$. The same parameter settings are used in Figures 1d–1f, where we test the performance of Algorithm 2. The results are similar to those of Algorithm 1 in Figures 1a–1c.

## 6    CONCLUSION AND FUTURE WORK

This paper proposes two simple least-squares-based algorithm for online adaptive control of LEQR in the finite-horizon episodic setting. We prove that the least-squares greedy algorithm can achieve a regret bound that is logarithmic in the number of episodes under a identifiability condition of the system. We also prove that the least-squares-based algorithm with exploration noise can achieve $\widetilde{\mathcal{O}}(\sqrt{N})$-regret when the identifiability condition is not satisfied. To the best of our knowledge, this is the first set of regret bounds for LEQR.

The study of regret analysis for risk-sensitive control with continuous state and action spaces is still in its infancy, and there are many open questions. For instance, it would be interesting to

---

[2]We choose $\gamma = 0.01$ to ensure the existence of the solution to the Riccati equation (5) for the values of $T$ under consideration.

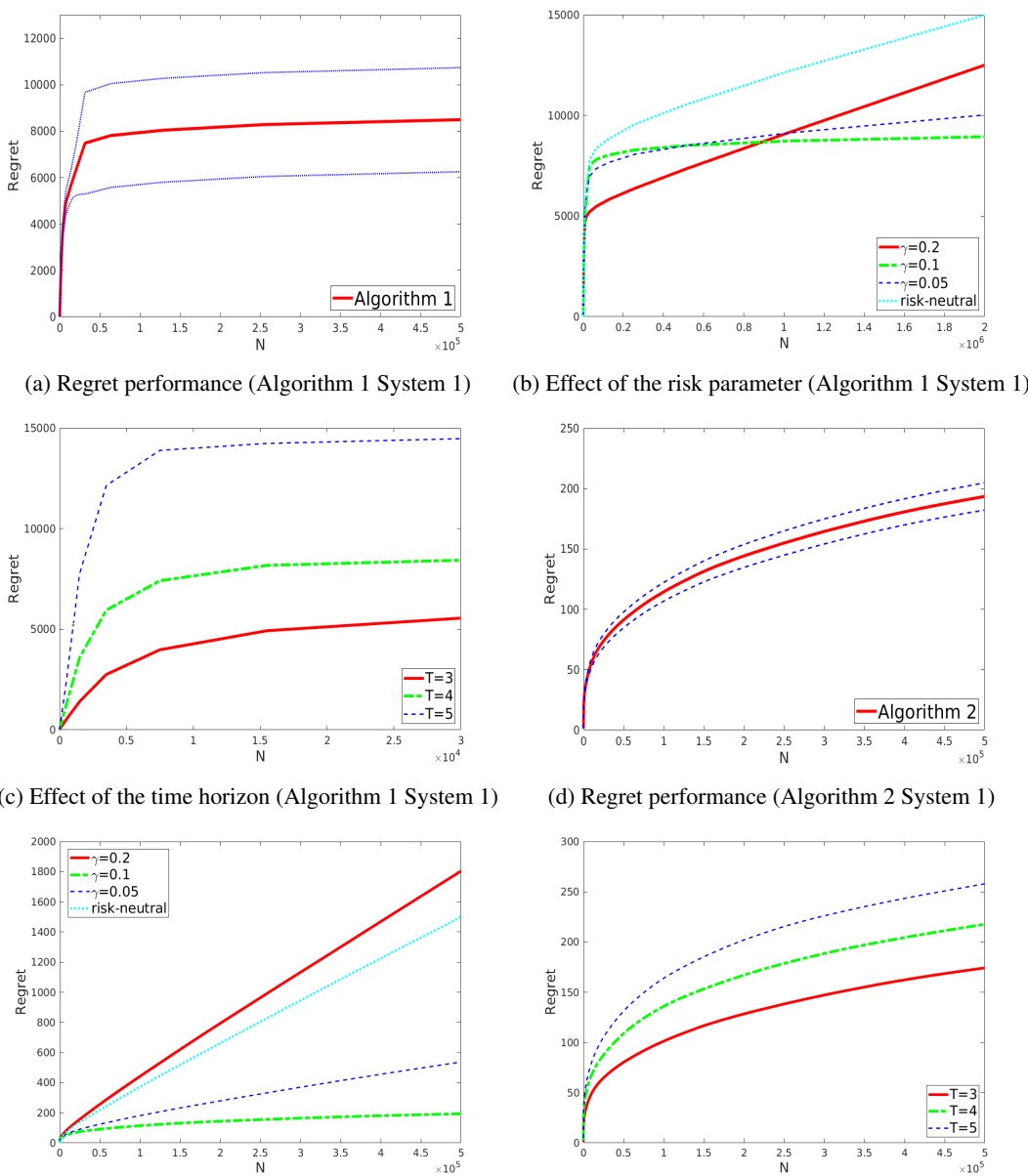

(a) Regret performance (Algorithm 1 System 1)  (b) Effect of the risk parameter (Algorithm 1 System 1)

(c) Effect of the time horizon (Algorithm 1 System 1)  (d) Regret performance (Algorithm 2 System 1)

(e) Effect of the risk parameter (Algorithm 2 System 1)  (f) Effect of the time horizon (Algorithm 2 System 1)

Figure 1: Simulation results in System 1

study regret bounds for LEQR in the infinite-horizon average-reward (non-episodic) setting. It is not straightforward to extend our current proof methods to this non-episodic setting because it is nontrivial to establish explicit perturbation bounds for the generalized algebraic Riccati equation for average-reward LEQR (see e.g. Cui et al. (2023)). Another significant question is to study regret bounds for online linear quadratic regulators with other risk measures such as coherent risk measures (see e.g. Lam et al. (2023)). Other interesting problems include lower bounds for online LEQR, regret bounds for LEQR with partially observable states and for more general risk-sensitive nonlinear control problems. We leave them for future work.

ACKNOWLEDGMENTS

We thank the area chair and anonymous referees for many constructive comments and suggestions. Xuefeng Gao and Xuedong He acknowledge support from the Hong Kong Research Grants Council [Grants 14201424, 14200123, 14212522, 14218022].

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

# A    REGRET ANALYSIS FOR THE LEAST-SQUARES GREEDY ALGORITHM

In this section, we carry out the regret analysis for the least-squares greedy algorithm in Section 3. We derive the high-probability bounds for the estimation error of system matrices in Appendix A.1. We do the perturbation analysis of Riccati equations in Appendix A.2. We simplify the suboptimality gap due to the controller mismatch in Appendix A.3. Finally, we combine the results derived/proved above and prove Theorem 1.

## A.1    BOUNDS FOR THE ESTIMATION ERROR OF SYSTEM MATRICES

In this section, we discuss the high probability bound for the estimation error of system matrices in Algorithm 1. We adapt the analysis framework in Basei et al. (2022) and use the Bernstein inequality for the sub-exponential random variables to derive the desired error bound.

To facilitate the presentation, we first introduce some notations. We fix the $l$-th epoch and define the following set

$$\Theta = \left\{ \hat{\theta} \in \mathbb{R}^{(n+m)\times n} \,\middle|\, \left\| \hat{\theta} - \theta \right\| \leq \rho \right\} \cup \left\{ \theta^1 \right\},$$

where $\rho > 0$ is a constant such that for any $\theta^l \in \Theta$, $\left\| \left( \mathbb{E}[V^l] \right)^{-1} \right\| \leq \mathcal{C}_2$ and $\left\| \mathbb{E}\left[ Y^l \right] \right\| \leq \mathcal{C}_2$ for some constant $\mathcal{C}_2 \geq 1$. We choose the initial number of episodes $m_1$ such that

$$\rho \geq 3\mathcal{C}_1 \sqrt{\frac{\log \left( \frac{(n+m)^2}{\delta_{j-1}} \right)}{m_{j-1}}}, \forall j \in \mathbb{N}^+\backslash\{1\},$$

where $\delta_{j-1} = \frac{\delta}{(j-1)^2}$, $m_{j-1} = 2^{j-2}m_1$ and $\mathcal{C}_1$ is a constant independent of $m_j, \forall j \in \mathbb{N}^+\backslash\{1\}$, but may depend on other constants including $m, n, T$. We will show how to choose $m_1$ in Section A.4. We also define the event

$$\mathcal{G}^l = \left\{ \theta^j \in \Theta, \forall j = 1, \cdots, l \right\}.$$

We will prove that $\mathbb{P}(\mathcal{G}^l) \geq 1 - \sum_{j=1}^{l-1} \delta_j$ in Section A.4. The following proposition is the main result of this section. Recall that $\theta^{l+1} = [A^{l+1}\ B^{l+1}]^\top$ are the estimated system matrices and $\theta = [A\ B]^\top$ are the true system matrices.

**Proposition 3.** *Conditional on event $\mathcal{G}^l$, there exists a constant $\mathcal{C}_3 \geq 1$ such that for $m_l \geq \mathcal{C}_3 \log \left( \frac{(n+m)^2}{\delta_l} \right)$, with probability at least $1 - 2\delta_l$,*

$$\left\| \theta^{l+1} - \theta \right\| \leq \mathcal{C}_1 \left( \sqrt{\frac{\log \left( \frac{(n+m)^2}{\delta_l} \right)}{m_l}} + \frac{\log \left( \frac{(n+m)^2}{\delta_l} \right)}{m_l} + \frac{\log^2 \left( \frac{(n+m)^2}{\delta_l} \right)}{m_l^2} \right).$$

The proof of Proposition 3 is long, and we discuss it in the new few sections.

### A.1.1    PRELIMINARIES

In this section, we recall the definition of sub-exponential random variables and state several well-known results about such random variables that will be used in our analysis later.

**Definition 1** (Definition 2.7 of Wainwright (2019)). *A random variable $X$ with mean $\mu = \mathbb{E}X$ is sub-exponential if there are non-negative parameters $(\nu, \alpha)$ such that $\mathbb{E}[e^{\lambda(X-\mu)}] \leq e^{\frac{\nu^2\lambda^2}{2}}$ for all $|\lambda| < \frac{1}{\alpha}$. Denote the set of such random variables as $SE(\nu^2, \alpha)$.*

**Lemma 1** (Bernstein Inequality, Proposition 2.9 of Wainwright (2019)). *Suppose that $X \in SE(\nu^2, \alpha)$, and let $\mu = \mathbb{E}X$. Then for any $\zeta > 0$, we have*

$$\mathbb{P}\left( |X - \mu| \geq \zeta \right) \leq 2\exp \left( -\min \left\{ \frac{\zeta^2}{2\nu^2}, \frac{\zeta}{2\alpha} \right\} \right).$$

**Lemma 2** (Lemma 5.1 of Alessandro (2018)). *If $X_i \in SE(\nu_i^2, \alpha_i), i \in [n]$, then*

$$\sum_{i=1}^n X_i \in \begin{cases} SE\left(\sum_{i=1}^n \nu_i^2, \max_{i \in [n]} \alpha_i\right) & \text{if } X_i \text{ are independent,} \\ SE\left(\left(\sum_{i=1}^n \nu_i\right)^2, \max_{i \in [n]} \alpha_i\right) & \text{if } X_i \text{ are not independent.} \end{cases}$$

**Lemma 3** (Lemma 2.7.7 of Vershynin (2018)). *Let $X$ and $Y$ be sub-Gaussian random variables. Then, $XY$ is sub-exponential.*

### A.1.2 PROPERTIES OF ESTIMATED SYSTEM MATRIX

In this section, we use the properties of sub-exponential random variables to derive some statistical properties for the estimated system matrix $\theta^l$.

The following lemma shows that every element of the state-action random sample vector $z_t^{l,k} = \left[x_t^{l,k\top} \ u_t^{l,k\top}\right]^\top$ is sub-Gaussian.

**Lemma 4.** *Consider the sample state (8) in section 3.1, conditional on event $\mathcal{G}^l$, we can prove that every element of the sample state $x_t^{l,k}$ and action vector $u_t^{l,k}$ is sub-Gaussian for any step $t$, episode $k$, epoch $l$.*

*Proof.* By the definition (8), we have

$$\begin{aligned} x_t^{l,k} &= Ax_{t-1}^{l,k} + Bu_{t-1}^{l,k} + w_{t-1}^{l,k} \\ &= (A + BK_{t-1}^l)x_{t-1}^{l,k} + w_{t-1}^{l,k} \\ &= (A + BK_{t-1}^l)(A + BK_{t-2}^l)x_{t-2}^{l,k} + (A + BK_{t-1}^l)w_{t-2}^{l,k} + w_{t-1}^{l,k}. \end{aligned}$$

Repeating this procedure, we can get

$$x_t^{l,k} = \left(\prod_{i=t-1}^0 (A + BK_i^l)\right) x_0^{l,k} + \sum_{j=0}^{t-1}\left(\prod_{i=t-1}^{j+1}(A+BK_i^l)\right)w_j^{l,k}, \tag{17}$$

where $\prod_{i=t-1}^t (A + BK_i^l) := I_n$. Recall the definition of $K_t^l$ in (5) by using $\theta^l$. It's continuous in $\theta^l \in \Theta$, so $K_t^l$ is uniformly bounded for any $\theta^l \in \Theta$ by the boundedness of $\Theta$, i.e. there exists some constant $M > 0$ such that $\sup_t \|K_t^l\| \leq M$. Because $x_0^{l,k} = x_0$ and $\{w_i^{l,k}\}_{i=0}^{t-1}$ are independent zero-mean normal random variables, we can then readily obtain from (17) that every element of $x_t^{l,k}$ is sub-Gaussian by the uniform boundedness of $K_t^l$. Similarly, we can prove that every element of $u_t^{l,k} = K_t^l x_t^{l,k}$ is sub-Gaussian, which completes the proof. $\square$

Recall the following matrices in (7) and (9).

$$\begin{aligned} V^l &= \sum_{t=0}^{T-1} z_t^l z_t^{l\top} & Y^l &= \sum_{t=0}^{T-1} z_t^l x_{t+1}^{l\top} \\ \bar{V}^l &= \frac{1}{m_l}\sum_{k=1}^{m_l}\sum_{t=0}^{T-1} z_t^{l,k} z_t^{l,k\top} & \bar{Y}^l &= \frac{1}{m_l}\sum_{k=1}^{m_l}\sum_{t=0}^{T-1} z_t^{l,k} x_{t+1}^{l,k\top}, \end{aligned} \tag{18}$$

where $V^l, \bar{V}^l \in \mathbb{R}^{(n+m)\times(n+m)}$ and $Y^l, \bar{Y}^l \in \mathbb{R}^{(n+m)\times n}$. We denote the elements of $V^l, Y^l, \bar{V}^l, \bar{Y}^l$ as

$$V_{i,j}^l = \sum_{t=0}^{T-1} z_{t,i}^l z_{t,j}^l, i,j \in [n+m]$$

$$Y_{i,j}^l = \sum_{t=0}^{T-1} z_{t,i}^l x_{t+1,j}^l, i \in [n+m], j \in [n]$$

$$\bar{V}_{i,j}^l = \frac{1}{m_l}\sum_{k=1}^{m_l}\sum_{t=0}^{T-1} z_{t,i}^{l,k} z_{t,j}^{l,k}, i,j \in [n+m]$$

$$\bar{Y}_{i,j}^l = \frac{1}{m_l}\sum_{k=1}^{m_l}\sum_{t=0}^{T-1} z_{t,i}^{l,k} x_{t+1,j}^{l,k}, i \in [n+m], j \in [n].$$

Then we have the following result from Lemma 4.

**Lemma 5.** *There exist non-negative parameters $\iota$ and $\eta$ such that $\bar{V}_{i,j}^l, \bar{Y}_{i,j}^l \in SE\left(\frac{\iota^2}{m_l}, \frac{\eta}{m_l}\right)$ for all $i, j$ and $l \in [L]$.*

*Proof.* By Lemma 3 and Lemma 4, we know that every element of $z_t^{l,k} z_t^{l,k\top}$ and $z_t^{l,k} x_{t+1}^{l,k\top}$ are sub-exponential random variables. That is, $z_{t,i}^{l,k} z_{t,j}^{l,k} \in SE\left((\nu_{t,i,j})^2, \alpha_{t,i,j}\right), i, j \in [n+m]$ and $z_{t,i}^{l,k} x_{t+1,j}^{l,k} \in SE\left((\omega_{t,i,j})^2, \beta_{t,i,j}\right), i \in [n+m], j \in [n]$. The subexponential parameters can be chosen independent of $l$ and $k$ by the proof of Lemma 4. If we denote by

$$\nu_t = \max_{i,j} \nu_{t,i,j}, \qquad \alpha_t = \max_{i,j} \alpha_{t,i,j},$$
$$\omega_t = \max_{i,j} \omega_{t,i,j}, \qquad \beta_t = \max_{i,j} \beta_{t,i,j},$$

then we have $z_{t,i}^{l,k} z_{t,j}^{l,k} \in SE\left((\nu_t)^2, \alpha_t\right)$ for any $k \in [m_l], i, j \in [n+m]$ and $z_{t,i}^{l,k} x_{t+1,j}^{l,k} \in SE\left((\omega_t)^2, \beta_t\right)$ for any $k \in [m_l], i \in [n+m], j \in [n]$.

By Lemma 2, for non-independent sub-exponential random variables, we obtain

$$\sum_{t=0}^{T-1} z_{t,i}^{l,k} z_{t,j}^{l,k} \in SE\left(\left(\sum_{t=0}^{T-1} \nu_t\right)^2, \max_t \alpha_t\right),$$
$$\sum_{t=0}^{T-1} z_{t,i}^{l,k} x_{t+1,j}^{l,k} \in SE\left(\left(\sum_{t=0}^{T-1} \omega_t\right)^2, \max_t \beta_t\right).$$

Applying Lemma 2 again, but for independent sub-exponential random variables, we infer that

$$\bar{V}_{i,j}^l = \frac{1}{m_l} \sum_{k=1}^{m_l} \sum_{t=0}^{T-1} z_{t,i}^{l,k} z_{t,j}^{l,k} \in SE\left(\frac{\sum_{k=1}^{m_l}\left(\sum_{t=0}^{T-1} \nu_t\right)^2}{m_l^2}, \frac{\max_t \alpha_t}{m_l}\right)$$
$$= SE\left(\frac{\left(\sum_{t=0}^{T-1} \nu_t\right)^2}{m_l}, \frac{\max_t \alpha_t}{m_l}\right),$$

$$\bar{Y}_{i,j}^l = \frac{1}{m_l} \sum_{k=1}^{m_l} \sum_{t=0}^{T-1} z_{t,i}^{l,k} x_{t+1,j}^{l,k} \in SE\left(\frac{\sum_{k=1}^{m_l}\left(\sum_{t=0}^{T-1} \omega_t\right)^2}{m_l^2}, \frac{\max_t \beta_t^l}{m_l}\right)$$
$$= SE\left(\frac{\left(\sum_{t=0}^{T-1} \omega_t\right)^2}{m_l}, \frac{\max_t \beta_t}{m_l}\right).$$

The proof is complete by letting

$$\iota = \max\left\{\sqrt{\left(\sum_{t=0}^{T-1} \nu_t\right)^2}, \sqrt{\left(\sum_{t=0}^{T-1} \omega_t\right)^2}\right\},$$
$$\eta = \max\left\{\max_t \alpha_t, \max_t \beta_t\right\}.$$

$\square$

We can now derive the concentration inequalities for $\bar{V}^l$ and $\bar{Y}^l$.

**Lemma 6.** *Conditional on event $\mathcal{G}^l$, we can derive that for any $\zeta > 0$,*

$$\max\left\{\mathbb{P}\left(\left|\bar{V}^l - \mathbb{E}V^l\right| \geq \zeta\right), \mathbb{P}\left(\left|\bar{Y}^l - \mathbb{E}Y^l\right| \geq \zeta\right)\right\}$$
$$\leq 2(n+m)^2 \exp\left(-\min\left\{\frac{m_l\zeta^2}{2\iota^2(n+m)^4}, \frac{m_l\zeta}{2\eta(n+m)^2}\right\}\right), \tag{19}$$

*where $|\cdot|$ is a matrix norm that represents the summation of the absolute value of all the elements of the matrix, e.g. $|A| = \sum_{i,j}|a_{i,j}|$.*

*Proof.* We first consider one element of the matrix $\bar{V}^l$. By Lemma 1 and Lemma 5, we have

$$\mathbb{P}\left(\left|\bar{V}^l_{i,j} - \mathbb{E}V^l_{i,j}\right| \geq \zeta\right) \leq 2\exp\left(-\min\left\{\frac{m_l\zeta^2}{2\iota^2}, \frac{m_l\zeta}{2\eta}\right\}\right).$$

Then, by the fact that $\mathbb{P}\left(\sum_{i=1}^M |X_i| \geq \zeta\right) \leq \sum_{i=1}^M \mathbb{P}\left(|X_i| \geq \frac{\zeta}{M}\right)$ for all $M \in \mathbb{N}$ and random variables $(X_i)_{i=1}^M$, we can derive the concentration inequality for $\left|\bar{V}^l - \mathbb{E}V^l\right|$:

$$\mathbb{P}\left(\left|\bar{V}^l - \mathbb{E}V^l\right| \geq \zeta\right)$$
$$= \mathbb{P}\left(\sum_{i=1}^{n+m}\sum_{j=1}^{n+m} \left|\bar{V}^l_{i,j} - \mathbb{E}V^l_{i,j}\right| \geq \zeta\right)$$
$$\leq \sum_{i=1}^{n+m}\sum_{j=1}^{n+m} \mathbb{P}\left(\left|\bar{V}^l_{i,j} - \mathbb{E}V^l_{i,j}\right| \geq \frac{\zeta}{(n+m)^2}\right)$$
$$\leq 2(n+m)^2 \exp\left(-\min\left\{\frac{m_l\zeta^2}{2\iota^2(n+m)^4}, \frac{m_l\zeta}{2\eta(n+m)^2}\right\}\right).$$

Similarly, we can derive the concentration probability for $\bar{Y}^l$:

$$\mathbb{P}\left(\left|\bar{Y}^l - \mathbb{E}Y^l\right| \geq \zeta\right)$$
$$= \mathbb{P}\left(\sum_{i=1}^{n+m}\sum_{j=1}^{n} \left|\bar{Y}^l_{i,j} - \mathbb{E}Y^l_{i,j}\right| \geq \zeta\right)$$
$$\leq \sum_{i=1}^{n+m}\sum_{j=1}^{n} \mathbb{P}\left(\left|\bar{Y}^l_{i,j} - \mathbb{E}Y^l_{i,j}\right| \geq \frac{\zeta}{(n+m)n}\right)$$
$$\leq 2(n+m)n \exp\left(-\min\left\{\frac{m_l\zeta^2}{2\iota^2(n+m)^2n^2}, \frac{m_l\zeta}{2\eta(n+m)n}\right\}\right)$$
$$\overset{(1)}{\leq} 2(n+m)^2 \exp\left(-\min\left\{\frac{m_l\zeta^2}{2\iota^2(n+m)^4}, \frac{m_l\zeta}{2\eta(n+m)^2}\right\}\right),$$

where inequality (1) follows from the fact that $n + m \geq n$.

Finally, combining the two probability inequalities above, we can obtain (19). □

In order to derive the probability bounds in Proposition 3, we prove that $\left\|\left(\mathbb{E}[V^l]\right)^{-1}\right\|$ and $\left\|\mathbb{E}[Y^l]\right\|$ are bounded by a positive constant for any $\theta^l$ lies in $\Theta$. The boundedness of $\left\|\mathbb{E}[Y^l]\right\|$ can be proved directly, because $\mathbb{E}[Y^l]$ is continuous in terms of $\theta^l$ according to the definition of $Y^l$ in (18). In terms of $\left\|\left(\mathbb{E}[V^l]\right)^{-1}\right\|$, we will use the following lemma to show that it's bounded when $\theta^l \in \Theta$. Similar results can be found in Proposition 3.10 of Basei et al. (2022).

**Lemma 7.** *The following properties are equivalent:*

*1. For the sequence of the controller $K_t, t = 0, \cdots, T-1$ defined in (5),*

$$\left\{v \in \mathbb{R}^{n+m} \middle| \begin{bmatrix} I & K_t^\top \end{bmatrix} v = 0, \forall t = 0, \cdots, T-1\right\} = \{0\};$$

2. $\mathbb{E}[V] \succ 0$, where $V = \sum_{t=0}^{T-1} z_t z_t^\top$ is generated by the optimal policy in (5);

3. There exists $\lambda_0 > 0$ such that $\lambda_{\min}\left(\mathbb{E}\left[V^l\right]\right) \geq \lambda_0$ for any estimated $\theta^l \in \Theta$.

*Proof.* We first prove property 1 $\iff$ property 2.

For simplicity of notation, let $h_t = [I \ K_t^\top]^\top$ and $H = [h_0, h_1, \cdots, h_{T-1}]$. Property 1 is equivalent to that there exists no nonzero $v$ such that $H^\top v = 0$, which is also equivalent to that for any $v \neq 0$, $v^\top H H^\top v > 0$, i.e.

$$HH^\top = \sum_{t=0}^{T-1} h_t h_t^\top = \sum_{t=0}^{T-1} \left[ \begin{array}{c} I_n \\ K_t \end{array} \right] \left[ I_n \ K_t^\top \right] \succ 0.$$

One can readily compute that

$$\mathbb{E}[V] = \mathbb{E}\left[ \sum_{t=0}^{T-1} z_t z_t^\top \right]$$
$$= \sum_{t=0}^{T-1} \left[ \begin{array}{c} I_n \\ K_t \end{array} \right] \mathbb{E}\left[ x_t x_t^\top \right] \left[ I_n \ K_t^\top \right] = H \mathrm{diag}\left( \mathbb{E}\left[ x_0 x_0^\top \right], \cdots, \mathbb{E}\left[ x_{T-1} x_{T-1}^\top \right] \right) H^\top, \quad (20)$$

where $\mathrm{diag}(\cdot)$ is the notation of a diagonal block matrix. Next we show that $\mathbb{E}\left[ x_t x_t^\top \right]$ is positive definite for each $t$. Similar to (17), we can expand the system dynamics under the true system matrix $\theta = (A, B)$ as

$$x_t = \left( \prod_{i=t-1}^{0} (A + BK_i) \right) x_0 + \sum_{j=0}^{t-1} \left( \prod_{i=t-1}^{j+1} (A + BK_i) \right) w_j, \quad (21)$$

where $\prod_{i=t-1}^{j+1}(A + BK_i)$ means $(A + BK_{t-1})(A + BK_{t-2})\cdots(A + BK_{j+1})$, and $\prod_{i=t-1}^{t}(A + BK_i) = I_n$. For simplicity of notation, let

$$\Phi_{t_1,t_0} = (A + BK_{t_1})(A + BK_{t_1-1})\cdots(A + BK_{t_0}), \quad \text{for any } t_1 \geq t_0. \quad (22)$$

When $t_1 < t_0$, we set $\Phi_{t_1,t_0} = I_n$. Then we have $x_t = \Phi_{t-1,0} x_0 + \sum_{j=0}^{t-1} \Phi_{t-1,j+1} w_j$. It follows that

$$\mathbb{E}\left[ x_t x_t^\top \right] = \Phi_{t-1,0} \mathbb{E}\left[ x_0 x_0^\top \right] \Phi_{t-1,0}^\top + \sum_{j=0}^{t-1} \Phi_{t-1,j+1} \mathbb{E}\left[ w_j w_j^\top \right] \Phi_{t-1,j+1}^\top$$

$$\overset{(1)}{=} \Phi_{t-1,0} x_0 x_0^\top \Phi_{t-1,0}^\top + \sum_{j=1}^{t} \Phi_{t-1,j} \Phi_{t-1,j}^\top$$

$$\overset{(2)}{=} \Phi_{t-1,0} x_0 x_0^\top \Phi_{t-1,0}^\top + I_n + \sum_{j=1}^{t-1} \Phi_{t-1,j} \Phi_{t-1,j}^\top$$

$$\succeq I_n,$$

where the equality (1) follows from the fact that $w_j \sim \mathcal{N}\left(0, I_n\right), j = 0, \cdots, t-1$, and equality (2) holds by the fact that $\Phi_{t-1,t} = I_n$. Then, we can prove that property 1 is equivalent to that for any $v \neq 0$,

$$v^\top \mathbb{E}[V] v = v^\top H \mathrm{diag}\left( \mathbb{E}\left[ x_0 x_0^\top \right], \cdots, \mathbb{E}\left[ x_{T-1} x_{T-1}^\top \right] \right) H^\top v > 0,$$

which is equivalent to property 2.

We next prove property 2 $\iff$ property 3.

In terms of property 3 $\implies$ property 2, it's obvious that when $\theta^l = \theta$, we have $\lambda_{\min}\left(\mathbb{E}[V]\right) \geq \lambda_0 > 0$, i.e. $\mathbb{E}[V] \succ 0$.

In order to prove property 2 $\implies$ property 3, we prove the continuity of $\mathbb{E}[V]$ in terms of the system matrices $\theta$. By the recursive formula of the discrete-time Riccati equations and the optimal controller

in (5), we can find that $P_t, \widetilde{P}_t$ and $K_t$ are continuous in terms of $\theta \in \Theta$. Recall that

$$\mathbb{E}[x_t x_t^\top] = \Phi_{t-1,0} x_0 x_0^\top \Phi_{t-1,0}^\top + I_n + \sum_{j=1}^{t-1} \Phi_{t-1,j} \Phi_{t-1,j}^\top, \qquad (23)$$

where $\Phi_{t-1,j}$ is defined in (22). Plugging (23) into (20), we can see that $\mathbb{E}[V]$ is continuous in terms of $\theta$. So for any $\theta^l \in \Theta$, there exists $\lambda_0 > 0$ such that $\lambda_{\min}\left(\mathbb{E}[V^l]\right) \geq \lambda_0$. $\qquad \square$

Now we are ready for the proof of Proposition 3.

*Proof of Proposition 3.* Recall the definition of $\bar{V}^l, \bar{Y}^l, V^l$ and $Y^l$ in (18), we have

$$
\begin{aligned}
&\left\| \theta^{l+1} - \theta \right\| \\
&= \left\| \left( \bar{V}^l + \frac{1}{m_l} I_{n+m} \right)^{-1} \bar{Y}^l - \left( \mathbb{E}\left[V^l\right] \right)^{-1} \mathbb{E}\left[Y^l\right] \right\| \\
&\leq \left\| \left( \bar{V}^l + \frac{1}{m_l} I_{n+m} \right)^{-1} - \left( \mathbb{E}\left[V^l\right] \right)^{-1} \right\| \cdot \left\| \bar{Y}^l \right\| + \left\| \left( \mathbb{E}\left[V^l\right] \right)^{-1} \right\| \cdot \left\| \bar{Y}^l - \mathbb{E}[Y^l] \right\| \\
&\overset{(1)}{\leq} \left\| \left( \bar{V}^l + \frac{1}{m_l} I_{n+m} \right)^{-1} \right\| \cdot \left\| \left( \mathbb{E}\left[V^l\right] \right)^{-1} \right\| \cdot \left\| \bar{Y}^l \right\| \cdot \left\| \bar{V}^l + \frac{1}{m_l} I_{n+m} - \mathbb{E}\left[V^l\right] \right\| \\
&\quad + \left\| \left( \mathbb{E}\left[V^l\right] \right)^{-1} \right\| \cdot \left\| \bar{Y}^l - \mathbb{E}\left[Y^l\right] \right\| \\
&\overset{(2)}{\leq} \mathcal{C}_2 \left( \left\| \left( \bar{V}^l + \frac{1}{m_l} I_{n+m} \right)^{-1} \right\| \cdot \left\| \bar{Y}^l \right\| \cdot \left\| \bar{V}^l + \frac{1}{m_l} I_{n+m} - \mathbb{E}\left[V^l\right] \right\| + \left\| \bar{Y}^l - \mathbb{E}\left[Y^l\right] \right\| \right),
\end{aligned}
\qquad (24)
$$

where inequality (1) holds by the fact that $E^{-1} - F^{-1} = E^{-1}(F - E)F^{-1}$, inequality (2) follows from the results in Lemma 7 that $\left\| \left( \mathbb{E}[V^l] \right)^{-1} \right\| \leq \mathcal{C}_2$. By Lemma 6 and the equivalence of matrix norms, with probability at least $1 - 2\delta_l$, we have $\left\| \bar{V}^l - \mathbb{E}\left[V^l\right] \right\| \leq \Delta_l$ and $\left\| \bar{Y}^l - \mathbb{E}\left[Y^l\right] \right\| \leq \Delta_l$, where

$$\Delta_l := \max \left\{ \sqrt{\frac{2\iota^2 (n+m)^5 \log\left(\frac{(n+m)^2}{\delta_l}\right)}{m_l}}, \frac{2\eta (n+m)^{2.5} \log\left(\frac{(n+m)^2}{\delta_l}\right)}{m_l} \right\}.$$

For notational simplicity, we denote by

$$\mathcal{C}_4 := \max \left\{ \sqrt{2\iota^2 (n+m)^5}, 2\eta(n+m)^{2.5} \right\}.$$

$\mathcal{C}_4$ is a constant depending on $m, n$ polynomially and depending on $\gamma$ exponentially. For simplicity, we ignore the $T$-dependence of $\mathcal{C}_4$. Then, we have

$$\Delta_l \leq \mathcal{C}_4 \max \left\{ \sqrt{\frac{\log\left(\frac{(n+m)^2}{\delta_l}\right)}{m_l}}, \frac{\log\left(\frac{(n+m)^2}{\delta_l}\right)}{m_l} \right\}.$$

Now we can use $\Delta_l$ to further bound the terms in (24). Let $m_l$ be large enough so that $\Delta_l + \frac{1}{m_l} \leq \frac{1}{2\mathcal{C}_2}$, i.e. $m_l \geq \mathcal{C}_3 \log\left(\frac{(n+m)^2}{\delta_l}\right)$ for some constant $\mathcal{C}_3 \geq 1$. Then, with probability at least $1 - 2\delta_l$, we have $\left\| \bar{V}^l - \mathbb{E}\left[V^l\right] + \frac{1}{m_l} I_{n+m} \right\| \leq \Delta_l + \frac{1}{m_l} \leq \frac{1}{2\mathcal{C}_2}$, and thus

$$\lambda_{\min}\left( \bar{V}^l + \frac{1}{m_l} I_{n+m} \right) \geq \lambda_{\min}\left( \mathbb{E}\left[V^l\right] \right) - \left\| \bar{V}^l - \mathbb{E}\left[V^l\right] + \frac{1}{m_l} I_{n+m} \right\| \geq \frac{1}{2\mathcal{C}_2}.$$

Then, we get $\left\|\left(\bar{V}^l + \frac{1}{m_l}I_{n+m}\right)^{-1}\right\| \leq 2\mathcal{C}_2$. In terms of $\left\|\bar{Y}^l\right\|$, we have

$$\left\|\bar{Y}^l\right\| = \left\|\bar{Y}^l - \mathbb{E}\left[Y^l\right] + \mathbb{E}\left[Y^l\right]\right\| \overset{(3)}{\leq} \mathcal{C}_2 + \left\|\bar{Y}^l - \mathbb{E}\left[Y^l\right]\right\| \leq \mathcal{C}_2 + \Delta_l,$$

where inequality (3) follows from the fact that $\left\|\mathbb{E}\left[Y^l\right]\right\| \leq \mathcal{C}_2$. Finally, substituting all the elements into (24), we can get

$$\left\|\theta^{l+1} - \theta\right\|$$

$$\leq \mathcal{C}_2\left(2\mathcal{C}_2 \cdot \left(\mathcal{C}_2 + \left\|\bar{Y}^l - \mathbb{E}\left[Y^l\right]\right\|\right) \cdot \left(\Delta_l + \frac{1}{m_l}\right) + \Delta_l\right)$$

$$\leq \mathcal{C}_2\left(2\mathcal{C}_2 \cdot \left(\mathcal{C}_2 + \Delta_l\right) \cdot \left(\Delta_l + \frac{1}{m_l}\right) + \Delta_l\right)$$

$$\overset{(4)}{\leq} 2\mathcal{C}_2^3\left((1 + \Delta_l)\left(\Delta_l + \frac{1}{m_l}\right) + \Delta_l\right)$$

$$\overset{(5)}{\leq} 8\mathcal{C}_2^3\left(\Delta_l + \Delta_l^2 + \frac{1}{m_l}\right)$$

$$\overset{(6)}{\leq} 16\mathcal{C}_2^3\mathcal{C}_4^2\left(\sqrt{\frac{\log\left(\frac{(n+m)^2}{\delta_l}\right)}{m_l}} + \frac{\log\left(\frac{(n+m)^2}{\delta_l}\right)}{m_l} + \frac{\log^2\left(\frac{(n+m)^2}{\delta_l}\right)}{m_l^2}\right),$$

where inequality (4) follows from $\mathcal{C}_2 \geq 1$, inequality (5) holds by the fact that $m_l \geq 1$ and inequality (6) holds because $\log\left(\frac{(n+m)^2}{\delta_l}\right) \geq 1$. The proof is hence complete. $\qquad\square$

Lemma 7 shows that Assumption 1 can be extended to the neighbourhood of the true system matrices $\theta$, and thus guarantee the well-posedness of the sample variance of the estimated system matrices within the neighbourhood. The following proposition provides a sufficient condition for Assumption 1.

**Proposition 4.** *If the parameters defined in Section 2.1 satisfies*

1. *$A \in \mathbb{R}^{n \times n}$ has full rank;*

2. *$Q \succ 0$ and $Q_T = 0$;*

3. *$B \in \mathbb{R}^{n \times m}$ has full column rank,*

*then for the sequence of the controller $K_t, t = 0, \cdots, T-1$ defined in (5), we have*

$$\left\{v \in \mathbb{R}^{n+m}\middle|[I \; K_t^\top]v = 0, \forall t = 0, \cdots, T-1\right\} = \{0\}.$$

*Proof.* Let $v = [v_1^\top \; v_2^\top]^\top$ satisfying $[I \; K_t^\top]v = 0, \forall t = 0, \cdots, T-1$, where $v_1 \in \mathbb{R}^n$, $v_2 \in \mathbb{R}^m$. Recall the optimal control defined in (5), by the condition $Q_T = 0$, we have $K_{T-1} = 0$, and thus $v_1 = 0$. Then, $[I \; K_t^\top]v = 0, \forall t = 0, \cdots, T-1$ is equivalent to $K_t^\top v_2 = 0, \forall t = 0, \cdots, T-1$. Substitute (5) into it, we can obtain

$$K_t^\top v_2 = -A^\top \widetilde{P}_{t+1}B\left(B^\top \widetilde{P}_{t+1}B + R\right)^{-1}v_2 = 0.$$

Recall that

$$\widetilde{P}_{t+1} = P_{t+1} + \gamma P_{t+1}\left(I_n - \gamma P_{t+1}\right)^{-1}P_{t+1},$$
$$P_t = Q + K_t^\top R K_t + (A + BK_t)^\top \widetilde{P}_{t+1}(A + BK_t), t = 0, \cdots, T-1.$$

We can prove that $P_t \succ 0$ for any $t = 0, \cdots, T-1$ by the mathematical induction. When $t = T$, $\widetilde{P}_T = 0$, and thus $P_{T-1} \succ 0$ by $Q \succ 0$ and $K_{T-1}^\top R K_{T-1} \succeq 0$. For any $t = 1, \cdots, T-1$,

assume that $P_{t+1} \succ 0$, we can prove that $\widetilde{P}_{t+1} \succ 0$, and thus $P_t \succ 0$ by $Q \succ 0$, $K_t^\top R K_t \succeq 0$ and $(A + BK_t)^\top \widetilde{P}_{t+1}(A + BK_t) \succeq 0$, which finish the mathematical induction. And by the condition $A \in \mathbb{R}^{n \times n}$ has full rank, we have

$$B \left( B^\top \widetilde{P}_{t+1} B + R \right)^{-1} v_2 = 0.$$

According to the setting in Section 2.1, $R \succ 0$, so $B^\top \widetilde{P}_{t+1} B + R \succ 0$. So $B \left( B^\top \widetilde{P}_{t+1} B + R \right)^{-1}$ has full column rank by the condition that $B \in \mathbb{R}^{n \times m}$ has full column rank, and thus $v_2 = 0$, which completes the proof. $\qquad \square$

### A.2 PERTURBATION ANALYSIS OF RICCATI EQUATION

In this section, we discuss perturbation analysis of Riccati Equation, i.e., how the solutions to Riccati Equation (5) change when we perturb the system matrices.

The main result of this section is the following lemma. We fix epoch $l$ in the analysis below and recall that $(A^l, B^l)$ are the estimators for the true system matrices $(A, B)$.

**Lemma 8.** *Assume $1 - \gamma \widetilde{\Gamma} > 0$ and fix any $\epsilon_l > 0$. Suppose $\|A^l - A\| \le \epsilon_l, \|B^l - B\| \le \epsilon_l$, then for any $t = 0, 1, \cdots, T - 1$, we have*

$$\|K_t^l - K_t\| \le (10\mathcal{V}^2 \mathcal{L} \widetilde{\Gamma}^4)^{T-t-1} \mathcal{V} \epsilon_l,$$
$$\|P_t^l - P_t\| \le (10\mathcal{V}^2 \mathcal{L} \widetilde{\Gamma}^4)^{T-t} \epsilon_l,$$

*where $\widetilde{\Gamma}, \mathcal{V}$ and $\mathcal{L}$ are defined in (13).*

To prove Lemma 8, we need the following result, which provides 'one-step' perturbation bounds for the solutions to Riccati equations.

**Lemma 9.** *Assume $1 - \gamma \widetilde{\Gamma} > 0$. For any $\epsilon_l > 0, W \ge 1$, assume $\|A^l - A\| \le \epsilon_l, \|B^l - B\| \le \epsilon_l$ and $\|P_{t+1}^l - P_{t+1}\| \le W \epsilon_l \le 1$ for a given $t \in \{0, \cdots, T - 1\}$. Then we have*

$$\|K_t^l - K_t\| \le \mathcal{V} W \epsilon_l,$$
$$\|P_t^l - P_t\| \le 10\mathcal{V}^2 \mathcal{L} \widetilde{\Gamma}^4 W \epsilon_l,$$

*where $\widetilde{\Gamma}, \mathcal{V}$ and $\mathcal{L}$ are given in (13).*

*Proof.* We first bound the perturbation of the optimal controller, i.e., $\Delta K_t^l = K_t^l - K_t$. Recall that

$$K_t = -(B^\top \widetilde{P}_{t+1} B + R)^{-1} B^\top \widetilde{P}_{t+1} A, \quad \text{and} \quad K_t^l = - \left( B^{l\top} \widetilde{P}_{t+1}^l B^l + R \right)^{-1} B^{l\top} \widetilde{P}_{t+1}^l A^l. \tag{25}$$

To bound $\Delta K_t^l$, we first bound $\left\| \widetilde{P}_{t+1}^l - \widetilde{P}_{t+1} \right\|$ as follows:

$$\left\| \widetilde{P}_{t+1}^l - \widetilde{P}_{t+1} \right\|$$
$$\overset{(1)}{=} \left\| (I_n - \gamma P_{t+1}^l)^{-1} P_{t+1}^l - (I_n - \gamma P_{t+1})^{-1} P_{t+1} \right\|$$
$$= \left\| (I_n - \gamma P_{t+1}^l)^{-1} P_{t+1}^l - (I_n - \gamma P_{t+1})^{-1} P_{t+1}^l + (I_n - \gamma P_{t+1})^{-1} P_{t+1}^l - (I_n - \gamma P_{t+1})^{-1} P_{t+1} \right\|$$
$$\le \left\| (I_n - \gamma P_{t+1}^l)^{-1} P_{t+1}^l - (I_n - \gamma P_{t+1})^{-1} P_{t+1}^l \right\| + \left\| (I_n - \gamma P_{t+1})^{-1} P_{t+1}^l - (I_n - \gamma P_{t+1})^{-1} P_{t+1} \right\|$$
$$\le \left\| (I_n - \gamma P_{t+1}^l)^{-1} - (I_n - \gamma P_{t+1})^{-1} \right\| \cdot \|P_{t+1}^l\| + \left\| (I_n - \gamma P_{t+1})^{-1} \right\| \cdot \|P_{t+1}^l - P_{t+1}\|.$$

Here, the equality (1) follows by the definition of $\widetilde{P}_{t+1}^l$ and $\widetilde{P}_{t+1}$, and the fact that

$$\widetilde{P}_t = P_t + \gamma P_t (I_n - \gamma P_t)^{-1} P_t$$
$$= \left[ I_n + \gamma P_t (I - \gamma P_t)^{-1} \right] P_t$$
$$= \left[ (I_n - \gamma P_t)(I_n - \gamma P_t)^{-1} + \gamma P_t (I_n - \gamma P_t)^{-1} \right] P_t$$
$$= (I_n - \gamma P_t)^{-1} P_t.$$

It follows from the fact that $E^{-1} - F^{-1} = E^{-1}(F - E)F^{-1}$ for any invertible matrix $E$ and $F$,

$$\left\| \widetilde{P}_{t+1}^l - \widetilde{P}_{t+1} \right\|$$

$$\leq \left\| (I_n - \gamma P_{t+1}^l)^{-1} \gamma (P_{t+1}^l - P_{t+1})(I_n - \gamma P_{t+1})^{-1} \right\| \cdot \| P_{t+1}^l \| + \| (I_n - \gamma P_{t+1})^{-1} \| \cdot \| P_{t+1}^l - P_{t+1} \|$$

$$\overset{(2)}{\leq} \frac{1}{1 - \gamma \| P_{t+1}^l \|} \cdot \frac{1}{1 - \gamma \| P_{t+1} \|} \cdot \gamma \| P_{t+1}^l - P_{t+1} \| \cdot \| P_{t+1}^l \| + \frac{1}{1 - \gamma \| P_{t+1} \|} \cdot \| P_{t+1}^l - P_{t+1} \|$$

$$\leq \frac{1}{1 - \gamma (W \epsilon_l + \Gamma)} \cdot \frac{1}{1 - \gamma \Gamma} \cdot \gamma W \epsilon_l (W \epsilon_l + \Gamma) + \frac{1}{1 - \gamma \Gamma} \cdot W \epsilon_l$$

$$\overset{(3)}{\leq} \frac{1}{(1 - \gamma \widetilde{\Gamma})^2} \gamma \widetilde{\Gamma} W \epsilon_l + \frac{1}{1 - \gamma \widetilde{\Gamma}} W \epsilon_l$$

$$= \left[ \frac{\gamma \widetilde{\Gamma}}{(1 - \gamma \widetilde{\Gamma})^2} + \frac{1}{1 - \gamma \widetilde{\Gamma}} \right] W \epsilon_l$$

$$= \frac{W \epsilon_l}{(1 - \gamma \widetilde{\Gamma})^2},$$

where the inequality (2) holds by the fact that for any matrix $E \in R^{n \times n}$, if $\| E \| < 1$, then $\| (I_n - E)^{-1} \| \leq \frac{1}{1 - \| E \|}$, and the inequality (3) holds because we assume that $\| P_{t+1}^l - P_{t+1} \| \leq W \epsilon_l \leq 1$.

To bound $\Delta K_t^l$, we next bound $\left\| B^\top \widetilde{P}_{t+1} B - B^{l\top} \widetilde{P}_{t+1}^l B^l \right\|$ in view of the expressions in (25):

$$\left\| B^\top \widetilde{P}_{t+1} B - B^{l\top} \widetilde{P}_{t+1}^l B^l \right\|$$

$$\leq \left\| B^\top \widetilde{P}_{t+1} B - B^\top \widetilde{P}_{t+1} B^l \right\| + \left\| B^\top \widetilde{P}_{t+1} B^l - B^\top \widetilde{P}_{t+1}^l B^l \right\| + \left\| B^\top \widetilde{P}_{t+1}^l B^l - B^{l\top} \widetilde{P}_{t+1}^l B^l \right\|$$

$$\leq \| B^\top \widetilde{P}_{t+1} \| \cdot \| B - B^l \| + \| B \| \cdot \| \widetilde{P}_{t+1} - \widetilde{P}_{t+1}^l \| \cdot \| B^l \| + \| B - B^l \| \cdot \| \widetilde{P}_{t+1}^l B^l \|$$

$$\leq \epsilon_l \Gamma^2 + \Gamma \mathcal{L} W \epsilon_l (\Gamma + \epsilon_l) + \epsilon_l (\mathcal{L} W \epsilon_l + \Gamma)(\Gamma + \epsilon_l)$$

$$\overset{(4)}{\leq} W \epsilon_l (\widetilde{\Gamma}^2 + \widetilde{\Gamma}^2 \mathcal{L} + (\epsilon_l \mathcal{L} + \Gamma) \widetilde{\Gamma})$$

$$\leq 2(\mathcal{L} + 1) \widetilde{\Gamma}^2 W \epsilon_l,$$

where inequality (4) holds by the fact that $W \epsilon_l \leq 1$. Similarly, we can derive that

$$\left\| B^\top \widetilde{P}_{t+1} A - B^{l\top} \widetilde{P}_{t+1}^l A^l \right\|$$

$$\leq \left\| B^\top \widetilde{P}_{t+1} A - B^\top \widetilde{P}_{t+1} A^l \right\| + \left\| B^\top \widetilde{P}_{t+1} A^l - B^\top \widetilde{P}_{t+1}^l A^l \right\| + \left\| B^\top \widetilde{P}_{t+1}^l A^l - B^{l\top} \widetilde{P}_{t+1}^l A^l \right\|$$

$$\leq 2(\mathcal{L} + 1) \widetilde{\Gamma}^2 W \epsilon_l.$$

Then, following a similar argument as in Lemma 2 of Mania et al. (2019), we can obtain

$$\| \Delta K_t^l \| = \| K_t - K_t^l \| \leq 2(L + 1) \widetilde{\Gamma}^3 W \epsilon_l.$$

Next we proceed to bound $\| P_t^l - P_t \|$. Recall that

$$P_t = Q + K_t^\top R K_t + (A + B K_t)^\top \widetilde{P}_{t+1}(A + B K_t),$$

$$P_t^l = Q + K_t^{l\top} R K_t^l + (A^l + B^l K_t^l)^\top \widetilde{P}_{t+1}^l (A^l + B^l K_t^l).$$

We can directly compute that

$$\left\| A + B K_t - A^l - B^l K_t^l \right\|$$

$$\leq \left\| A - A^l \right\| + \left\| B K_t - B K_t^l \right\| + \left\| B K_t^l - B^l K_t^l \right\|$$

$$\leq \epsilon_l + \Gamma \mathcal{V} W \epsilon_l + \epsilon_l (\mathcal{V} W \epsilon_l + \Gamma)$$

$$\overset{(5)}{\leq} \epsilon_l + \Gamma \mathcal{V} W \epsilon_l + \widetilde{\Gamma} \mathcal{V} W \epsilon_l$$

$$\leq 2 \mathcal{V} \widetilde{\Gamma} W \epsilon_l,$$

where inequality (5) holds by the fact that $\epsilon_l(\mathcal{V}W\epsilon_l + \Gamma) \leq \mathcal{V}W\epsilon_l + \Gamma\epsilon_l W\mathcal{V}$ when both $W$ and $\mathcal{V}$ are larger than 1. Similarly, we can derive that

$$
\begin{aligned}
&\left\| K_t^\top R K_t - K_t^{l\top} R K_t^l \right\| \\
&\leq \left\| K_t^\top R K_t - K_t^\top R K_t^l \right\| + \left\| K_t^\top R K_t^l - K_t^{l\top} R K_t^l \right\| \\
&\leq \Gamma^2 \mathcal{V}W\epsilon_l + \mathcal{V}W\epsilon_l \Gamma (\mathcal{V}W\epsilon_l + \Gamma) \\
&\leq 2\mathcal{V}^2 \widetilde{\Gamma}^2 W \epsilon_l.
\end{aligned}
$$

In addition, we can derive that

$$
\begin{aligned}
&\left\| (A + BK_t)^\top \widetilde{P}_{t+1}(A + BK_t) - (A + BK_t^l)^\top \widetilde{P}_{t+1}^l (A + BK_t^l) \right\| \\
&\leq \left\| (A + BK_t)^\top \widetilde{P}_{t+1}(A + BK_t) - (A + BK_t^l)^\top \widetilde{P}_{t+1}(A + BK_t) \right\| \\
&\quad + \left\| (A + BK_t^l)^\top \widetilde{P}_{t+1}(A + BK_t) - (A + BK_t^l)^\top \widetilde{P}_{t+1}^l (A + BK_t) \right\| \\
&\quad + \left\| (A + BK_t^l)^\top \widetilde{P}_{t+1}^l (A + BK_t) - (A + BK_t^l)^\top \widetilde{P}_{t+1}^l (A + BK_t^l) \right\| \\
&\leq 2\mathcal{V}\widetilde{\Gamma}^4 W \epsilon_l + 2\mathcal{L}\mathcal{V}\widetilde{\Gamma}^4 W \epsilon_l + 4\mathcal{V}^2 \mathcal{L}\widetilde{\Gamma}^4 W \epsilon_l \\
&\leq 8\mathcal{V}^2 \mathcal{L}\widetilde{\Gamma}^4 W \epsilon_l.
\end{aligned}
$$

It then follows that

$$
\| P_t^l - P_t \| \leq 10\mathcal{V}^2 \mathcal{L}\widetilde{\Gamma}^4 W \epsilon_l.
$$

The proof is therefore complete. $\qquad\square$

With Lemma 9, we are now ready to prove Lemma 8.

*Proof of Lemma 8.* By definition we know that $P_T^l = P_T = Q_T$, and thus we have $\| P_T^l - P_T \| \leq \epsilon_l$. By Lemma 9, we can derive that at time $T - 1$,

$$
\begin{aligned}
\| K_{T-1}^l - K_{T-1} \| &\leq \mathcal{V}\epsilon_l, \\
\| P_{T-1}^l - P_{T-1} \| &\leq (10\mathcal{V}^2 \mathcal{L}\widetilde{\Gamma}^4)\epsilon_l,
\end{aligned}
$$

which implies that

$$
\begin{aligned}
\| K_{T-2}^l - K_{T-2} \| &\leq (10\mathcal{V}^2 \mathcal{L}\widetilde{\Gamma}^4)\mathcal{V}\epsilon_l, \\
\| P_{T-2}^l - P_{T-2} \| &\leq (10\mathcal{V}^2 \mathcal{L}\widetilde{\Gamma}^4)^2 \epsilon_l.
\end{aligned}
$$

Applying Lemma 9 recursively, we obtain for any $t = 0, \cdots, T - 1$.

$$
\begin{aligned}
\| K_t^l - K_t \| &\leq (10\mathcal{V}^2 \mathcal{L}\widetilde{\Gamma}^4)^{T-t-1}\mathcal{V}\epsilon_l, \\
\| P_t^l - P_t \| &\leq (10\mathcal{V}^2 \mathcal{L}\widetilde{\Gamma}^4)^{T-t}\epsilon_l,
\end{aligned}
$$

which completes the proof. $\qquad\square$

### A.3 Suboptimality Gap Due to the Controller Mismatch

In this section, we will simplify the performance gap between the total cost under policy $\pi^{l,k}$ and the total cost under the optimal policy. We recall the corresponding total cost under entropic risk,

$$
J_0^{\pi^{l,k}}\left( x_0^{l,k} \right) = \frac{1}{\gamma} \log \mathbb{E} \exp\left( \frac{\gamma}{2} \left( \sum_{t=0}^{T-1} \left( x_t^{l,k\top} Q x_t^{l,k} + u_t^{l,k\top} R u_t^{l,k} \right) + x_T^{l,k\top} Q_T x_T^{l,k} \right) \right),
$$

where $u_t^{l,k} = K_t^l x_t^{l,k}$, and $K_t^l$ is obtained by substituting $(A^l, B^l)$ into equation 5.

Let $\mathbb{H}_t^{l,k}$ be the set of possible histories up to the $t$-th step in the $k$-th episode of epoch $l$. Then, one sample of the history up to the $t$-th step in the $k$-th episode of epoch $l$ is

$$
H_t^{l,k} = \left( x_0^{1,1}, u_0^{1,1}, \cdots, x_T^{1,1}, x_0^{1,2}, \cdots, x_0^{2,1}, \cdots, x_T^{2,1}, \cdots, x_0^{l,k}, \cdots, x_t^{l,k}, u_t^{l,k} \right).
$$

We also introduce some new notations, which will be heavily used in the regret analysis. For any $t = 1, \cdots, T - 2$, we define the following recursive equations:

$$
\begin{aligned}
D^l_{T-1} &= \Delta K^{l\top}_{T-1}(R + B^\top \widetilde{P}_T B)\Delta K^l_{T-1}, \\
\widetilde{D}^l_{T-1} &= (I_n - \gamma D^l_{T-1})^{-1} D^l_{T-1}, \\
D^l_t &= \Delta K^{l\top}_t \left( R + B^\top \widetilde{P}_{t+1} B \right) \Delta K^l_t + (A + BK^l_t)^\top \widetilde{D}^l_{t+1}(A + BK^l_t), \\
\widetilde{D}^l_t &= (I_n - \gamma D^l_t)^{-1} D^l_t, \\
D^l_0 &= \Delta K^{l\top}_0 \left( R + B^\top \widetilde{P}_1 B \right) \Delta K^l_0 + (A + BK^l_0)^\top \widetilde{D}^l_1(A + BK^l_0),
\end{aligned}
\tag{26}
$$

where $\Delta K^l_t = K^l_t - K_t$ and $\widetilde{P}_T$ is defined in (5). In the following parts, we still consider the risk-averse setting, where $\gamma > 0$. The following proposition is the key result of this section.

**Proposition 5.** *We can simplify the performance gap in the k-th episode of epoch l to*

$$
J^{\pi^{l,k}}_0(x^{l,k}_0) - J^\star_0(x^{l,k}_0) = -\frac{1}{2\gamma} \sum_{t=1}^{T-1} \log \left( \det \left( I_n - \gamma D^l_t \right) \right) + \frac{1}{2} x^{l,k\top}_0 D^l_0 x^{l,k}_0.
\tag{27}
$$

*where $D^l_t$ is defined in (26).*

In order to prove Proposition 5, we introduce Lemma 10, see p.8 of Jacobson (1973).

**Lemma 10** (Jacobson (1973)). *Consider the linear dynamic system $x_{t+1} = Ax_t + Bu_t + w_t, w_t \sim \mathcal{N}(0, I_n), t = 0, \cdots, T - 1$. For any sequence of positive semidefinite matrix $E_{t+1}$ satisfying $I_n - \gamma E_{t+1} \succ 0$, we have*

$$
\mathbb{E} \left[ \exp \left( \frac{\gamma}{2} x^\top_{t+1} E_{t+1} x_{t+1} \right) \Big| x_t, u_t \right]
$$
$$
= (\det(I_n - \gamma E_{t+1}))^{-\frac{1}{2}} \exp \left( \frac{\gamma}{2}(Ax_t + Bu_t)^\top \widetilde{E}_{t+1}(Ax_t + Bu_t) \right),
$$

*where $\widetilde{E}_{t+1} = E_{t+1} + \gamma E_{t+1}(I_n - \gamma E_{t+1})^{-1} E_{t+1}$.*

We apply Lemma 10 to simplify the performance gap $J^{\pi^{l,k}}_0(x^{l,k}_0) - J^\star_0(x^{l,k}_0)$ in the $k$-th episode of epoch $l$ in the following lemma.

**Lemma 11.** *We can simplify the performance gap as*

$$
J^{\pi^{l,k}}_0(x^{l,k}_0) - J^\star_0(x^{l,k}_0) = \frac{1}{\gamma} \log \mathbb{E} \left[ \exp \left( \frac{\gamma}{2} \sum_{t=0}^{T-1} x^{l,k\top}_t \Delta K^{l\top}_t (R + B^\top \widetilde{P}_{t+1} B)\Delta K^l_t x^{l,k}_t \right) \Big| x^{l,k}_0, H^{l,k-1}_T \right],
\tag{28}
$$

*where $\Delta K^l_t = K^l_t - K_t$.*

*Proof.* Denote $J_t(x^{l,k}_t) = \frac{1}{2} \left( x^{l,k\top}_t P_t x^{l,k}_t - \sum_{i=t}^{T-1} \frac{1}{\gamma} \log \det (I - \gamma P_{t+1}) \right), t = 0, \cdots, T - 1$, which is the dynamic programming equations of LEQR problem. When $t = T$, $J_T(x^{l,k}_T) = x^{l,k\top}_T Q_T x^{l,k}_T$.

By the definition of $J^{\pi^{l,k}}_0(x^{l,k}_0)$ and $J^\star_0(x^{l,k}_0)$, we have

$$
J^{\pi^{l,k}}_0(x^{l,k}_0) - J^\star_0(x^{l,k}_0)
$$
$$
= \frac{1}{\gamma} \log \mathbb{E} \left[ \exp \left( \frac{\gamma}{2} \left( \sum_{t=0}^{T-1} \left( x^{l,k\top}_t Q x^{l,k}_t + u^{l,k\top}_t R u^{l,k}_t \right) + x^{l,k\top}_T Q_T x^{l,k}_T \right) \right) \Big| x^{l,k}_0, H^{l,k-1}_T \right] - J_0(x^{l,k}_0)
$$
$$
= \frac{1}{\gamma} \log \mathbb{E} \left[ \exp \left( \frac{\gamma}{2} \left( \sum_{t=0}^{T-1} \left( \left( x^{l,k\top}_t Q x^{l,k}_t + u^{l,k\top}_t R u^{l,k}_t \right) + J_t(x^{l,k}_t) - J_t(x^{l,k}_t) \right) \right. \right. \right.
$$
$$
\left. \left. \left. + x^{l,k\top}_T Q_T x^{l,k}_T \right) \right) \Big| x^{l,k}_0, H^{l,k-1}_T \right] - J_0(x^{l,k}_0).
$$

$$
\tag{29}
$$

Recall that $J_T(x_T^{l,k}) = x_T^{l,k\top} Q_T x_T^{l,k}$, we have

$$J_0^{\pi^{l,k}}(x_0^{l,k}) - J_0^{\star}(x_0^{l,k})$$

$$= \frac{1}{\gamma} \log \mathbb{E}\left[ \exp\left( \frac{\gamma}{2}\left( \sum_{t=0}^{T-1} \left( \left( x_t^{l,k\top} Q x_t^{l,k} + u_t^{l,k\top} R u_t^{l,k} \right) + J_t(x_t^{l,k}) - J_t(x_t^{l,k}) \right) \right.\right.\right.$$

$$\left.\left.\left. + J_T(x_T^{l,k}) \right) \right) \middle| x_0^{l,k}, H_T^{l,k-1} \right] - \frac{1}{\gamma} \log\left( \exp(J_0(x_0^{l,k})) \right)$$

$$\overset{(1)}{=} \frac{1}{\gamma} \log \mathbb{E}\left[ \exp\left( \frac{\gamma}{2} \sum_{t=0}^{T-1} \left( \left( x_t^{l,k\top} Q x_t^{l,k} + u_t^{l,k\top} R u_t^{l,k} \right) + J_{t+1}(x_{t+1}^{l,k}) - J_t(x_t^{l,k}) \right) \right) \middle| x_0^{l,k}, H_T^{l,k-1} \right]$$

$$\overset{(2)}{=} \frac{1}{\gamma} \log \mathbb{E}\left[ \exp\left( \gamma \sum_{t=0}^{T-1} \left( \frac{1}{2} x_t^{l,k\top}(Q + K_t^{l\top} R K_t^l) x_t^{l,k} + \frac{1}{2} x_{t+1}^{l,k\top} P_{t+1} x_{t+1}^{l,k} - \frac{1}{2} x_t^{l,k\top} P_t x_t^{l,k} \right.\right.\right.$$

$$\left.\left.\left. + \frac{1}{2\gamma} \log \det(I_n - \gamma P_{t+1}) \right) \right) \middle| x_0^{l,k}, H_T^{l,k-1} \right],$$

where equality (1) holds by canceling out the $J_0(x_0^{l,k})$ inside and outside the entropic risk, and equality (2) follows from the definition of the total cost under entropic risk and $u_t^{l,k} = K_t^l x_t^{l,k}$. By the law of total expectation, i.e. $\mathbb{E}[X|Z] = \mathbb{E}[\mathbb{E}[X|Y,Z]|Z]$ for any random variables $X, Y, Z$, we consider the conditional expectation

$$\mathbb{E}\left[ \exp\left( \gamma\left( \frac{1}{2} x_t^{l,k\top}(Q + K_t^{l\top} R K_t^l) x_t^{l,k} + \frac{1}{2} x_{t+1}^{l,k\top} P_{t+1} x_{t+1}^{l,k} - \frac{1}{2} x_t^{l,k\top} P_t x_t^{l,k} \right.\right.\right.$$

$$\left.\left.\left. + \frac{1}{2\gamma} \log \det(I_n - \gamma P_{t+1}) \right) \right) \middle| H_t^{l,k} \right]$$

$$= \exp\left( \gamma\left( \frac{1}{2} x_t^{l,k\top}(Q + K_t^{l\top} R K_t^l) x_t^{l,k} - \frac{1}{2} x_t^{l,k\top} P_t x_t^{l,k} + \frac{1}{2\gamma} \log \det(I_n - \gamma P_{t+1}) \right) \right)$$

$$\times \mathbb{E}\left[ \exp\left( \frac{\gamma}{2} x_{t+1}^{l,k\top} P_{t+1} x_{t+1}^{l,k} \right) \middle| H_t^{l,k} \right] \tag{30}$$

$$\overset{(3)}{=} \exp\left( \gamma\left( \frac{1}{2} x_t^{l,k\top}(Q + K_t^{l\top} R K_t^l) x_t^{l,k} - \frac{1}{2} x_t^{l,k\top} P_t x_t^{l,k} + \frac{1}{2\gamma} \log \det(I_n - \gamma P_{t+1}) \right) \right)$$

$$\times (\det(I_n - \gamma P_{t+1}))^{-1/2} \exp\left[ \frac{\gamma}{2}\left( x_t^{l,k\top}(A + B K_t^l)^\top \widetilde{P}_{t+1}(A + B K_t^l) x_t^{l,k} \right) \right]$$

$$= \exp\left[ \frac{\gamma}{2}\left( x_t^{l,k\top}(Q + K_t^{l\top} R K_t^l + (A + B K_t^l)^\top \widetilde{P}_{t+1}(A + B K_t^l)) x_t^{l,k} - x_t^{l,k\top} P_t x_t^{l,k} \right) \right],$$

where the equality (3) follows from Lemma 10.

Recall that $\Delta K_t^l = K_t^l - K_t$ and $P_t = Q + K_t^\top R K_t + (A + B K_t)^\top \widetilde{P}_{t+1}(A + B K_t)$. Then the RHS of Equation (30) becomes

$$\exp\left[ \frac{\gamma}{2} x_t^{l,k\top}\left( Q + (\Delta K_t^l + K_t)^\top R(\Delta K_t^l + K_t) \right.\right.$$

$$\left.\left. + (A + B(\Delta K_t^l + K_t))^\top \widetilde{P}_{t+1}(A + B(\Delta K_t^l + K_t))) x_t^{l,k} - \frac{\gamma}{2} x_t^{l,k\top} P_t x_t^{l,k} \right]$$

$$= \exp\left[ \frac{\gamma}{2} x_t^{l,k\top} \Delta K_t^{l\top}(R + B^\top \widetilde{P}_{t+1} B) \Delta K_t^l x_t^{l,k} + \gamma x_t^{l,k\top} \Delta K_t^{l\top}\left( (R + B^\top \widetilde{P}_{t+1} B) K_t + B^\top \widetilde{P}_{t+1} A \right) x_t^{l,k} \right]$$

$$\overset{(4)}{=} \exp\left[ \frac{\gamma}{2} x_t^{l,k\top} \Delta K_t^{l\top}(R + B^\top \widetilde{P}_{t+1} B) \Delta K_t^l x_t^{l,k} \right],$$

$$\tag{31}$$

where the equality (4) holds by the fact that $K_t = -(R + B^\top \widetilde{P}_{t+1} B)^{-1} B^\top \widetilde{P}_{t+1} A$. Finally, substituting (31) into (30) and then substituting (30) into (29), we can get (28). $\qquad\square$

With Lemma 11, we are now ready to prove Proposition 5.

*Proof of Proposition 5.* We prove the result recursively. When $t = T - 1$, we have

$$\mathbb{E}\left[\exp\left(\frac{\gamma}{2}x_{T-1}^{l,k\top}\Delta K_{T-1}^{l\top}(R + B^\top \widetilde{P}_T B)\Delta K_{T-1}^l x_{T-1}^{l,k}\right)\Big| H_{T-2}^{l,k}\right]$$

$$= \mathbb{E}\left[\exp\left(\frac{\gamma}{2}x_{T-1}^{l,k\top}D_{T-1}^l x_{T-1}^{l,k}\right)\Big| H_{T-2}^{l,k}\right]$$

$$\stackrel{(1)}{=} \left(\det(I_n - \gamma D_{T-1}^l)\right)^{-\frac{1}{2}}\exp\left\{\frac{\gamma}{2}\left[x_{T-2}^{l,k\top}(A + BK_{T-2}^l)^\top \widetilde{D}_{T-1}^l(A + BK_{T-2}^l)x_{T-2}^{l,k}\right]\right\},$$

where equality (1) follows from Lemma 10 and $u_{T-1}^{l,k} = K_{T-1}^l x_{T-1}^{l,k}$. When $t = T - 2$, we have

$$\mathbb{E}\left[\exp\left(\frac{\gamma}{2}\sum_{t=T-2}^{T-1}\left(x_t^{l,k\top}\Delta K_t^{l\top}(R + B^\top \widetilde{P}_{t+1}B)\Delta K_t^l x_t^{l,k}\right)\right)\Big| H_{T-3}^{l,k}\right]$$

$$= \left(\det(I_n - \gamma D_{T-1}^l)\right)^{-\frac{1}{2}}\mathbb{E}\left[\exp\left(\frac{\gamma}{2}\left[x_{T-2}^{l,k\top}\left(\Delta K_{T-2}^{l\top}(R + B^\top \widetilde{P}_{T-1}B)\Delta K_{T-2}^l\right.\right.\right.\right.$$

$$\left.\left.\left.\left. + (A + BK_{T-2}^l)^\top \widetilde{D}_{T-1}^l(A + BK_{T-2}^l))x_{T-2}^{l,k}\right]\right)\Big| H_{T-3}^{l,k}\right]\right.$$

$$= \left(\det(I_n - \gamma D_{T-1}^l)\right)^{-\frac{1}{2}}\mathbb{E}\left[\exp\left(\frac{\gamma}{2}x_{T-2}^{l,k\top}D_{T-2}^l x_{T-2}^{l,k}\right)\Big| H_{T-3}^{l,k}\right]$$

$$= \prod_{t=T-2}^{T-1}\left(\det(I_n - \gamma D_t^l)\right)^{-\frac{1}{2}}\exp\left(\frac{\gamma}{2}\left[x_{T-3}^{l,k\top}(A + BK_{T-3}^l)^\top \widetilde{D}_{T-2}^l(A + BK_{T-3}^l)x_{T-3}^{l,k}\right]\right).$$

When $t = i, i = 1, \cdots, T - 1$, similarly, we have

$$\mathbb{E}\left[\exp\left(\frac{\gamma}{2}\sum_{t=i}^{T-1}\left(x_t^{l,k\top}\Delta K_t^{l\top}(R + B^\top \widetilde{P}_{t+1}B)\Delta K_t^l x_t^{l,k}\right)\right)\Big| H_{i-1}^{l,k}\right]$$

$$= \prod_{t=i}^{T-1}\left(\det(I_n - \gamma D_t^l)\right)^{-\frac{1}{2}}\exp\left(\frac{\gamma}{2}\left[x_{i-1}^{l,k\top}(A + BK_{i-1}^l)^\top \widetilde{D}_i^l(A + BK_{i-1}^l)x_{i-1}^{l,k}\right]\right).$$

Repeating this procedure, we get

$$J_0^{\pi^{l,k}}(x_0^{l,k}) - J_0^\star(x_0^{l,k})$$

$$= \frac{1}{\gamma}\log\mathbb{E}\left[\prod_{t=1}^{T-1}\left(\det(I_n - \gamma D_t^l)\right)^{-\frac{1}{2}}\right.$$

$$\left.\times \exp\left(\frac{\gamma}{2}\left[x_0^{l,k\top}\left((A + BK_0^l)^\top \widetilde{D}_1^l(A + BK_0^l) + \Delta K_0^{l\top}(R + B^\top \widetilde{P}_1 B)\Delta K_0^l\right)x_0^{l,k}\right]\right)\Big| x_0^{l,k}, H_T^{l,k-1}\right]$$

$$= \frac{1}{\gamma}\log\mathbb{E}\left[\prod_{t=1}^{T-1}\left(\det(I_n - \gamma D_t^l)\right)^{-\frac{1}{2}}\exp\left(\frac{\gamma}{2}x_0^{l,k\top}D_0^l x_0^{l,k}\right)\Big| x_0^{l,k}, H_T^{l,k-1}\right]$$

$$\stackrel{(2)}{=} \frac{1}{\gamma}\log\left(\prod_{t=1}^{T-1}\left(\det(I_n - \gamma D_t^l)\right)^{-\frac{1}{2}}\exp\left(\frac{\gamma}{2}x_0^{l,k\top}D_0^l x_0^{l,k}\right)\right)$$

$$= -\frac{1}{2\gamma}\sum_{t=1}^{T-1}\log\left(\det\left(I_n - \gamma D_t^l\right)\right) + \frac{1}{2}x_0^{l,k\top}D_0^l x_0^{l,k},$$

where inequality (2) holds because $D_t^l, t = 0, \cdots, T - 1$ is based on the data from epoch 1 to epoch $l - 1$. □

A.4 PROOF OF THEOREM 1

Now, we can derive the regret upper bound for Algorithm 1. Before we derive the high probability bounds for (27), we introduce some new notations and provide the bounds for $D_t^l$ in (26). Recall that

$$
\begin{aligned}
\psi_{T-1} &= 2\widetilde{\Gamma}^3, \\
\psi_t &= 2\widetilde{\Gamma}^3 (10\mathcal{V}^2 \mathcal{L}\widetilde{\Gamma}^4)^{2(T-t-1)} + 12\widetilde{\Gamma}^4 \psi_{t+1}, \ t = 0, \cdots, T-2,
\end{aligned}
\tag{32}
$$

where the definitions of $\mathcal{V}$ and $\mathcal{L}$ are given in (13). Assume that for any $t = 1, \cdots, T-1$, $l \in [L]$, we have

$$
\gamma \le \frac{1}{2\psi_t \mathcal{V}^2 \epsilon_l^2}.
\tag{33}
$$

We can choose a proper constant $\mathcal{C}_0$ for the initial epoch size $m_1$ in Theorem 1 so that $\gamma$ can satisfy assumptions in (33) when it satisfies the assumption of $I_n - \gamma P_{t+1} \succ 0$ and $I_n - \gamma P_{t+1}^l \succ 0$ in (5). Because $D_t^l$ are defined recursively, we obtain the bounds recursively from step $T-1$ to step 1. At step $T-1$,

$$
\begin{aligned}
\left\| D_{T-1}^l \right\| &= \left\| \Delta K_{T-1}^{l\top} \left( R + B^\top \widetilde{P}_T B \right) \Delta K_{T-1}^l \right\| \\
&\le \left\| R + B^\top \widetilde{P}_T B \right\| \cdot \left\| \Delta K_{T-1}^l \right\|^2 \\
&\overset{(1)}{\le} 2\widetilde{\Gamma}^3 \mathcal{V}^2 \epsilon_l^2 \\
&= \psi_{T-1} \mathcal{V}^2 \epsilon_l^2,
\end{aligned}
$$

where inequality (1) follows from the definition of $\widetilde{\Gamma}$ in (13) and Lemma 8. In terms of the bound for $\widetilde{D}_{T-1}^l$, we have

$$
\begin{aligned}
\left\| \widetilde{D}_{T-1}^l \right\| &= \left\| \left( I_n - \gamma D_{T-1}^l \right)^{-1} D_{T-1} \right\| \\
&\le \left\| \left( I_n - \gamma D_{T-1}^l \right)^{-1} \right\| \cdot \left\| D_{T-1}^l \right\| \\
&\overset{(2)}{\le} \frac{\| D_{T-1}^l \|}{1 - \gamma \| D_{T-1}^l \|} \\
&\overset{(3)}{\le} 2\| D_{T-1}^l \| \\
&= 2\psi_{T-1} \mathcal{V}^2 \epsilon_l^2,
\end{aligned}
$$

where inequality (2) holds by the fact that for any matrix $M \in R^{n \times n}$, if $\|M\| < 1$, then $\|(I_n - M)^{-1}\| \le \frac{1}{1-\|M\|}$ and inequality (3) follows from the assumption in (33). At step $T-2$, we have

$$
\begin{aligned}
\left\| D_{T-2}^l \right\| &= \left\| \Delta K_{T-2}^{l\top} \left( R + B^\top \widetilde{P}_{T-1} B \right) \Delta K_{T-2}^l + \left( A + BK_{T-2}^l \right)^\top \widetilde{D}_{T-1}^l \left( A + BK_{T-2}^l \right) \right\| \\
&\le 2\widetilde{\Gamma}^3 \left( 10\mathcal{V}^2 \mathcal{L}\widetilde{\Gamma}^4 \right)^2 \mathcal{V}^2 \epsilon_l^2 + \left\| A + B(\Delta K_{T-2}^l + K_{T-2}) \right\|^2 \cdot \left\| \widetilde{D}_{T-1}^l \right\| \\
&\overset{(4)}{\le} 2\widetilde{\Gamma}^3 \left( 10\mathcal{V}^2 \mathcal{L}\widetilde{\Gamma}^4 \right)^2 \mathcal{V}^2 \epsilon_l^2 + \left( 6\widetilde{\Gamma}^4 + 3\widetilde{\Gamma}^2 \left( 10\mathcal{V}^2 \mathcal{L}\widetilde{\Gamma}^4 \right)^2 \mathcal{V}^2 \epsilon_l^2 \right) \cdot 2\psi_{T-1} \mathcal{V}^2 \epsilon_l^2 \\
&= \left( 2\widetilde{\Gamma}^3 \left( 10\mathcal{V}^2 \mathcal{L}\widetilde{\Gamma}^4 \right)^2 + 12\widetilde{\Gamma}^4 \psi_{T-1} \right) \mathcal{V}^2 \epsilon_l^2 + o(\epsilon_l^2) \\
&= \psi_{T-2} \mathcal{V}^2 \epsilon_l^2 + o(\epsilon_l^2),
\end{aligned}
$$

where inequality (4) follows from the fact that $\left\| \sum_{t=1}^K x_t \right\|^2 \le K \sum_{t=1}^K \|x_t\|^2$. Similarly, we have

$$
\left\| \widetilde{D}_{T-2}^l \right\| \le 2\psi_{T-2} \mathcal{V}^2 \epsilon_l^2 + o(\epsilon_l^2).
$$

For $t = T - 2, \cdots, 1$, we can recursively derive that

$$\|D_t^l\| \le \psi_t \mathcal{V}^2 \epsilon_l^2 + o(\epsilon_l^2),$$
$$\left\| \widetilde{D}_t^l \right\| \le 2\psi_t \mathcal{V}^2 \epsilon_l^2 + o(\epsilon_l^2),$$
$$\|D_0^l\| \le \psi_0 \mathcal{V}^2 \epsilon_l^2 + o(\epsilon_l^2). \tag{34}$$

According to Lemma 5, the performance loss in the $k$-th episode of epoch $l$ is

$$J_0^{\pi^{l,k}}(x_0^{l,k}) - J_0^\star(x_0^{l,k})$$
$$= -\frac{1}{2\gamma} \sum_{t=1}^{T-1} \log \left( \det \left( I_n - \gamma D_t^l \right) \right) + \frac{1}{2} x_0^{l,k\top} D_0^l x_0^{l,k} \tag{35}$$
$$\overset{(5)}{\le} -\frac{1}{2\gamma} \sum_{t=1}^{T-1} \log \left( 1 - \gamma \|D_t^l\| \right)^n + \frac{1}{2} \|x_0^{l,k}\|^2 \|D_0^l\|.$$

Here, inequality (5) holds because $I_n - \gamma D_t^l \succeq \left( 1 - \gamma \|D_t^l\| \right) I_n \succeq (1 - \gamma(\psi_t \mathcal{V}^2 \epsilon_l^2 + o(\epsilon_l^2)))I_n \succ 0$ and $\det \left( \left( 1 - \gamma \|D_t^l\| \right) I_n \right) = \left( 1 - \gamma \|D_t^l\| \right)^n$. Substituting the inequalities in (34) into (35), we obtain

$$J_0^{\pi^{l,k}}(x_0^{l,k}) - J_0^\star(x_0^{l,k})$$
$$\overset{(6)}{\le} -\frac{n}{2\gamma} \sum_{t=1}^{T-1} \log \left( 1 - \gamma \left( \psi_t \mathcal{V}^2 \epsilon_l^2 + o\left( \epsilon_l^2 \right) \right) \right) + \frac{1}{2} \|x_0\|^2 \left( \psi_0 \mathcal{V}^2 \epsilon_l^2 + o\left( \epsilon_l^2 \right) \right)$$
$$\overset{(7)}{\le} \frac{n}{2} \left( \sum_{t=1}^{T-1} \psi_t \mathcal{V}^2 \epsilon_l^2 + o\left( \epsilon_l^2 \right) \right) + \frac{1}{2} \|x_0\|^2 \left( \psi_0 \mathcal{V}^2 \epsilon_l^2 + o\left( \epsilon_l^2 \right) \right)$$
$$= \frac{n}{2} \sum_{t=1}^{T-1} \psi_t \mathcal{V}^2 \epsilon_l^2 + \frac{1}{2} \|x_0\|^2 \psi_0 \mathcal{V}^2 \epsilon_l^2 + o(\epsilon_l^2),$$

inequality (6) holds by the inequalities in (34), and inequality (7) follows from the fact that $\log(1 + y) \le y$ for any $y > -1$.

Now, we can substitute the high probability bounds derived in Section A.1 into (35). Recall that conditional on event $\mathcal{G}^{l-1}$ in Lemma 3, with probability at least $1 - 2\delta_{l-1}$, we have

$$\left\| \theta^l - \theta \right\| \le \epsilon_l := \mathcal{C}_1 \left( \sqrt{\frac{\log \left( \frac{(m+n)^2}{\delta_{l-1}} \right)}{m_{l-1}}} + \frac{\log \left( \frac{(m+n)^2}{\delta_{l-1}} \right)}{m_{l-1}} + \frac{\log^2 \left( \frac{(m+n)^2}{\delta_{l-1}} \right)}{m_{l-1}^2} \right).$$

Similar to the procedure in page 26 in Basei et al. (2022), we set $\delta_{l-1} = \frac{\delta}{(l-1)^2}$, $m_{l-1} = 2^{l-2} m_1$, $m_1 = \mathcal{C}_0(-\log \delta)$, where $\delta \in (0, \frac{3}{\pi^2})$ and $\mathcal{C}_0$ is a finite positive constant that satisfies

$$\mathcal{C}_0 \ge \mathcal{C}_3 \sup_{l \in \mathbb{N}^+ \backslash \{1\}, \delta \in \left(0, \frac{2}{\pi^2}\right)} \left\{ \left\{ \frac{\log \left( \frac{(m+n)^2}{\delta_{l-1}} \right)}{2^{l-2}(-\log \delta)} \right\} \Big/ \min \left\{ \left( \frac{\rho}{3\mathcal{C}_1} \right)^2, 1 \right\} \right\},$$

where $\rho$ is defined at the beginning of Appendix A.1. Then, we have $m_{l-1} \ge \mathcal{C}_3 \log \left( \frac{(m+n)^2}{\delta_{l-1}} \right)$ and thus

$$\mathcal{C}_1 \left( \sqrt{\frac{\log \left( \frac{(m+n)^2}{\delta_{l-1}} \right)}{m_{l-1}}} + \frac{\log \left( \frac{(m+n)^2}{\delta_{l-1}} \right)}{m_{l-1}} + \frac{\log^2 \left( \frac{(m+n)^2}{\delta_{l-1}} \right)}{m_{l-1}^2} \right) \overset{(8)}{\le} 3\mathcal{C}_1 \sqrt{\frac{\log \left( \frac{(m+n)^2}{\delta_{l-1}} \right)}{m_{l-1}}} \le \rho, \forall l \in \mathbb{N}^+ \backslash \{1\},$$

where inequality (8) holds because $\mathcal{C}_3 \ge 1$ in Proposition 3. By a similar mathematical induction on page 27 in Basei et al. (2022), we can prove the following event

$$\mathcal{G} = \left\{ \|\theta^l - \theta\| \le 3\mathcal{C}_1 \sqrt{\frac{\log \left( \frac{(m+n)^2}{\delta_{l-1}} \right)}{m_{l-1}}}, \forall l \in \mathbb{N}^+ \backslash \{1\} \right\} \cup \{\theta^1 \in \Theta\} \tag{36}$$

holds with probability at least $1 - 2\sum_{l=2}^{\infty} \delta_{l-1} = 1 - \frac{\pi^2\delta}{3}$, i.e. $\mathbb{P}(\mathcal{G}) \geq 1 - \frac{\pi^2\delta}{3}$.

Under the event $\mathcal{G}$, which satisfies $\mathbb{P}(\mathcal{G}) \geq 1 - \frac{\pi^2\delta}{3}$, we can derive that

$\text{Regret}(N)$

$$
= \sum_{i=1}^{N} \left( J^{\pi^i}(x_0^i) - J^{\star}(x_0^i) \right)
$$

$$
= \sum_{l=1}^{L} \sum_{k=1}^{m_l} \left( J^{\pi^{l,k}}(x_0^{l,k}) - J^{\star}(x_0^{l,k}) \right)
$$

$$
\overset{(9)}{\leq} m_1 \left[ \frac{n}{2} \sum_{t=1}^{T-1} \psi_t \mathcal{V}^2 \epsilon_1^2 + \frac{1}{2} \|x_0\|^2 \psi_0 \mathcal{V}^2 \epsilon_1^2 + o(\epsilon_1^2) \right] + \sum_{l=2}^{L} m_l \left[ \frac{n}{2} \sum_{t=1}^{T-1} \psi_t \mathcal{V}^2 \epsilon_l^2 + \frac{1}{2} \|x_0\|^2 \psi_0 \mathcal{V}^2 \epsilon_l^2 + o(\epsilon_l^2) \right]
$$

$$
\overset{(10)}{\leq} m_1 \left[ \frac{n}{2} \sum_{t=1}^{T-1} \psi_t \mathcal{V}^2 \epsilon_1^2 + \frac{1}{2} \|x_0\|^2 \psi_0 \mathcal{V}^2 \epsilon_1^2 + o(\epsilon_1^2) \right] + \sum_{l=2}^{L} m_l \left[ \frac{n}{2} \sum_{t=1}^{T-1} \psi_t \mathcal{V}^2 \cdot 9\mathcal{C}_1^2 \cdot \frac{\log\left( \frac{(m+n)^2}{\delta_{l-1}} \right)}{m_{l-1}} \right.
$$

$$
\left. + \frac{1}{2} \|x_0\|^2 \psi_0 \mathcal{V}^2 \cdot 9\mathcal{C}_1^2 \cdot \frac{\log\left( \frac{(m+n)^2}{\delta_{l-1}} \right)}{m_{l-1}} + o\left( \frac{\log\left( \frac{(m+n)^2}{\delta_{l-1}} \right)}{m_{l-1}} \right) \right]
$$

$$
\overset{(11)}{\leq} \mathcal{C}_{\text{high}} + m_1 \left[ \frac{n}{2} \sum_{t=1}^{T-1} \psi_t \mathcal{V}^2 \epsilon_1^2 + \frac{1}{2} \|x_0\|^2 \psi_0 \mathcal{V}^2 \epsilon_1^2 \right]
$$

$$
+ \left[ 9\mathcal{C}_1^2 \mathcal{V}^2 \left( n \sum_{t=1}^{T-1} \psi_t + \|x_0\|^2 \psi_0 \right) \right] \cdot \sum_{l=2}^{L} \left( \log\left( \frac{m+n}{\sqrt{\delta}} \right) + \log(l-1) \right)
$$

$$
\overset{(12)}{\leq} \mathcal{C} \left( \sum_{t=0}^{T-1} \psi_t \right) \left[ \log\left( \frac{m+n}{\sqrt{\delta}} \right) L + L \log L \right],
$$

where inequality (9) follows from (35), inequality (10) follows from the definition of the event $\mathcal{G}$ in (36), $\psi_t$ is defined in (32), $\mathcal{C}_{\text{high}}$ in inequality (11) is a constant depends on $T, \gamma, m, n, \mathcal{V}, \widetilde{\Gamma}$ polynomially and it can bound the higher order term in inequality (10), and inequality (12) holds by Stirling's formula: $\sum_{l=2}^{L} \log(l-1) = \log((L-1)!) \leq \mathcal{C}'(L-1)\log(L-1)$, where $\mathcal{C}'$ is a positive constant. The expression of $\mathcal{C}$ is given by

$$
\mathcal{C} := \text{Polynomial}\left( \mathcal{C}_1, \mathcal{C}', \mathcal{V}, n, \epsilon_1, n, m_1, \|x_0\| \right),
$$

where $\epsilon_1$ is the estimation error in the first epoch, $m_1$ is the number of episodes in the first epoch, $\psi_t$ is defined in (32), $\mathcal{C}_1 = 16\mathcal{C}_2^3\mathcal{C}_4^2$ is from the proof of Proposition 3, and $\mathcal{V}$ is defined in (13).

## B  REGRET ANALYSIS OF THE LEAST-SQUARES-BASED ALGORITHM WITH EXPLORATION NOISE

In this section, we prove Theorem 2 discussed in Section 4. The proof structure of Theorem 2 is similar to the proof structure of Theorem 1. We present the high probability bounds for the estimation error of system matrices in Section B.1, the perturbation analysis of Riccati equations in Section B.2, and the simplification of the suboptimality gap resulting from controller mismatch in Section B.3.

### B.1  BOUNDS FOR THE ESTIMATION ERROR OF SYSTEM MATRICES

In this section, we derive the high probability bound for the estimation error of system matrices in Algorithm 2. Different from Section A.1, we adapt the classical self-normalized martingale analysis framework to derive the desired error bound.

Similar as in Section A.1, we fix the $k$-th episode and define the following compact set

$$
\Xi = \left\{ \hat{\theta} \in \mathbb{R}^{(n+m)\times n} \,\middle|\, \left\| \hat{\theta} - \theta \right\| \leq \varpi \right\} \cup \{\theta^1\},
$$

where $\varpi > 0$ is a constant that satisfies

$$\varpi \geq \max \left\{ \frac{2n}{\lambda} \left( \log \left( \frac{3n^2 N}{\delta^2} \right) + (n+m) \log \left( 1 + \frac{\tilde{c} N \log \left( \frac{3TN^2}{\delta} \right)}{\lambda} \right) \right) + 2(n+m)^2 \widetilde{\Gamma}^2, \right.$$

$$\left. \frac{80n}{cT} \left( \log \left( \frac{4n^2 N}{\delta^2} \right) + (n+m) \log \left( 1 + \frac{\tilde{c} N \log \left( \frac{4TN^2}{\delta} \right)}{\lambda} \right) \right) + \frac{80\lambda(n+m)^2 \widetilde{\Gamma}^2}{cT} \right\}. \tag{37}$$

Here, $\widetilde{\Gamma}$ is defined in (13), $\lambda$ is the regularization parameter and $c, \tilde{c} > 0$ are two constants independent of $k$ and $N$ but may depend on other constants including $n, m, \gamma$. The explicit expression of $c$ and $\tilde{c}$ can be found in (44) and (53). For any estimated $\tilde{\theta} \in \Xi$, there exists a universal constant $C_K > 0$ such that

$$\left\| \widetilde{K}_t \right\| \leq C_K, \quad \forall t, \tag{38}$$

where $\widetilde{K}_t$ is the control corresponding to $\tilde{\theta}$ and it's continuous in terms of $\tilde{\theta}$ according to (5). We also define the following event

$$\widetilde{\mathcal{G}}^k = \{ \theta^i \in \Xi, \forall i = 1, \cdots, k \}. \tag{39}$$

We will prove $\mathbb{P}(\widetilde{\mathcal{G}}^k) \geq 1 - \sum_{i=1}^{k-1} \frac{\delta}{N-1} = 1 - \frac{(k-1)\delta}{N-1}$ in Section B.4.

The main result of this section is the following proposition, which provides the high probability bound for the estimation error of system matrices estimated in Algorithm 2.

**Proposition 6.** *Let $\delta \in \left( 0, \frac{1}{4} \right)$. Conditional on event $\widetilde{\mathcal{G}}^k$, when $kT \geq 200 \left( 3(n+m) + \log \left( \frac{1}{\delta} \right) \right)$, with probability at least $1 - 4\delta$,*

$$\left\| \theta^{k+1} - \theta \right\|^2 \leq \frac{80n}{cT\sqrt{k}} \left( \log \left( \frac{n^2}{\delta^2} \right) + (n+m) \log \left( 1 + \frac{\tilde{c} k \log \left( \frac{TN}{\delta} \right)}{\lambda} \right) \right) + \frac{80\lambda(n+m)^2 \widetilde{\Gamma}^2}{cT\sqrt{k}},$$

*where $\widetilde{\Gamma}$ is defined in (13), the explicit expressions of $c$ and $\tilde{c}$ can be found in (44) and (53), $n$ is the dimension of the system state vector, $m$ is the dimension of the control vector and $\lambda$ is the regularization parameter. When $kT < 200 \left( 3(n+m) + \log \left( \frac{1}{\delta} \right) \right)$, with probability at least $1 - 3\delta$,*

$$\left\| \theta^{k+1} - \theta \right\|^2 \leq \frac{2n}{\lambda} \left( \log \left( \frac{n^2}{\delta^2} \right) + (n+m) \log \left( 1 + \frac{\tilde{c} k \log \left( \frac{TN}{\delta} \right)}{\lambda} \right) \right) + 2(n+m)^2 \widetilde{\Gamma}^2.$$

The proof of Proposition 6 is long, and we will discuss it in the following subsections.

### B.1.1 PRELIMINARIES

In this section, we recall an important high probability bound, known as self-normalized bound for vector-valued martingales. It will be used in the derivation of the bounds for the estimation error of system matrices.

**Lemma 12** (Theorem 1 in Abbasi-Yadkori et al. (2011)). *Let $\{\mathcal{F}_t\}_{t=0}^{\infty}$ be a filtration. Let $\{\eta_t\}_{t=0}^{\infty}$ be a real-valued stochastic process such that $\eta_t$ is $\mathcal{F}_{t+1}$-measurable and $\eta_t$ is conditionally $R$-sub-Gaussian for some $R \geq 0$ i.e.*

$$\mathbb{E} \left[ e^{\lambda \eta_t} \Big| \mathcal{F}_t \right] \leq \exp \left( \frac{\lambda^2 R^2}{2} \right), \forall \lambda \in \mathbb{R}.$$

*Let $\{X_t\}_{t=0}^{\infty}$ be an $\mathbb{R}^d$-valued stochastic process such that $X_t$ is $\mathcal{F}_t$-measurable. Assume that $V$ is a $d \times d$ positive definite matrix. For any $t \geq 0$, define*

$$\bar{V}_t = V + \sum_{s=0}^{t} X_s X_s^\top, \qquad S_t = \sum_{s=0}^{t} \eta_s X_s.$$

*Then, for any $\delta > 0$, with probability at least $1 - \delta$, for all $t \geq 0$,*

$$\|S_t\|_{\bar{V}_t^{-1}}^2 \leq 2R^2 \log \left( \frac{\det \left( \bar{V}_t \right)^{1/2} \det(V)^{-1/2}}{\delta} \right),$$

*where $\|S_t\|_{\bar{V}_t^{-1}}^2 = S_t^\top \left( \bar{V}_t \right)^{-1} S_t$.*

### B.1.2 SELF-NORMALIZED BOUNDS FOR THE ESTIMATION ERROR OF SYSTEM MATRICES

In this section, we analyze the estimation error based on bounds for the self-normalized martingale. Similar to Section A.3, let $\mathbb{H}_t^k$ be the set of possible histories up to step $t$ in the $k$-th episode. Denote the history up to step $t$ in the $k$-th episode by

$$\mathcal{H}_t^k = \left( x_0^1, u_0^1, \cdots, x_T^1, x_0^2, \cdots, x_0^k, \cdots, x_{t-1}^k, u_{t-1}^k, x_t^k, u_t^k \right). \tag{40}$$

The following lemma is a modified version of Theorem 2 in Abbasi-Yadkori et al. (2011) and Lemma 6 in Cohen et al. (2019), which provides a coarse self-normalized bound for the estimation error.

**Lemma 13.** *For any $\delta \in (0,1)$, with probability at least $1 - \delta$, we have*

$$\mathrm{Tr}\left( (\theta^{k+1} - \theta)^\top \bar{V}^k (\theta^{k+1} - \theta) \right) \le 2n \log\left( \frac{n^2}{\delta^2} \frac{\det(\bar{V}^k)}{\det(\lambda I)} \right) + 2\lambda \|\theta\|_F^2, \tag{41}$$

*where $\theta^{k+1}$ is the estimated system matrix defined in (15), $\theta$ is the true system matrix, $\lambda$ is the regularization parameter and*

$$\bar{V}^k = \lambda I + \sum_{i=1}^{k} \sum_{t=0}^{T-1} z_t^i z_t^{i\top}.$$

*Proof.* We first follow Lemma 6 in Cohen et al. (2019) to simplify $\theta^{k+1} - \theta$. Recall that

$$x_{t+1}^i = \theta^\top z_t^i + w_t^i, \quad w_t^i \sim \mathcal{N}(0, I_n)$$

where $z_t^i = \left[ x_t^{i\top} \; u_t^{i\top} \right]^\top$. Together with (15), we can obtain

$$\theta^{k+1} = \left( \bar{V}^k \right)^{-1} \left( \sum_{i=1}^{k} \sum_{t=0}^{T-1} z_t^i \left( z_t^{i\top} \theta + w_t^{i\top} \right) \right)$$

$$= \left( \bar{V}^k \right)^{-1} \left( \lambda \theta + \sum_{i=1}^{k} \sum_{t=0}^{T-1} z_t^i z_t^{i\top} \theta + \sum_{i=1}^{k} \sum_{t=0}^{T-1} z_t^i w_t^{i\top} - \lambda \theta \right)$$

$$= \theta + \left( \bar{V}^k \right)^{-1} \left( S_{T-1}^k - \lambda \theta \right),$$

where we denote $S_{T-1}^k = \sum_{i=1}^{k} \sum_{t=0}^{T-1} z_t^i w_t^{i\top}$ for the simplicity of notation. Then, we obtain

$$\mathrm{Tr}\left( (\theta^{k+1} - \theta)^\top \bar{V}^k (\theta^{k+1} - \theta) \right)$$

$$= \mathrm{Tr}\left( (S_{T-1}^k - \lambda\theta)^\top (\bar{V}^k)^{-1} (S_{T-1}^k - \lambda\theta) \right)$$

$$= \mathrm{Tr}\left( S_{T-1}^{k\top} (\bar{V}^k)^{-1} S_{T-1}^k + \lambda^2 \theta^\top (\bar{V}^k)^{-1} \theta - \lambda S_{T-1}^{k\top} (\bar{V}^k)^{-1} \theta - \lambda \theta^\top (\bar{V}^k)^{-1} S_{T-1}^k \right)$$

$$\overset{(1)}{\le} \mathrm{Tr}\left( S_{T-1}^{k\top} (\bar{V}^k)^{-1} S_{T-1}^k + \lambda^2 \theta^\top (\bar{V}^k)^{-1} \theta \right) + 2 \left\| \lambda \theta^\top (\bar{V}^k)^{-\frac{1}{2}} \right\|_F \cdot \left\| (\bar{V}^k)^{-\frac{1}{2}} S_{T-1}^k \right\|_F$$

$$\overset{(2)}{\le} 2\,\mathrm{Tr}\left( S_{T-1}^{k\top} (\bar{V}^k)^{-1} S_{T-1}^k \right) + 2\lambda^2 \mathrm{Tr}\left( \theta^\top (\bar{V}^k)^{-1} \theta \right)$$

$$\overset{(3)}{\le} 2\,\mathrm{Tr}\left( S_{T-1}^{k\top} (\bar{V}^k)^{-1} S_{T-1}^k \right) + 2\lambda \|\theta\|_F^2. \tag{42}$$

Here, we use Cauchy–Schwarz inequality $|\mathrm{Tr}(EF)| \le \|E\|_F \|F\|_F$ for any matrix $E$ and $F$ to obtain inequality (1), we use the inequality $2ab \le a^2 + b^2$ for any $a$ and $b$ to obtain inequality (2), and we use the fact that $\bar{V}^k \succeq \lambda I$ to obtain inequality (3).

We further bound $\mathrm{Tr}\left( S_{T-1}^{k\top} (\bar{V}^k)^{-1} S_{T-1}^k \right)$ in (42) to get the result in (41). Let $S_t^k(j) = \sum_{i=1}^{k} \sum_{s=0}^{t} z_s^i w_s^i(j), j = 1, \cdots, n, t = 0, \cdots, T-1, k = 1, \cdots, N$, where $w_s^i(j)$ is the $j$-th element of the random vector $w_s^i$. Recall the trajectory in (40), $z_s^i$ is $\mathcal{H}_s^i$-measurable for any step $s$

in the $i$-th episode and $w_s^i(j)$ is $\mathcal{H}_{s+1}^i$-measurable for any step $s$ in the $i$-th episode. Therefore, we can apply Lemma 12 and obtain that with probability at least $1 - \frac{\delta}{n}$,

$$S_{T-1}^k(j)^\top \left(\bar{V}^k\right)^{-1} S_{T-1}^k(j) \leq 2\log\left(\frac{n}{\delta}\frac{\det(\bar{V}^k)^{1/2}}{\det(\lambda I)^{1/2}}\right).$$

By a union bound, we can obtain that with probability at least $1 - \delta$,

$$\mathrm{Tr}\left(S_{T-1}^{k\top}\left(\bar{V}^k\right)^{-1} S_{T-1}^k\right) = \sum_{j=1}^n S_{T-1}^k(j)^\top \left(\bar{V}^k\right)^{-1} S_{T-1}^k(j) \leq n\log\left(\frac{n^2}{\delta^2}\frac{\det(\bar{V}^k)}{\det(\lambda I)}\right). \tag{43}$$

On combining (42) with (43), we can obtain (41). $\qquad\square$

After deriving the coarse self-normalized bounds in (41), we need to find the upper and lower bounds for $\bar{V}^k$ to obtain the result in Proposition 6. We follow the proof of Theorem 20 in Cohen et al. (2019) to derive the high probability lower bound for $\bar{V}^k$. The main difference is that we consider a decaying exploration noise while they consider a nondecaying exploration noise. The next lemma provides a lower bound for the conditional expectation of $z_t^k z_t^{k\top}, \forall k, t$, which is a modification of Lemma 34 in Cohen et al. (2019).

**Lemma 14.** *For all episode $k$ and step $t$, conditional on event $\widetilde{\mathcal{G}}^k$, we have*

$$\mathbb{E}\left[z_t^k z_t^{k\top} \big| \mathcal{H}_{t-1}^k\right] \succeq \frac{c}{\sqrt{k}} I_{m+n}, \quad t \neq 0,$$

*where $c > 0$ is a constant satisfying*

$$c \leq \frac{C_K^2 - C_K\sqrt{C_K^2 + 4} + 2}{2} \tag{44}$$

*with $C_K$ defined in (38) and $\frac{C_K^2 - C_K\sqrt{C_K^2+4}+2}{2} \in (0,1)$.*

*Proof.* Recall that $z_t^k = \left[x_t^{k\top}, u_t^{k\top}\right]^\top$, we have

$$\mathbb{E}\left[z_t^k z_t^{k\top} \big| \mathcal{H}_{t-1}^k\right] = \begin{bmatrix} I_n \\ K_t^k \end{bmatrix} \mathbb{E}\left[x_t^k x_t^{k\top} \big| \mathcal{H}_{t-1}^k\right] \begin{bmatrix} I_n & K_t^{k\top} \end{bmatrix} + \begin{bmatrix} 0 & 0 \\ 0 & \frac{1}{\sqrt{k}}I_m \end{bmatrix}$$

$$\overset{(1)}{\succeq} \begin{bmatrix} I_n \\ K_t^k \end{bmatrix} \begin{bmatrix} I_n & K_t^{k\top} \end{bmatrix} + \begin{bmatrix} 0 & 0 \\ 0 & \frac{1}{\sqrt{k}}I_m \end{bmatrix}$$

$$= \begin{bmatrix} \left(1 - \frac{c}{\sqrt{k}}\right)I_n & K_t^{k\top} \\ K_t^k & K_t^k K_t^{k\top} + \frac{1}{\sqrt{k}}(1-c)I_m \end{bmatrix} + \frac{c}{\sqrt{k}}I_{n+m}$$

$$\overset{(2)}{\succeq} \begin{bmatrix} \sqrt{1 - \frac{c}{\sqrt{k}}}I_n \\ \frac{K_t^k}{\sqrt{1-\frac{c}{\sqrt{k}}}} \end{bmatrix} \begin{bmatrix} \sqrt{1 - \frac{c}{\sqrt{k}}}I_n & \frac{K_t^{k\top}}{\sqrt{1-\frac{c}{\sqrt{k}}}} \end{bmatrix} + \frac{c}{\sqrt{k}}I_{n+m}$$

$$\succeq \frac{c}{\sqrt{k}}I_{n+m}.$$

Here, inequality (1) follows from the fact that $z_{t-1}^k = \left[x_{t-1}^{k\top}, u_{t-1}^{k\top}\right]^\top$ is $\mathcal{H}_{t-1}^k$-measurable and

$$\mathbb{E}\left[x_t^k x_t^{k\top} \big| \mathcal{H}_{t-1}^k\right] = \mathbb{E}\left[\left(Ax_{t-1}^k + Bu_{t-1}^k + w_{t-1}^k\right)\left(Ax_{t-1}^k + Bu_{t-1}^k + w_{t-1}^k\right)^\top \big| \mathcal{H}_{t-1}^k\right]$$

$$= \left(Ax_{t-1}^k + Bu_{t-1}^k\right)\left(Ax_{t-1}^k + Bu_{t-1}^k\right)^\top + \mathbb{E}\left[w_{t-1}^k w_{t-1}^{k\top} \big| \mathcal{H}_{t-1}^k\right]$$

$$\succeq I_n.$$

For inequality (2), when $0 < c \leq \frac{C_K^2 - C_K\sqrt{C_K^2+4}+2}{2}$, we can obtain

$$\frac{1}{1 - \frac{c}{\sqrt{k}}}K_t^k K_t^{k\top} \preceq K_t^k K_t^{k\top} + \frac{1}{\sqrt{k}}(1-c)I_m. \tag{45}$$

We can prove that (45) is equivalent to $\frac{c}{1-\frac{c}{\sqrt{k}}}K_t^k K_t^{k\top} \preceq \frac{c}{1-c}K_t^k K_t^{k\top} \preceq \frac{c}{1-c}C_K^2 I_m \preceq (1-c)I_m$.

Solving the inequality $\frac{c}{1-c}C_K^2 \leq 1 - c$, we can obtain $0 < c \leq \frac{C_K^2 - C_K\sqrt{C_K^2+4}+2}{2}$. $\qquad\square$

With the lower bound for the conditional expectation of $z_t^k z_t^{k\top}, \forall k, t$, we can derive the high probability lower bound as Lemma 33 in Cohen et al. (2019).

**Lemma 15.** *Let $\delta \in (0,1)$. Conditional on event $\widetilde{\mathcal{G}}^k$, when $kT \geq 200 \left(600(n+m) + \log\left(\frac{1}{\delta}\right)\right)$, with probability at least $1 - \delta$, we have*

$$\bar{V}^k \succeq \frac{cT\sqrt{k}}{40} I_{n+m}. \tag{46}$$

*Proof.* Let $e \in \mathbb{S}^{n+m-1}$, where $\mathbb{S}^{n+m-1} = \{v \in \mathbb{R}^{n+m} | \|v\|_2 = 1\}$. Let $I_t^k = e^\top z_t^k$ and let $\mathcal{Y}_t^k$ be an indicator random variable that equals 1 if $(I_t^k)^2 > \frac{c}{2\sqrt{k}}$ and 0 otherwise. By the similar arguments as in the proof of Lemma 35 in Cohen et al. (2019), we can prove that

$$\mathbb{E}\left[\mathcal{Y}_t^k | \mathcal{H}_{t-1}^k\right] = \mathbb{P}\left(\mathcal{Y}_t^k = 1 | \mathcal{H}_{t-1}^k\right) \geq \frac{1}{5}, \quad \text{if } t \neq 0. \tag{47}$$

Let $U_t^k = \mathcal{Y}_t^k - \mathbb{E}\left[\mathcal{Y}_t^k | \mathcal{H}_{t-1}^k\right]$. Then, $(U_t^k)$ is a martingale difference sequence with $|U_t^k| \leq 1, \forall k, t$. So we can use Azuma-Hoeffding inequality to derive the high probability bound: with probability at least $1 - \delta$,

$$\sum_{i=1}^{k} \sum_{t=1}^{T-1} U_t^i \geq -\sqrt{2kT \log\left(\frac{1}{\delta}\right)} \stackrel{(1)}{\geq} -\frac{kT}{10}, \tag{48}$$

where inequality (1) holds when $kT \geq 200 \log\left(\frac{1}{\delta}\right)$. On combining $U_t^i = \mathcal{Y}_t^i - \mathbb{E}\left[\mathcal{Y}_t^i | \mathcal{H}_{t-1}^i\right]$ with (48), we can obtain with probability at least $1 - \delta$,

$$\sum_{i=1}^{k} \sum_{t=1}^{T-1} \mathcal{Y}_t^i \geq \sum_{i=1}^{k} \sum_{t=1}^{T-1} \mathbb{E}\left[\mathcal{Y}_t^i | \mathcal{H}_{t-1}^i\right] - \frac{kT}{10} \stackrel{(2)}{\geq} \frac{kT}{5} - \frac{kT}{10} = \frac{kT}{10}, \tag{49}$$

where inequality (2) follows from (47). Denote $V^k = \sum_{i=1}^{k} \sum_{t=1}^{T-1} z_t^i z_t^{i\top}$. Then, we can get with probability at least $1 - \delta$,

$$e^\top V^k e = \sum_{i=1}^{k} \sum_{t=1}^{T-1} \left(I_t^i\right)^2 \stackrel{(3)}{\geq} \sum_{i=1}^{k} \sum_{t=1}^{T-1} \mathcal{Y}_t^i \frac{c}{2\sqrt{i}} \geq \sum_{i=1}^{k} \sum_{t=1}^{T-1} \mathcal{Y}_t^i \frac{c}{2\sqrt{k}} \stackrel{(4)}{\geq} \frac{kT}{10} \cdot \frac{c}{2\sqrt{k}} = \frac{\sqrt{k}Tc}{20},$$

where inequality (3) follows from the definition of $I_t^i$, inequality (4) holds by (49). Finally, by the similar $\frac{1}{4}$-net argument in the proof of Theorem 20 in Cohen et al. (2019), we can prove that when $kT \geq 200 \left(3(n+m) + \log\left(\frac{1}{\delta}\right)\right)$, with probability at least $1 - \delta$,

$$\left\|(V^k)^{-1}\right\| \leq \frac{40}{cT\sqrt{k}},$$

which is equivalent to

$$\bar{V}^k \succeq V^k \succeq \sum_{i=1}^{k} \sum_{t=1}^{T-1} z_t^i z_t^{i\top} \succeq \frac{cT\sqrt{k}}{40} I_{n+m}.$$

$\square$

In addition to the lower bound of $\bar{V}^k$, we also need to find the upper bound of $\bar{V}^k$ to get the final high probability bound for the estimation error of system matrices. In the following lemma, we provide the high probability upper bound for $\|x_t^k\|$, which plays a vital role in deriving the high probability upper bound of $\bar{V}^k$.

**Lemma 16.** *Let $\delta \in \left(0, \frac{1}{2}\right)$. Conditional on the event $\widetilde{\mathcal{G}}^k$, with probability at least $1 - 2\delta$, for all $0 \leq t \leq T$, we have*

$$\|x_t^k\| \leq 6 \left(\widetilde{\Gamma}(1 + C_K)\right)^t \left(n^{\frac{3}{4}} + m^{\frac{3}{4}}\right) \max\{\|x_0\|, 1\} \widetilde{\Gamma} \log^{\frac{1}{2}}\left(\frac{TN}{\delta}\right), \tag{50}$$

*where $\widetilde{\Gamma}$ is defined in (13) and $C_K$ is defined (38).*

*Proof.* Recall that $x_t^k = Ax_{t-1}^k + Bu_{t-1}^k + w_{t-1}^k = (A + BK_{t-1}^k)x_{t-1}^k + Bg_{t-1}^k + w_{t-1}^k$. Similar to (21), we can simplify $x_t^k$ to

$$x_t^k = \left( \prod_{j=t-1}^{0} \left( A + BK_j^k \right) \right) x_0^k + \sum_{r=0}^{t-1} \left( \prod_{j=t-1}^{r+1} \left( A + BK_j^k \right) \right) \left( Bg_r^k + w_r^k \right),$$

where $\prod_{j=t-1}^{r+1}(A + BK_j^k) = (A + BK_{t-1}^k)(A + BK_{t-2}^k)\cdots(A + BK_{r+1}^k)$, and $\prod_{i=t-1}^{t}(A + BK_i^k) = I_n$. Similar to Theorem 21 and Lemma 32 of Cohen et al. (2019), we can use Hanson-Wright inequality in Proposition 1.1 of Hsu et al. (2012) to derive that

$$\mathbb{P} \left( \|w_r^k\|^2 \leq 5n^{\frac{3}{2}} \log\left(\frac{TN}{\delta}\right), \|g_r^k\|^2 \leq 5m^{\frac{3}{2}} \frac{1}{\sqrt{k}} \log\left(\frac{TN}{\delta}\right), \forall k, r \right) \geq 1 - 2\delta \quad (51)$$

Then, we can bound the state vector by

$$\|x_t^k\| \leq \left\| \prod_{j=t-1}^{0} (A + BK_j^k) \right\| \cdot \|x_0^k\| + \sum_{r=0}^{T-1} \left\| \prod_{j=t-1}^{r+1} (A + BK_j^k) \right\| \cdot \|Bg_r^k + w_r^k\|$$

$$\leq \prod_{j=t-1}^{0} \left( \|A\| + \|B\| \cdot \|K_j^k\| \right) \|x_0^k\| + \sum_{r=0}^{t-1} \prod_{j=t-1}^{r+1} \left( \|A\| + \|B\| \cdot \|K_j^k\| \right) \left( \|B\| \cdot \|g_r^k\| + \|w_r^k\| \right)$$

$$\overset{(1)}{\leq} \widetilde{\Gamma}^t (1 + C_K)^t \|x_0\| + \sum_{r=0}^{t-1} \sqrt{5} \widetilde{\Gamma}^{t-r-1} (1 + C_K)^{t-r-1} \left( \frac{1}{k^{\frac{1}{4}}} m^{\frac{3}{4}} \widetilde{\Gamma} + n^{\frac{3}{4}} \right) \log^{\frac{1}{2}} \left( \frac{TN}{\delta} \right)$$

$$= \widetilde{\Gamma}^t (1 + C_K)^t \|x_0\| + \frac{\widetilde{\Gamma}^t (1 + C_K)^t - 1}{\widetilde{\Gamma}(1 + C_K) - 1} \cdot \sqrt{5} \left( \frac{1}{k^{\frac{1}{4}}} m^{\frac{3}{4}} \widetilde{\Gamma} + n^{\frac{3}{4}} \right) \log^{\frac{1}{2}} \left( \frac{TN}{\delta} \right)$$

$$\overset{(2)}{\leq} \left( \widetilde{\Gamma}(1 + C_K) \right)^t \|x_0\| + 5 \left( \widetilde{\Gamma}(1 + C_K) \right)^{t-1} \left( m^{\frac{3}{4}} + n^{\frac{3}{4}} \right) \widetilde{\Gamma} \log^{\frac{1}{2}} \left( \frac{TN}{\delta} \right)$$

$$\leq 6 \left( \widetilde{\Gamma}(1 + C_K) \right)^t \left( n^{\frac{3}{4}} + m^{\frac{3}{4}} \right) \max\{\|x_0\|, 1\} \widetilde{\Gamma} \log^{\frac{1}{2}} \left( \frac{TN}{\delta} \right),$$

where inequality (1) holds by the inequalities in (51) and inequality (2) follows from the fact that $\widetilde{\Gamma}(1 + C_K) - 1 \geq \widetilde{\Gamma}(1 + C_K) - \frac{1}{2}\widetilde{\Gamma}(1 + C_K) = \frac{1}{2}\widetilde{\Gamma}(1 + C_K)$ and $k \geq 1$. $\qquad \square$

With the result in Lemma 16, we can derive the high probability bound for $\|\bar{V}^k\|$.

**Lemma 17.** *Let $\delta \in (0,1)$. Conditional on event $\widetilde{\mathcal{G}}^k$, with probability at least $1 - 2\delta$, we have*

$$\|\bar{V}^k\| \leq \lambda + (1 + 2C_K^2) \left[ \frac{72 \left( (\widetilde{\Gamma}(1 + C_K))^{2T} - 1 \right)}{\left( \widetilde{\Gamma}(1 + C_K) \right)^2 - 1} \cdot (n^{\frac{3}{2}} + m^{\frac{3}{2}}) \cdot \max\left\{ \|x_0\|^2, 1 \right\} \widetilde{\Gamma}^2 k \log\left( \frac{TN}{\delta} \right) \right]$$

$$+ 20m^{\frac{3}{2}} \sqrt{k}T \log\left( \frac{TN}{\delta} \right).$$

*Proof.* Recall that $\bar{V}^k = \lambda I_{n+m} + \sum_{i=1}^{k} \sum_{t=0}^{T-1} z_t^i z_t^{i\top}, z_t^i = [x_t^{i\top}, u_t^{i\top}]^\top$. We have

$$\|\bar{V}^k\| \leq \lambda + \sum_{i=1}^{k} \sum_{t=0}^{T-1} \|z_t^i\|^2$$

$$= \lambda + \sum_{i=1}^{k} \sum_{t=0}^{T-1} \left( \|x_t^i\|^2 + \|K_t^i x_t^i + g_t^i\|^2 \right)$$

$$\overset{(1)}{\leq} \lambda + \sum_{i=1}^{k} \sum_{t=0}^{T-1} \left( \|x_t^i\|^2 + 2\|K_t^i\|^2 \cdot \|x_t^i\|^2 + 2\|g_t^i\|^2 \right)$$

$$\overset{(2)}{\leq} \lambda + \sum_{i=1}^{k} \sum_{t=0}^{T-1} \left( (1 + 2C_K^2)\|x_t^i\|^2 + 2\|g_t^i\|^2 \right),$$

(52)

where inequality (1) follows from the fact that $\|u + v\|^2 \le 2\|u\|^2 + 2\|v\|^2, \forall u, v$, inequality (2) holds by (38). Combine the results in Lemma 16 with (52), with probability at least $1 - 2\delta$, we can get

$$\|\bar{V}^k\| \overset{(3)}{\le} \lambda + \sum_{i=1}^{k} \sum_{t=0}^{T-1} \left[ (1 + 2C_K^2) \left( 72 \left( \widetilde{\Gamma}(1 + C_K) \right)^{2t} \left( n^{\frac{3}{2}} + m^{\frac{3}{2}} \right) \max\left\{ \|x_0\|^2, 1 \right\} \widetilde{\Gamma}^2 \log\left( \frac{TN}{\delta} \right) \right) \right.$$

$$\left. + \frac{10 m^{\frac{3}{2}}}{\sqrt{i}} \log\left( \frac{TN}{\delta} \right) \right]$$

$$= \lambda + (1 + 2C_K^2) \left[ \frac{72 \left( \left( \widetilde{\Gamma}(1 + C_K) \right)^{2T} - 1 \right)}{\left( \widetilde{\Gamma}(1 + C_K) \right)^2 - 1} \cdot (n^{\frac{3}{2}} + m^{\frac{3}{2}}) \cdot \max\left\{ \|x_0\|^2, 1 \right\} \widetilde{\Gamma}^2 k \log\left( \frac{TN}{\delta} \right) \right]$$

$$+ 20 m^{\frac{3}{2}} \sqrt{k} T \log\left( \frac{TN}{\delta} \right).$$

where inequality (3) follows from (50) and (51) in Lemma 16 and the fact that $\|u + v\|^2 \le 2\|u\|^2 + 2\|v\|^2, \forall u, v$. $\qquad\square$

For the simplicity of notation, we denote

$$\tilde{c} = (1 + 2C_K^2) \left[ \frac{72 \left( \left( \widetilde{\Gamma}(1 + C_K) \right)^{2T} - 1 \right)}{\left( \widetilde{\Gamma}(1 + C_K) \right)^2 - 1} \cdot (n^{\frac{3}{2}} + m^{\frac{3}{2}}) \cdot \max\left\{ \|x_0\|^2, 1 \right\} \widetilde{\Gamma}^2 \right] + 20 m^{\frac{3}{2}} T,$$

$$(53)$$

which is a constant independent of $k$ and $N$. Then, we can get

$$\|\bar{V}^k\| \le \lambda + \tilde{c} k \log\left( \frac{TN}{\delta} \right). \tag{54}$$

Now we are ready to prove Proposition 6.

*Proof of Proposition 6.* We can simplify (41) as follows:

$$\lambda_{\min}\left( \bar{V}^k \right) \left\| \theta^{k+1} - \theta \right\|_F^2 \le \mathrm{Tr}\left( \left( \theta^{k+1} - \theta \right)^\top \bar{V}^k \left( \theta^{k+1} - \theta \right) \right)$$

$$\le 2n \log\left( \frac{n^2}{\delta^2} \frac{\det(\bar{V}^k)}{\det(\lambda I)} \right) + 2\lambda \|\theta\|_F^2 \tag{55}$$

$$\overset{(1)}{\le} 2n \left( \log\left( \frac{n^2}{\delta^2} \right) + (n + m) \log\left( \frac{\|\bar{V}^k\|}{\lambda} \right) \right) + 2\lambda \|\theta\|_F^2,$$

where inequality (1) follows from the fact that $\det(M) \le \det(\lambda_{\max}(M) I_n) = \lambda_{\max}^n(M), \forall M \in \mathbb{R}^{n \times n}$. When $kT \ge 200 \left( 3(n + m) + \log\left( \frac{1}{\delta} \right) \right)$, substituting (46) and (54) into (55), with probability at least $1 - 4\delta$, we have

$$\left\| \theta^{k+1} - \theta \right\|^2 \le \left\| \theta^{k+1} - \theta \right\|_F^2$$

$$\le \frac{80n}{cT\sqrt{k}} \left( \log\left( \frac{n^2}{\delta^2} \right) + (n + m) \log\left( 1 + \frac{\lambda + \tilde{c} k \log\left( \frac{TN}{\delta} \right)}{\lambda} \right) \right) + \frac{80\lambda(n + m)^2 \widetilde{\Gamma}^2}{cT\sqrt{k}}.$$

When $kT < 200 \left( 3(n + m) + \log\left( \frac{1}{\delta} \right) \right)$, because $\bar{V}^k = \lambda I + \sum_{i=1}^{k} \sum_{t=0}^{T-1} z_t^i z_t^{i\top} \succeq \lambda I$, with probability at least $1 - 3\delta$,

$$\left\| \theta^{k+1} - \theta \right\|^2 \le \left\| \theta^{k+1} - \theta \right\|_F^2$$

$$\le \frac{2n}{\lambda} \left( \log\left( \frac{n^2}{\delta^2} \right) + (n + m) \log\left( 1 + \frac{\lambda + \tilde{c} k \log\left( \frac{TN}{\delta} \right)}{\lambda} \right) \right) + 2(n + m)^2 \widetilde{\Gamma}^2.$$

The proof is therefore complete. $\qquad\square$

## B.2  PERTURBATION ANALYSIS OF RICCATI EQUATION

The perturbation analysis of Riccati equation under Algorithm 1 and Algorithm 2 is the same. So we can get the similar bounds of Riccati perturbation by replacing $\epsilon_l$ with $\epsilon_k$ in Lemma 8, where $\epsilon_k = \max\{\|A^k - A\|, \|B^k - B\|\}$. The modified version of Lemma 8 is presented in the following lemma.

**Lemma 18.** *Assume $1 - \gamma\widetilde{\Gamma} > 0$ and fix any $\epsilon_k > 0$. Suppose $\|A^k - A\| \leq \epsilon_k, \|B^k - B\| \leq \epsilon_k$, then for any $t = 0, 1, \cdots, T - 1$, we have*

$$\|K_t^k - K_t\| \leq (10\mathcal{V}^2\mathcal{L}\widetilde{\Gamma}^4)^{T-t-1}\mathcal{V}\epsilon_k,$$
$$\|P_t^k - P_t\| \leq (10\mathcal{V}^2\mathcal{L}\widetilde{\Gamma}^4)^{T-t}\epsilon_k,$$

*where the definitions of $\mathcal{V}$, $\mathcal{L}$ and $\widetilde{\Gamma}$ can be found in (13).*

## B.3  SUBOPTIMALITY GAP DUE TO THE CONTROLLER MISMATCH

In this section, we will connect the gap between the total cost under policy $\pi^k$ and the total cost under the optimal policy with the estimation error and the perturbation of Riccati equation in Appendix B.1 and B.2. The proof framework is similar to the framework in Appendix A.3 except that we need to analyse the additional exploration noise added to the control. We define the total cost under entropic risk following policy $\pi^k$ (with slight abuse of notations) by

$$J_0^{\pi^k}\left(x_0^k\right) = \frac{1}{\gamma}\log\mathbb{E}\exp\left(\frac{\gamma}{2}\left(\sum_{t=0}^{T-1}\left(x_t^{k\top}Qx_t^k + u_t^{k\top}Ru_t^k\right) + x_T^{k\top}Q_Tx_T^k\right)\right),$$

where $u_t^k = K_t^k x_t^k + g_t^k, g_t^k \sim \mathcal{N}\left(0, \frac{1}{\sqrt{k}}I_m\right)$, $K_t^k$ is obtained by substituting $(A^k, B^k)$ into (5). Similar to Appendix A.3, we introduce the following new notations used in the regret analysis. For any $t = 0, 1, \cdots, T - 2$, we define the following recursive equations:

$$D_{T-1}^k = \Delta K_{T-1}^{k\top}(R + B^\top\widetilde{P}_TB)\Delta K_{T-1}^k,$$
$$E_{T-1}^k = \sigma_k\left(RK_{T-1}^k + B^\top\widetilde{P}_T(A + BK_{T-1}^k)\right),$$
$$F_{T-1}^k = \sigma_k^2\left(R + B^\top\widetilde{P}_TB\right),$$
$$U_{T-1}^k = D_{T-1}^k + \gamma E_{T-1}^{k\top}(I_m - \gamma F_{T-1}^k)^{-1}E_{T-1}^k,$$
$$\widetilde{U}_{T-1}^k = (I_n - \gamma U_{T-1}^k)^{-1}U_{T-1}^k,$$
$$D_t^k = \Delta K_t^{k\top}\left(R + B^\top\widetilde{P}_{t+1}B\right)\Delta K_t^k + (A + BK_t^k)^\top\widetilde{U}_{t+1}^k(A + BK_t^k),$$
$$E_t^k = \sigma_k\left(RK_t^k + B^\top(\widetilde{P}_{t+1} + \widetilde{U}_{t+1}^k)(A + BK_t^k)\right),$$
$$F_t^k = \sigma_k^2\left(R + B^\top(\widetilde{P}_{t+1} + \widetilde{U}_{t+1}^k)B\right),$$
$$U_t^k = D_t^k + \gamma E_t^{k\top}(I_m - \gamma F_t^k)^{-1}E_t^k,$$
$$\widetilde{U}_t^k = (I_n - \gamma U_t^k)^{-1}U_t^k,$$

(56)

where $\Delta K_t^k := K_t^k - K_t, \sigma_k := k^{-\frac{1}{4}}$ and $\widetilde{P}_T$ is defined in (5).

We then follow the proof framework of Appendix A.3 to derive the bounds for the suboptimality gap due to the controller mismatch. The key result of this section is the following proposition.

**Proposition 7.** *We have*

$$J_0^{\pi^k}(x_0^k) - J_0^\star(x_0^k)$$
$$= -\frac{1}{2\gamma}\sum_{t=0}^{T-1}\log\det\left(I_n - \gamma F_t^k\right) - \frac{1}{2\gamma}\sum_{t=1}^{T-1}\log\det\left(I_m - \gamma U_t^k\right) + \frac{1}{2}x_0^{k\top}U_0^kx_0^k,$$

(57)

*where $F_t^k$ and $U_t^k$ are defined in (56).*

In order to prove Proposition 7, we extend Lemma 10 and prove the following result. Recall that $\sigma_k = k^{-\frac{1}{4}}$.

**Lemma 19.** *For any $t \in [T-1]$, we have*

$$
E\left[\exp\left(\frac{\gamma}{2}\begin{bmatrix} x_t^k \\ \sigma_k^{-1}g_t^k \end{bmatrix}^\top \begin{bmatrix} D_t^k & E_t^{k\top} \\ E_t^k & F_t^k \end{bmatrix} \begin{bmatrix} x_t^k \\ \sigma_k^{-1}g_t^k \end{bmatrix}\right)\bigg|\mathcal{H}_{t-1}^k\right]
$$

$$
= \left(\det(I_n - \gamma U_t^k)\right)^{-1/2} \cdot \left(\det(I_m - \gamma F_t^k)\right)^{-1/2}
$$

$$
\times \exp\left\{\frac{\gamma}{2}\left[x_{t-1}^{k\top}(A + BK_{t-1}^k)^\top \widetilde{U}_t^k(A + BK_{t-1}^k)x_{t-1}^k \right.\right.
$$

$$
\left.\left. + 2g_{t-1}^{k\top}B^\top \widetilde{U}_t^k(A + BK_{t-1}^k)x_{t-1}^k + g_{t-1}^{k\top}B^\top \widetilde{U}_t^k Bg_{t-1}^k\right]\right\}, \tag{58}
$$

*where given $(x_{t-1}^k, u_{t-1}^k)$,*

$$
\begin{bmatrix} x_t^k \\ \sigma_k^{-1}g_t^k \end{bmatrix} \sim \mathcal{N}\left(\begin{bmatrix} Ax_{t-1}^k + Bu_{t-1}^k \\ 0 \end{bmatrix}, I_{n+m}\right).
$$

*Proof.* We obtain from Lemma 10 that

$$
\mathbb{E}\left[\exp\left(\frac{\gamma}{2}\begin{bmatrix} x_t^k \\ \sigma_k^{-1}g_t^k \end{bmatrix}^\top \begin{bmatrix} D_t^k & E_t^{k\top} \\ E_t^k & F_t^k \end{bmatrix} \begin{bmatrix} x_t^k \\ \sigma_k^{-1}g_t^k \end{bmatrix}\right)\bigg|\mathcal{H}_{t-1}^k\right]
$$

$$
= \left(\det\left(I_{m+n} - \gamma\begin{bmatrix} D_t^k & E_t^{k\top} \\ E_t^k & F_t^k \end{bmatrix}\right)\right)^{-\frac{1}{2}}
$$

$$
\times \exp\left(\frac{\gamma}{2}\begin{bmatrix} Ax_{t-1}^k + Bu_{t-1}^k \\ 0 \end{bmatrix}^\top \left(I_{m+n} - \gamma\begin{bmatrix} D_t^k & E_t^{k\top} \\ E_t^k & F_t^k \end{bmatrix}\right)^{-1}\right. \tag{59}
$$

$$
\left. \cdot \begin{bmatrix} D_t^k & E_t^{k\top} \\ E_t^k & F_t^k \end{bmatrix}\begin{bmatrix} Ax_{t-1}^k + Bu_{t-1}^k \\ 0 \end{bmatrix}\right).
$$

We perform a block Gauss–Jordan elimination, take the inverse of the matrix in the determinant of (59), and obtain

$$
\left(I_{n+m} - \gamma\begin{bmatrix} D_t^k & E_t^{k\top} \\ E_t^k & F_t^k \end{bmatrix}\right)^{-1}
$$

$$
= \begin{bmatrix} I_n - \gamma D_t^k & -\gamma E_t^{k\top} \\ -\gamma E_t^k & I_m - \gamma F_t^k \end{bmatrix}^{-1}
$$

$$
= \begin{bmatrix} I_n & 0 \\ (I_m - \gamma F_t^k)^{-1}\gamma E_t^k & I_m \end{bmatrix} \times \begin{bmatrix} (I_n - \gamma D_t^k - \gamma^2 E_t^{k\top}(I_m - \gamma F_t^k)^{-1}E_t^k)^{-1} & 0 \\ 0 & (I_m - \gamma F_t^k)^{-1} \end{bmatrix}
$$

$$
\times \begin{bmatrix} I_n & \gamma E_t^{k\top}(I_m - \gamma F_t^k)^{-1} \\ 0 & I_m \end{bmatrix}
$$

$$
= \begin{bmatrix} EL_1 & EL_2 \\ EL_3 & EL_4 \end{bmatrix},
$$

where

$$
EL_1 = \left(I_n - \gamma D_t^k - \gamma^2 E_t^{k\top}(I_m - \gamma F_t^k)^{-1}E_t^k\right)^{-1},
$$

$$
EL_2 = (I_n - \gamma D_t^k - \gamma^2 E_t^{k\top}(I_m - \gamma F_t^k)^{-1}E_t^k)^{-1}\gamma E_t^{k\top}(I_m - \gamma F_t^k)^{-1}),
$$

$$
EL_3 = (I_m - \gamma F_t^k)^{-1}\gamma E_t^k(I_n - \gamma D_t^k - \gamma^2 E_t^{k\top}(I_m - \gamma F_t^k)^{-1}E_t^k)^{-1},
$$

$$
EL_4 = (I_m - \gamma F_t^k)^{-1}\gamma E_t^k(I_n - \gamma D_t^k - \gamma^2 E_t^{k\top}(I_m - \gamma F_t^k)^{-1}E_t^k)^{-1}\gamma E_t^{k\top}(I_m - \gamma F_t^k)^{-1}
$$

$$
+ (I_m - \gamma F_t^k)^{-1}.
$$

Then, we can obtain

$$
\left( \det \left( I_{n+m} - \gamma \begin{bmatrix} D_t^k & E_t^{k\top} \\ E_t^k & F_t^k \end{bmatrix} \right) \right)^{-\frac{1}{2}}
$$
$$
= \left( \det \left( I_n - \gamma D_t^k - \gamma^2 E_t^{k\top}(I_m - \gamma F_t^k)^{-1} E_t^k \right) \right)^{-\frac{1}{2}} \cdot \left( \det(I_m - \gamma F_t^k) \right)^{-\frac{1}{2}} \tag{60}
$$
$$
= \left( \det \left( I_n - \gamma U_t^k \right) \right)^{-\frac{1}{2}} \cdot \left( \det(I_m - \gamma F_t^k) \right)^{-\frac{1}{2}},
$$

where $U_t^k = D_t^k + \gamma E_t^{k\top}(I_m - \gamma F_t^k)^{-1} E_t^k$, and

$$
\begin{bmatrix} Ax_{t-1}^k + Bu_{t-1}^k \\ 0 \end{bmatrix}^\top \left( I_{n+m} - \gamma \begin{bmatrix} D_t^k & E_t^{k\top} \\ E_t^k & F_t^k \end{bmatrix} \right)^{-1} \begin{bmatrix} D_t^k & E_t^{k\top} \\ E_t^k & F_t^k \end{bmatrix} \begin{bmatrix} Ax_{t-1}^k + Bu_{t-1}^k \\ 0 \end{bmatrix}
$$
$$
= (Ax_{t-1}^k + Bu_{t-1}^k)^\top \left[ \left( I_n - \gamma D_t^k - \gamma E_t^{k\top}(I_m - \gamma F_t^k)^{-1} E_t^k \right)^{-1} \right.
$$
$$
\left. \times \left( D_t^k + \gamma E_t^{k\top}(I_m - \gamma F_t^k)^{-1} E_t^k \right) \right] (Ax_{t-1}^k + Bu_{t-1}^k)
$$
$$
= \left( (A + BK_{t-1}^k)x_{t-1}^k + Bg_{t-1}^k \right)^\top \widetilde{U}_t^k \left( (A + BK_{t-1}^k)x_{t-1}^k + Bg_{t-1}^k \right)
$$
$$
= x_{t-1}^{k\top}(A + BK_{t-1}^k)^\top \widetilde{U}_t^k(A + BK_{t-1}^k)x_{t-1}^k + 2g_{t-1}^{k\top}B^\top \widetilde{U}_t^k(A + BK_{t-1}^k)x_{t-1}^k + g_{t-1}^{k\top}B^\top \widetilde{U}_t^k Bg_{t-1}^k, \tag{61}
$$

where $\widetilde{U}_t^k = (I_n - \gamma U_t^k)^{-1} U_t^k$. On combining (60) and (61) with (59), we can obtain (58). □

The following lemma is an extension of Lemma 11, which provides a coarse simplification of the performance gap in one episode.

**Lemma 20.** *With Lemma 10, we can simplify the performance gap to*

$$
J_0^{\pi^k}(x_0^k) - J_0^\star(x_0^k)
$$
$$
= \frac{1}{\gamma} \log \mathbb{E} \left[ \exp \left( \frac{\gamma}{2} \sum_{t=0}^{T-1} \left( x_t^{k\top} \Delta K_t^{k\top}(R + B^\top \widetilde{P}_{t+1} B) \Delta K_t^k x_t^k \right. \right. \right.
$$
$$
+ 2g_t^{k\top}(RK_t^k + B^\top \widetilde{P}_{t+1}(A + BK_t^k))x_t^k
$$
$$
\left. \left. \left. + g_t^{k\top}(R + B^\top \widetilde{P}_{t+1} B)g_t^k \right) \right) \middle| x_0^k, \mathcal{H}_T^{k-1} \right],
$$

*where $\Delta K_t^k = K_t^k - K_t$ and $\mathcal{H}_T^{k-1}$ is defined in (40).*

*Proof.* By a similar procedure as in (29), we can derive that

$$
J_0^{\pi^k}(x_0^k) - J_0^\star(x_0^k)
$$
$$
= \frac{1}{\gamma} \log \mathbb{E} \left[ \exp \left( \frac{\gamma}{2} \sum_{t=0}^{T-1} \left( c_t(x_t^k, u_t^k) + J_{t+1}(x_{t+1}^k) - J_t(x_t^k) \right) \right) \middle| x_0^k, \mathcal{H}_T^{k-1} \right]
$$
$$
\stackrel{(1)}{=} \frac{1}{\gamma} \log \mathbb{E} \left[ \exp \left( \gamma \sum_{t=0}^{T-1} \left( \frac{1}{2} x_t^{k\top}(Q + K_t^{k\top} RK_t^k)x_t^k + g_t^{k\top} RK_t^k x_t^k + \frac{1}{2} g_t^{k\top} Rg_t^k \right. \right. \right.
$$
$$
\left. \left. \left. + \frac{1}{2} x_{t+1}^{k\top} P_{t+1} x_{t+1}^k - \frac{1}{2} x_t^{k\top} P_t x_t^k + \frac{1}{2\gamma} \log \det(I_n - \gamma P_{t+1}) \right) \right) \middle| x_0^k, \mathcal{H}_T^{k-1} \right], \tag{62}
$$

where equality (1) follows from the definition of the total cost under entropic risk and $u_t^k = K_t^k x_t^k + g_t^k$. Again, we apply the law of total expectation, and compute

$$
\mathbb{E}\left[\exp\left(\gamma\Big(\frac{1}{2}x_t^{k\top}(Q + K_t^{k\top}RK_t^k)x_t^k + g_t^{k\top}RK_t^k x_t^k + \frac{1}{2}g_t^{k\top}Rg_t^k\right.\right.
$$

$$
\left.\left. + \frac{1}{2}x_{t+1}^{k\top}P_{t+1}x_{t+1}^k - \frac{1}{2}x_t^{k\top}P_t x_t^k + \frac{1}{2\gamma}\log\det(I_n - \gamma P_{t+1})\Big)\right)\bigg|\mathcal{H}_t^k\right]
$$

$$
= \exp\left(\gamma\Big(\frac{1}{2}x_t^{k\top}(Q + K_t^{k\top}RK_t^k)x_t^k - \frac{1}{2}x_t^{k\top}P_t x_t^k + g_t^{k\top}RK_t^k x_t^k + \frac{1}{2}g_t^{k\top}Rg_t^k\right.
$$

$$
\left. + \frac{1}{2\gamma}\log\det(I_n - \gamma P_{t+1})\Big)\right) \times \mathbb{E}\left[\exp\left(\frac{\gamma}{2}x_{t+1}^{k\top}P_{t+1}x_{t+1}^k\right)\bigg|\mathcal{H}_t^k\right]
$$

(63)

It follows from Lemma 10 that

$$
(63) = \exp\left(\gamma\Big(\frac{1}{2}x_t^{k\top}(Q + K_t^{k\top}RK_t^k)x_t^k - \frac{1}{2}x_t^{k\top}P_t x_t^k + g_t^{k\top}RK_t^k x_t^k + \frac{1}{2}g_t^{k\top}Rg_t^k\right.
$$

$$
\left. + \frac{1}{2\gamma}\log\det(I_n - \gamma P_{t+1})\Big)\right) \times (\det(I_n - \gamma P_{t+1}))^{-1/2}
$$

$$
\times \exp\left[\frac{\gamma}{2}x_t^{k\top}(A + BK_t^k)^\top \widetilde{P}_{t+1}(A + BK_t^k)x_t^k + \gamma g_t^{k\top}B^\top\widetilde{P}_{t+1}(A + BK_t^k)x_t^k\right.
$$

$$
\left. + \frac{\gamma}{2}g_t^{k\top}B^\top\widetilde{P}_{t+1}Bg_t^k\right]
$$

$$
= \exp\left[\frac{\gamma}{2}\Big(x_t^{k\top}(Q + K_t^{k\top}RK_t^k + (A + BK_t^k)^\top\widetilde{P}_{t+1}(A + BK_t^k))x_t^k - x_t^{k\top}P_t x_t^k\Big)\right.
$$

$$
\left. + \gamma g_t^{k\top}(RK_t^k + B^\top\widetilde{P}_{t+1}(A + BK_t^k))x_t^k + \frac{\gamma}{2}g_t^{k\top}(R + B^\top\widetilde{P}_{t+1}B)g_t^k\right],
$$

Substituting $\Delta K_t^k = K_t^k - K_t$ and the Riccati equation $P_t = Q + K_t^\top RK_t + (A + BK_t)^\top\widetilde{P}_{t+1}(A + BK_t)$ into (63), we obtain

$$
\exp\left[\frac{\gamma}{2}x_t^{k\top}\Big(Q + (\Delta K_t^k + K_t)^\top R(\Delta K_t^k + K_t) + (A + B(\Delta K_t^k + K_t))^\top\widetilde{P}_{t+1}(A + B(\Delta K_t^k + K_t))\Big)x_t^k\right.
$$

$$
\left. + \gamma g_t^{k\top}(RK_t^k + B^\top\widetilde{P}_{t+1}(A + BK_t^k))x_t^k + \frac{\gamma}{2}g_t^{k\top}(R + B^\top\widetilde{P}_{t+1}B)g_t^k - \frac{\gamma}{2}x_t^{k\top}P_t x_t^k\right]
$$

$$
= \exp\left[\frac{\gamma}{2}x_t^{k\top}\Delta K_t^{k\top}(R + B^\top\widetilde{P}_{t+1}B)\Delta K_t^k x_t^k + \gamma x_t^{k\top}\Delta K_t^{k\top}\big((R + B^\top\widetilde{P}_{t+1}B)K_t + B^\top\widetilde{P}_{t+1}A\big)x_t^k\right.
$$

$$
\left. + \gamma g_t^{k\top}(RK_t^k + B^\top\widetilde{P}_{t+1}(A + BK_t^k))x_t^k + \frac{\gamma}{2}\,g_t^{k\top}(R + B^\top\widetilde{P}_{t+1}B)g_t^k\right]
$$

$$
\overset{(2)}{=} \exp\left[\frac{\gamma}{2}x_t^{k\top}\Delta K_t^{k\top}(R + B^\top\widetilde{P}_{t+1}B)\Delta K_t^k x_t^k + \gamma g_t^{k\top}(RK_t^k + B^\top\widetilde{P}_{t+1}(A + BK_t^k))x_t^k\right.
$$

$$
\left. + \frac{\gamma}{2}g_t^{k\top}(R + B^\top\widetilde{P}_{t+1}B)g_t^k\right],
$$

(64)

where the equality (2) holds by the fact that $K_t = -(R + B^\top\widetilde{P}_{t+1}B)^{-1}B^\top\widetilde{P}_{t+1}A$. Substituting (64) into (63) and then substituting (63) into (62), we can get

$$
J_0^{\pi^k}(x_0^k) - J_0^\star(x_0^k)
$$

$$
= \frac{1}{\gamma}\log\mathbb{E}\left[\exp\left(\frac{\gamma}{2}\sum_{t=0}^{T-1}\Big(x_t^{k\top}\Delta K_t^{k\top}(R + B^\top\widetilde{P}_{t+1}B)\Delta K_t^k x_t^k\right.\right.
$$

$$
\left.\left. + 2g_t^{k\top}(RK_t^k + B^\top\widetilde{P}_{t+1}(A + BK_t^k))x_t^k + g_t^{k\top}(R + B^\top\widetilde{P}_{t+1}B)g_t^k\Big)\right)\bigg|x_0^k, \mathcal{H}_T^{k-1}\right].
$$

The proof is complete. □

Combining Lemma 19 with Lemma 20, we can prove Proposition 7.

*Proof of Proposition 7.* We prove the result recursively. Recall that $\sigma_k = k^{-\frac{1}{4}}$. When $t = T - 1$,

$$
\mathbb{E}\left[ \exp\left( \frac{\gamma}{2}\left( x_{T-1}^{k\top}\Delta K_{T-1}^{k\top}(R + B^\top \widetilde{P}_T B)\Delta K_{T-1}^k x_{T-1}^k \right.\right.\right.
$$

$$
\left.\left.\left. + 2\sigma_k\sigma_k^{-1}g_{T-1}^{k\top}(RK_{T-1}^k + B^\top \widetilde{P}_T(A + BK_{T-1}^k))x_{T-1}^k + \sigma_k^2\sigma_k^{-2}g_{T-1}^{k\top}(R + B^\top \widetilde{P}_T B)g_{T-1}^k \right)\right)\right|\mathcal{H}_{T-2}^k\right]
$$

$$
\overset{(1)}{=} \mathbb{E}\left[ \exp\left( \frac{\gamma}{2}\left( x_{T-1}^{k\top}D_{T-1}^k x_{T-1}^k + 2\sigma_k^{-1}g_{T-1}^{k\top}E_{T-1}^k x_{T-1}^k + \sigma_k^{-2}g_{T-1}^{k\top}F_{T-1}^k g_{T-1}^k \right)\right)\Big|\mathcal{H}_{T-2}^k\right]
$$

$$
\overset{(2)}{=} \left(\det(I_n - \gamma U_{T-1}^k)\right)^{-\frac{1}{2}}\left(\det(I_m - \gamma F_{T-1}^k)\right)^{-\frac{1}{2}}
$$

$$
\times \exp\left\{ \frac{\gamma}{2}\left[ x_{T-2}^{k\top}(A + BK_{T-2}^k)^\top \widetilde{U}_{T-1}^k(A + BK_{T-2}^k)x_{T-2}^k \right.\right.
$$

$$
\left.\left. + 2g_{T-2}^{k\top}B^\top \widetilde{U}_{T-1}^k(A + BK_{T-2}^k)x_{T-2}^k + g_{T-2}^{k\top}B^\top \widetilde{U}_{T-1}^k B g_{T-2}^k\right],\right.
$$

$$
\tag{65}
$$

where equality (1) holds by (56), and equality (2) follows from Lemma 19. When $t = T - 2$, we have

$$
\mathbb{E}\left[ \exp\left( \frac{\gamma}{2}\sum_{t=T-2}^{T-1}\left( x_t^{k\top}\Delta K_t^{k\top}(R + B^\top \widetilde{P}_{t+1}B)\Delta K_t^k x_t^k + 2\sigma_k\sigma_k^{-1}g_t^{k\top}(RK_t^k + B^\top \widetilde{P}_{t+1}(A + BK_t^k))x_t^k \right.\right.\right.
$$

$$
\left.\left.\left. + \sigma_k^2\sigma_k^{-2}g_t^{k\top}(R + B^\top \widetilde{P}_{t+1}B)g_t^k \right)\right)\Big|\mathcal{H}_{T-3}^k\right]
$$

$$
\overset{(3)}{=} \left(\det(I_n - \gamma U_{T-1}^k)\right)^{-\frac{1}{2}}\left(\det(I_m - \gamma F_{T-1}^k)\right)^{-\frac{1}{2}}
$$

$$
\times \mathbb{E}\left[ \exp\left( \frac{\gamma}{2}\left[ x_{T-2}^{k\top}\left(\Delta K_{T-2}^{k\top}(R + B^\top \widetilde{P}_{T-1}B)\Delta K_{T-2}^k + (A + BK_{T-2}^k)^\top \widetilde{U}_{T-1}^k(A + BK_{T-2}^k)\right)x_{T-2}^k \right.\right.\right.
$$

$$
\left.\left.\left. + 2\sigma_k\sigma_k^{-1}g_{T-2}^{k\top}(RK_{T-2}^k + B^\top(\widetilde{P}_{T-1} + \widetilde{U}_{T-1}^k)(A + BK_{T-2}^l))x_{T-2}^k \right.\right.\right.
$$

$$
\left.\left.\left. + \sigma_k^2\sigma_k^{-2}g_{T-2}^{k\top}(R + B^\top(\widetilde{P}_{T-1} + \widetilde{U}_{T-1}^k)B)g_{T-2}^k\right]\right)\Big|\mathcal{H}_{T-3}^k\right]
$$

$$
\overset{(4)}{=} \left(\det(I_n - \gamma U_{T-1}^k)\right)^{-\frac{1}{2}}\left(\det(I_m - \gamma F_{T-1}^k)\right)^{-\frac{1}{2}}
$$

$$
\times \mathbb{E}\left[ \exp\left( \frac{\gamma}{2}\left( x_{T-2}^{k\top}D_{T-2}^k x_{T-2}^k + 2\sigma_k^{-1}g_{T-2}^{k\top}E_{T-2}^k x_{T-2}^k + \sigma_k^{-2}g_{T-2}^{k\top}F_{T-2}^k g_{T-2}^k \right)\right)\Big|\mathcal{H}_{T-3}^k\right]
$$

$$
\overset{(5)}{=} \prod_{t=T-2}^{T-1}\left(\det(I_n - \gamma U_t^k)\right)^{-\frac{1}{2}}\left(\det(I_m - \gamma F_t^k)\right)^{-\frac{1}{2}}
$$

$$
\times \exp\left( \frac{\gamma}{2}\left[ x_{T-3}^{k\top}(A + BK_{T-3}^k)^\top \widetilde{U}_{T-2}^k(A + BK_{T-3}^k)x_{T-3}^k \right.\right.
$$

$$
\left.\left. + 2g_{T-3}^{k\top}B^\top \widetilde{U}_{T-2}^k(A + BK_{T-3}^k)x_{T-3}^k + g_{T-3}^{k\top}B^\top \widetilde{U}_{T-2}^k B g_{T-3}^k\right]\right),
$$

where equality (3) holds by applying the law of total expectation and applying (65), equality (4) follows from (56) and equality (5) still holds by applying Lemma 19. When $t = i$, we can similarly

derive

$$
\mathbb{E}\left[\exp\left(\frac{\gamma}{2}\sum_{t=i}^{T-1}\left(x_t^{k\top}\Delta K_t^{k\top}(R+B^\top\widetilde{P}_{t+1}B)\Delta K_t^k x_t^k\right.\right.\right.
$$

$$
\left.\left.\left.+2\sigma_k\sigma_k^{-1}g_t^{k\top}(RK_t^k+B^\top\widetilde{P}_{t+1}(A+BK_t^k))x_t^k+\sigma_k^2\sigma_k^{-2}g_t^{k\top}(R+B^\top\widetilde{P}_{t+1}B)g_t^k\right)\right)\bigg|\mathcal{H}_{i-1}^k\right]
$$

$$
=\prod_{t=i}^{T-1}\left(\det(I_n-\gamma U_t^k)\right)^{-\frac{1}{2}}\left(\det(I_m-\gamma F_t^k)\right)^{-\frac{1}{2}}
$$

$$
\times\exp\left(\frac{\gamma}{2}\Big[x_{i-1}^{k\top}(A+BK_{i-1}^k)^\top\widetilde{U}_i^k(A+BK_{i-1}^k)x_{i-1}^k\right.
$$

$$
\left.+2g_{i-1}^{k\top}B^\top\widetilde{U}_i^k(A+BK_{i-1}^k)x_{i-1}^k+g_{i-1}^{k\top}B^\top\widetilde{U}_i^k Bg_{i-1}^k\Big]\right).
$$

Repeating this procedure, we can obtain

$$
J_0^{\pi^k}(x_0^k)-J_0^\star(x_0^k)
$$

$$
=\frac{1}{\gamma}\log\left(\prod_{t=1}^{T-1}\left(\det(I_n-\gamma U_t^k)\right)^{-\frac{1}{2}}\left(\det(I_m-\gamma F_t^k)\right)^{-\frac{1}{2}}\right)
$$

$$
+\frac{1}{\gamma}\log\mathbb{E}\left[\exp\left(\frac{\gamma}{2}\Big[x_0^{k\top}\left(\Delta K_0^{k\top}(R+B^\top\widetilde{P}_1B)\Delta K_0^k+(A+BK_0^k)^\top\widetilde{U}_1^k(A+BK_0^k)\right)x_0^k\right.\right.
$$

$$
\left.\left.+2g_0^{k\top}(RK_0^k+B^\top(\widetilde{P}_1+\widetilde{U}_1^k)(A+BK_0^l))x_0^k+\sigma_k^2\sigma_k^{-2}g_0^{k\top}(R+B^\top(\widetilde{P}_1+\widetilde{U}_1^k)B)g_0^k\Big]\right)\bigg|x_0^k,H_T^{k-1}\right]
$$

$$
=\frac{1}{\gamma}\log\left(\prod_{t=1}^{T-1}\left(\det(I_n-\gamma U_t^k)\right)^{-\frac{1}{2}}\left(\det(I_m-\gamma F_t^k)\right)^{-\frac{1}{2}}\right)
$$

$$
+\frac{1}{\gamma}\log\mathbb{E}\left[\exp\left(\frac{\gamma}{2}\left(x_0^{k\top}D_0^k x_0^k+2\sigma_k^{-1}g_0^{k\top}E_0^k x_0^k+\sigma_k^{-2}g_0^{k\top}F_0^k g_0^k\right)\right)\bigg|x_0^k,H_T^{k-1}\right]
$$

$$
\overset{(6)}{=}-\frac{1}{2\gamma}\sum_{t=1}^{T-1}\left[\log\det\left(I_n-\gamma U_t^k\right)+\log\det\left(I_m-\gamma F_t^k\right)\right]
$$

$$
+\frac{1}{2}x_0^{k\top}D_0^k x_0^k-\frac{1}{2\gamma}\log\det\left(I_m-\gamma F_0^k\right)+\frac{1}{2}\gamma x_0^{k\top}E_0^{k\top}\left(I_m-\gamma F_0^k\right)^{-1}E_0^k x_0^k
$$

$$
\overset{(7)}{=}-\frac{1}{2\gamma}\sum_{t=0}^{T-1}\log\det\left(I_n-\gamma F_t^k\right)-\frac{1}{2\gamma}\sum_{t=1}^{T-1}\log\det\left(I_m-\gamma U_t^k\right)+\frac{1}{2}x_0^{k\top}U_0^k x_0^k,
$$

where equality (6) holds by directly calculating the conditional expectation of quadratic function of $g_0^k$ in the second term of the previous equality and equality (7) follows from the definition of $U_0^k$ in (56). $\qquad\square$

## B.4 PROOF OF THEOREM 2

Now, we can derive the regret upper bound for Algorithm 2. Similar to Appendix A.4, we derive the bounds for the equations in (56). We recursively define the following constants similarly as (32) in Appendix A.4. For any $t=0,\cdots,T-2$,

$$
\alpha_{T-1}=2\widetilde{\Gamma}^3,
$$

$$
\beta_{T-1}=0,
$$

$$
\alpha_t=2\widetilde{\Gamma}\left(10\mathcal{V}^2\mathcal{L}\widetilde{\Gamma}^4\right)^{2(T-t-1)}+12\widetilde{\Gamma}^4\alpha_{t+1},
$$

$$
\beta_t=12\widetilde{\Gamma}^4+12\widetilde{\Gamma}^4\beta_{t+1},
$$

where $\mathcal{V}$ and $\mathcal{L}$ are defined in (13). To derive the regret bounds, we assume that for any $t = 0, \cdots, T-1$,

$$\gamma \leq \frac{1}{2\alpha_t \mathcal{V}^2 \epsilon_k^2 + 2\beta_t \cdot 5\widetilde{\Gamma}^5 \sigma_k^2 + 10\widetilde{\Gamma}^5 \sigma_k^2}. \tag{66}$$

We are now ready to derive the bounds for $D_t^k, E_t^k, F_t^k, U_t^k, \widetilde{U}_t^k$ recursively from step $T-1$ to step $0$ conditional on event $\widetilde{\mathcal{G}}^k$ in (39). At step $T-1$,

$$\begin{aligned}
\|D_{T-1}^k\| &= \left\|\Delta K_{T-1}^{k\top}(R + B^\top \widetilde{P}_T B)\Delta K_{T-1}^k\right\| \\
&\leq \|R + B^\top \widetilde{P}_T B\| \cdot \|\Delta K_{T-1}^k\|^2 \\
&\overset{(1)}{\leq} 2\widetilde{\Gamma}^3 \mathcal{V}^2 \epsilon_k^2 \\
&= \alpha_{T-1}\mathcal{V}^2 \epsilon_k^2 + 5\widetilde{\Gamma}^5 \beta_{T-1}\sigma_k^2,
\end{aligned} \tag{67}$$

where inequality (1) holds by the definition of $\widetilde{\Gamma}$ in (13) and Lemma 18. Similarly,

$$\begin{aligned}
\|E_{T-1}^k\| &= \left\|\sigma_k(RK_{T-1}^k + B^\top \widetilde{P}_T(A + BK_{T-1}^k))\right\| \\
&\leq \sigma_k\|R(\Delta K_{T-1}^k + K_{T-1})\| \\
&\quad + \sigma_k\|B\| \cdot \|\widetilde{P}_T\| \cdot \left(\|A\| + \|B\| \cdot \|K_{T-1}\| + \|B\| \cdot \|\Delta K_{T-1}^k\|\right) \\
&\overset{(2)}{\leq} \sigma_k\widetilde{\Gamma}(\mathcal{V}\epsilon_k + \widetilde{\Gamma}) + \sigma_k\widetilde{\Gamma}^2(2\widetilde{\Gamma}^2 + \widetilde{\Gamma}\mathcal{V}\epsilon_k) \\
&= 2\sigma_k\widetilde{\Gamma}^4 + \sigma_k\widetilde{\Gamma}^2 + \sigma_k(\widetilde{\Gamma} + \widetilde{\Gamma}^3)\mathcal{V}\epsilon_k \\
&\leq 3\sigma_k\widetilde{\Gamma}^4 + 2\sigma_k\widetilde{\Gamma}^3 \mathcal{V}\epsilon_k
\end{aligned} \tag{68}$$

where inequality (2) holds by the definition of $\widetilde{\Gamma}$ in (13) and Lemma 18. We also have

$$\left\|F_{T-1}^k\right\| = \left\|\sigma_k^2(R + B^\top \widetilde{P}_T B)\right\| \leq 2\widetilde{\Gamma}^3 \sigma_k^2. \tag{69}$$

Substitute (67), (68) and (69) into $U_{T-1}^k$, we can derive that

$$\begin{aligned}
\|U_{T-1}^k\| &= \|D_{T-1}^k + \gamma E_{T-1}^{k\top}(I_m - \gamma F_{T-1}^k)^{-1}E_{T-1}^k\| \\
&\leq \|D_{T-1}^k\| + \gamma \|E_{T-1}^{k\top}(I_m - \gamma F_{T-1}^k)^{-1}E_{T-1}^k\| \\
&\leq \|D_{T-1}^k\| + \gamma\|(I_m - \gamma F_{T-1}^k)^{-1}\| \cdot \|E_{T-1}^k\|^2 \\
&\overset{(3)}{\leq} 2\widetilde{\Gamma}^3 \mathcal{V}^2 \epsilon_k^2 + \gamma\left(3\sigma_k\widetilde{\Gamma}^4 + 2\sigma_k\widetilde{\Gamma}^3 \mathcal{V}\epsilon_k\right)^2 \cdot \frac{1}{1 - 2\widetilde{\Gamma}^3\gamma\sigma_k^2} \\
&= 2\widetilde{\Gamma}^3 \mathcal{V}^2 \epsilon_k^2 + \gamma\left(9\sigma_k^2\widetilde{\Gamma}^8 + 4\sigma_k^2\widetilde{\Gamma}^6\mathcal{V}^2\epsilon_k^2 + 12\sigma_k^2\widetilde{\Gamma}^7\mathcal{V}\epsilon_k\right) \cdot \frac{1}{1 - 2\widetilde{\Gamma}^3\gamma\sigma_k^2} \\
&\overset{(4)}{\leq} 2\widetilde{\Gamma}^3 \mathcal{V}^2 \epsilon_k^2 + 18\gamma\sigma_k^2\widetilde{\Gamma}^8 + o(\epsilon_k^2) \\
&\overset{(5)}{\leq} 2\widetilde{\Gamma}^3 \mathcal{V}^2 \epsilon_k^2 + 5\sigma_k^2\widetilde{\Gamma}^5 + o(\epsilon_k^2) \\
&= \alpha_{T-1}\mathcal{V}^2 \epsilon_k^2 + 5\widetilde{\Gamma}^5 \beta_{T-1}\sigma_k^2 + 5\widetilde{\Gamma}^5\sigma_k^2 + o(\epsilon_k^2),
\end{aligned} \tag{70}$$

where inequality (3) follows by substituting (67), (68) and (69) into (70) and the fact that for any matrix $M \in R^{n \times n}$, if $\|M\| < 1$, then $\|(I_n - M)^{-1}\| \leq \frac{1}{1-\|M\|}$, inequality (4) holds by the assumption in (66), i.e. $1 - 2\widetilde{\Gamma}^3\gamma\sigma_k^2 \geq 1 - \frac{1}{2}\sigma_k^2 \geq \frac{1}{2}$, and inequality (5) still holds by assumption (66), i.e. $18\gamma\sigma_k^2\widetilde{\Gamma}^8 \leq 18\sigma_k^2\widetilde{\Gamma}^8 \cdot \frac{1}{4\widetilde{\Gamma}^3} \leq 5\widetilde{\Gamma}^5\sigma_k^2$. Note that conditional on event $\widetilde{\mathcal{G}}^k$, $\epsilon_k^2$ is of order $\frac{1}{\sqrt{k}}$, so $\epsilon_k^2$ and $\sigma_k^2$ share the same order conditional on event $\widetilde{\mathcal{G}}^k$. Then, we can obtain

$$\left\|\widetilde{U}_{T-1}^k\right\| = \|(I_n - \gamma U_t^k)^{-1}U_t^k\| \overset{(6)}{\leq} \frac{\|U_{T-1}^k\|}{1 - \gamma\|U_{T-1}^k\|} \overset{(7)}{\leq} 2\|U_{T-1}^k\|, \tag{71}$$

where inequality (6) still holds by the fact that for any matrix $M \in R^{n \times n}$, if $\|M\| < 1$, then $\|(I_n - M)^{-1}\| \leq \frac{1}{1-\|M\|}$, and inequality (7) holds by the assumption in (66), i.e. $1 - \gamma\|U_{T-1}^k\| \geq 1 - \frac{1}{2} = \frac{1}{2}$. It follows from (70) that

$$\left\|\widetilde{U}_{T-1}^k\right\| \leq 2\alpha_{T-1}\mathcal{V}^2\epsilon_k^2 + 10\widetilde{\Gamma}^5\beta_{T-1}\sigma_k^2 + 10\widetilde{\Gamma}^5\sigma_k^2 + o(\epsilon_k^2).$$

With the bounds in the $(T-1)$-th step, we can recursively derive the bounds in the $(T-2)$-th step. At step $T-2$, by the similar arguments in step $T-1$, we can obtain

$$\begin{aligned}
\left\|D_{T-2}^k\right\| &= \left\|\Delta K_{T-2}^{k\top}\left(R + B^\top \widetilde{P}_{T-1}B\right)\Delta K_{T-2}^k + (A + BK_{T-2}^k)^\top \widetilde{U}_{T-1}^k(A + BK_{T-2}^k)\right\| \\
&\leq \left\|R + B^\top \widetilde{P}_{T-1}B\right\| \cdot \left\|\Delta K_{T-2}^k\right\|^2 + \left\|A + B\Delta K_{T-2}^k + BK_{T-2}\right\|^2 \cdot \left\|\widetilde{U}_{T-1}^k\right\| \\
&\overset{(8)}{\leq} 2\widetilde{\Gamma}^3\left(10\mathcal{V}^2\mathcal{L}\widetilde{\Gamma}^4\right)^2\mathcal{V}^2\epsilon_k^2 \\
&\quad + 3\left(\widetilde{\Gamma}^2 + \widetilde{\Gamma}^4 + \widetilde{\Gamma}^2\left(10\mathcal{V}^2\mathcal{L}\widetilde{\Gamma}^4\right)^2\mathcal{V}^2\epsilon_k^2\right)\left(4\widetilde{\Gamma}^3\mathcal{V}^2\epsilon_k^2 + 10\sigma_k^2\widetilde{\Gamma}^5 + o(\epsilon_k^2)\right) \\
&\leq 2\widetilde{\Gamma}^3\left(10\mathcal{V}^2\mathcal{L}\widetilde{\Gamma}^4\right)^2\mathcal{V}^2\epsilon_k^2 + 12\widetilde{\Gamma}^4\left(2\widetilde{\Gamma}^3\mathcal{V}^2\epsilon_k^2 + 5\widetilde{\Gamma}^5\sigma_k^2\right) + o(\epsilon_k^2) \\
&= \alpha_{T-2}\mathcal{V}^2\epsilon_k^2 + 5\widetilde{\Gamma}^5\beta_{T-2}\sigma_k^2 + o(\epsilon_k^2),
\end{aligned}$$

where inequality (8) holds by Lemma 18, (71) and the fact that $\left\|\sum_{t=1}^K x_t\right\|^2 \leq K\sum_{t=1}^K \|x_t\|^2$. Similar to (68), we have

$$\begin{aligned}
\left\|E_{T-2}^k\right\| &= \left\|\sigma_k\left(RK_{T-2}^k + B^\top\left(\widetilde{P}_{T-1} + \widetilde{U}_{T-1}^k\right)(A + BK_{T-2}^k)\right)\right\| \\
&\leq \sigma_k\left\|RK_{T-2}^k + B^\top\widetilde{P}_{T-1}(A + BK_{T-2}^k)\right\| + \sigma_k\left\|B^\top\widetilde{U}_{T-1}^k(A + BK_{T-2}^k)\right\| \\
&\leq \sigma_k\left\|R\Delta K_{T-2}^k + RK_{T-2}\right\| + \sigma_k\|B\| \cdot \left\|\widetilde{P}_{T-1}\right\| \cdot \left\|A + B\Delta K_{T-2}^k + BK_{T-2}\right\| \\
&\quad + \sigma_k\|B\| \cdot \left\|\widetilde{U}_{T-1}^k\right\| \cdot \left\|A + B\Delta K_{T-2}^k + BK_{T-2}\right\| \\
&\overset{(9)}{\leq} \sigma_k\widetilde{\Gamma}\left(\left(10\mathcal{V}^2\mathcal{L}\widetilde{\Gamma}^4\right)\mathcal{V}\epsilon_k + \widetilde{\Gamma}\right) + \sigma_k\widetilde{\Gamma}^2\left(2\widetilde{\Gamma}^2 + \widetilde{\Gamma}\left(10\mathcal{V}^2\mathcal{L}\widetilde{\Gamma}^4\right)\mathcal{V}\epsilon_k\right) \\
&\quad + \sigma_k\widetilde{\Gamma}\left(4\widetilde{\Gamma}^3\mathcal{V}^2\epsilon_k^2 + 10\sigma_k^2\widetilde{\Gamma}^5\right) \cdot \left(2\widetilde{\Gamma}^2 + \widetilde{\Gamma}\left(10\mathcal{V}^2\mathcal{L}\widetilde{\Gamma}^4\right)\mathcal{V}\epsilon_k\right) \\
&= \sigma_k\left(\widetilde{\Gamma}\left(10\mathcal{V}^2\mathcal{L}\widetilde{\Gamma}^4\right) + \widetilde{\Gamma}^3\left(10\mathcal{V}^2\mathcal{L}\widetilde{\Gamma}^4\right)\right)\mathcal{V}\epsilon_k + \sigma_k\left(\widetilde{\Gamma}^2 + 2\widetilde{\Gamma}^4\right) + o(\epsilon_k^2) \\
&\leq 2\sigma_k\widetilde{\Gamma}^3\left(10\mathcal{V}^2\mathcal{L}\widetilde{\Gamma}^4\right)\mathcal{V}\epsilon_k + 3\widetilde{\Gamma}^4\sigma_k + o(\epsilon_k^2),
\end{aligned}$$

where inequality (9) follows from Lemma 18 and (70). Similar to (69), we have

$$\left\|F_{T-2}^k\right\| = \left\|\sigma_k^2\left(R + B^\top\left(\widetilde{P}_{T-1} + \widetilde{U}_{T-1}^k\right)B\right)\right\| \leq 2\widetilde{\Gamma}^3\sigma_k^2 + o(\epsilon_k^2).$$

Similar to (70), we have

$$\begin{aligned}
\left\|U_{T-2}^k\right\| &\leq \left\|D_{T-2}^k\right\| + \gamma\|(I_m - \gamma F_{T-2}^k)^{-1}\| \cdot \left\|E_{T-2}^k\right\|^2 \\
&\leq 2\widetilde{\Gamma}^3\left(10\mathcal{V}^2\mathcal{L}\widetilde{\Gamma}^4\right)^2\mathcal{V}^2\epsilon_k^2 + 6\widetilde{\Gamma}^4\left(4\widetilde{\Gamma}^3\mathcal{V}^2\epsilon_k^2 + 10\widetilde{\Gamma}^5\sigma_k^2\right) + 18\gamma\sigma^2\widetilde{\Gamma}^8\sigma_k^2 + o(\epsilon_k^2) \\
&\overset{(10)}{\leq} 2\widetilde{\Gamma}^3\left(10\mathcal{V}^2\mathcal{L}\widetilde{\Gamma}^4\right)^2\mathcal{V}^2\epsilon_k^2 + 6\widetilde{\Gamma}^4\left(4\widetilde{\Gamma}^3\mathcal{V}^2\epsilon_k^2 + 10\widetilde{\Gamma}^5\sigma_k^2\right) + 5\widetilde{\Gamma}^5\sigma_k^2 + o(\epsilon_k^2) \\
&= \alpha_{T-2}\mathcal{V}^2\epsilon_k^2 + 5\widetilde{\Gamma}^5\beta_{T-2}\sigma_k^2 + 5\widetilde{\Gamma}^5\sigma_k^2 + o(\epsilon_k^2),
\end{aligned} \tag{72}$$

where inequality (10) follows from the assumption (66) and the similar arguments in (70). Then, by (72) and assumption (66), we can get

$$\left\|\widetilde{U}_{T-2}^k\right\| \leq 2\alpha_{T-2}\mathcal{V}^2\epsilon_k^2 + 10\widetilde{\Gamma}^5\beta_{T-2}\sigma_k^2 + 10\widetilde{\Gamma}^5\sigma_k^2 + o(\epsilon_k^2).$$

Repeat this procedure from step $T-1$ to step $0$, we can get the following recursive inequalities. For any $t = 0, \cdots, T-2$,

$$
\begin{aligned}
\left\|D_{T-1}^k\right\| &\leq \alpha_{T-1}\mathcal{V}^2\epsilon_k^2 + 5\widetilde{\Gamma}^5\beta_{T-1}\sigma_k^2, \\
\left\|F_{T-1}^k\right\| &\leq 2\widetilde{\Gamma}^3\sigma_k^2, \\
\left\|U_{T-1}^k\right\| &\leq \alpha_{T-1}\mathcal{V}^2\epsilon_k^2 + 5\widetilde{\Gamma}^5\beta_{T-1}\sigma_k^2 + 5\widetilde{\Gamma}^5\sigma_k^2 + o(\epsilon_k^2), \\
\left\|D_t^k\right\| &\leq \alpha_t\mathcal{V}^2\epsilon_k^2 + 5\widetilde{\Gamma}^5\beta_t\sigma_k^2 + o(\epsilon_k^2), \\
\left\|F_t^k\right\| &\leq 2\widetilde{\Gamma}^3\sigma_k^2 + o(\epsilon_k^2), \\
\left\|U_t^k\right\| &\leq \alpha_t\mathcal{V}^2\epsilon_k^2 + 5\widetilde{\Gamma}^5\beta_t\sigma_k^2 + 5\widetilde{\Gamma}^5\sigma_k^2 + o(\epsilon_k^2).
\end{aligned}
\tag{73}
$$

Substituting (73) into (57) in Lemma 7, we have

$$
\begin{aligned}
&J_0^{\pi^k}(x_0^k) - J_0^\star(x_0^k) \\
&= -\frac{1}{2\gamma}\sum_{t=1}^{T-1}\log\det\left(I_n - \gamma U_t^k\right) - \frac{1}{2\gamma}\sum_{t=0}^{T-1}\log\det\left(I_m - \gamma F_t^k\right) + \frac{1}{2}x_0^{k\top}U_0^k x_0^k \\
&\overset{(11)}{\leq} -\frac{1}{2\gamma}\sum_{t=1}^{T-1}\log\left(1 - \gamma\left\|U_t^k\right\|\right)^n - \frac{1}{2\gamma}\sum_{t=0}^{T-1}\log\left(1 - \gamma\left\|F_t^k\right\|\right)^m + \frac{1}{2}\left\|U_0^k\right\|\cdot\left\|x_0^k\right\|^2 \\
&\overset{(12)}{\leq} -\frac{1}{2\gamma}\sum_{t=1}^{T-1}n\log\left(1 - \gamma\left(\alpha_t\mathcal{V}^2\epsilon_k^2 + 5\widetilde{\Gamma}^5\beta_t\sigma_k^2 + 5\widetilde{\Gamma}^5\sigma_k^2 + o(\epsilon_k^2)\right)\right) \\
&\quad - \frac{1}{2\gamma}\sum_{t=0}^{T-1}m\log\left(1 - \gamma\left(2\widetilde{\Gamma}^3\sigma_k^2 + o(\epsilon_k^2)\right)\right) + \frac{1}{2}\left(\alpha_0\mathcal{V}^2\epsilon_k^2 + 5\widetilde{\Gamma}^5\beta_0\sigma_k^2 + 5\widetilde{\Gamma}^5\sigma_k^2 + o(\epsilon_k^2)\right)\|x_0\|^2 \\
&\overset{(13)}{\leq} \frac{n}{2}\sum_{t=1}^{T-1}\left(\alpha_t\mathcal{V}^2\epsilon_k^2 + 5\widetilde{\Gamma}^5\beta_t\sigma_k^2 + 5\widetilde{\Gamma}^5\sigma_k^2 + o(\epsilon_k^2)\right) + \frac{m}{2}\sum_{t=0}^{T-1}\left(2\widetilde{\Gamma}^3\sigma_k^2 + o(\epsilon_k^2)\right) \\
&\quad + \frac{1}{2}\left(\alpha_0\mathcal{V}^2\epsilon_k^2 + 5\widetilde{\Gamma}^5\beta_0\sigma_k^2 + 5\widetilde{\Gamma}^5\sigma_k^2 + o(\epsilon_k^2)\right)\|x_0\|^2,
\end{aligned}
\tag{74}
$$

where inequality (11) holds because

$$
I_n - \gamma U_t^k \succeq \left(1 - \gamma\left\|U_t^k\right\|\right)I_n \succeq \left(1 - \gamma\left(\alpha_t\mathcal{V}^2\epsilon_k^2 + 5\widetilde{\Gamma}^5\beta_t\sigma_k^2 + 5\widetilde{\Gamma}^5\sigma_k^2 + o(\epsilon_k^2)\right)\right)I_n \succ 0,
$$

and

$$
I_m - \gamma F_t^k \succeq \left(1 - \gamma\left\|F_t^k\right\|\right)I_m \succeq \left(1 - \gamma\left(2\widetilde{\Gamma}^3\sigma_k^2 + o(\epsilon_k^2)\right)\right)I_m \succ 0,
$$

inequality (12) follows from the inequalities in (73), inequality (13) holds by the fact that $\log(1 + x) \leq x$ for any $x > -1$.

Then, substituting the high probability bounds derived in Appendix B.1 into (74), we can further bound $J_0^{\pi^k}(x_0^k) - J_0^\star(x_0^k)$. According to Proposition 6, conditional on event $\widetilde{\mathcal{G}}^k$ defined in (39), when $kT \geq 200\left(3(n+m) + \log\left(\frac{4N}{\delta}\right)\right)$, with probability at least $1 - \frac{\delta}{N-1}$, we have

$$
\left\|\theta^{k+1} - \theta\right\|^2 \leq \epsilon_k := \frac{C_N}{\sqrt{k}}.
\tag{75}
$$

where

$$
C_N = \frac{160n}{cT}\left(\log\left(\frac{4nN}{\delta}\right) + (n+m)\log\left(1 + \frac{\tilde{c}N\log\left(\frac{4TN^2}{\delta}\right)}{\lambda}\right)\right) + \frac{80\lambda(n+m)^2\widetilde{\Gamma}^2}{cT},
\tag{76}
$$

and $c, \tilde{c}$ are defined in (44) and (53). Denote $\tilde{k} = \left\lceil\frac{200\left(3(n+m) + \log\left(\frac{4N}{\delta}\right)\right)}{T}\right\rceil$. When $k > \tilde{k}$, the estimation error bounds are given by (75).

By a similar mathematical induction as discussed in Section A.4 and page 27 in Basei et al. (2022), we can prove that the event $\widetilde{\mathcal{G}} = \left\{ \|\theta^k - \theta\| \leq \varpi, \forall k = 2, \cdots, N \right\} \cup \left\{ \theta^1 \in \Xi \right\}$ holds with probability at least $1 - \sum_{i=2}^{N} \frac{\delta}{N-1} = 1 - \delta$, i.e. $\mathbb{P}\left(\widetilde{\mathcal{G}}\right) \geq 1 - \delta$, where $\varpi$ is defined in (37).

Finally, conditional on the event $\widetilde{\mathcal{G}}$, we can derive an upper bound for $\mathrm{Regret}(N)$. Note that

$$\mathrm{Regret}(N) = \sum_{k=1}^{\tilde{k}} \left( J^{\pi^k}(x_0^k) - J^{\star}(x_0^k) \right) + \sum_{k=\tilde{k}+1}^{N} \left( J^{\pi^k}(x_0^k) - J^{\star}(x_0^k) \right), \tag{77}$$

where $\tilde{k} = \left\lceil \frac{200\left(3(n+m)+\log\left(\frac{4N}{\delta}\right)\right)}{T} \right\rceil$. We bound the two terms in (77) separately. We first bound the regret incurred up to the $\tilde{k}$-th episode. We have

$$\sum_{k=1}^{\tilde{k}} \left( J^{\pi^k}(x_0^k) - J^{\star}(x_0^k) \right) \leq \sum_{k=1}^{\tilde{k}} J^{\pi^k}(x_0^k) \leq \sum_{k=1}^{\tilde{k}} \frac{1}{\gamma} \log \mathbb{E} \exp \left( \frac{\gamma\widetilde{\Gamma}}{2} \left( \sum_{t=0}^{T-1} \left( \|x_t^k\|^2 + \|u_t^k\|^2 \right) + \|x_T^k\|^2 \right) \right). \tag{78}$$

It follows from (51) in Lemma 16 that

$$\sum_{k=1}^{\tilde{k}} \left( J^{\pi^k}(x_0^k) - J^{\star}(x_0^k) \right)$$

$$\leq \frac{\widetilde{\Gamma}\tilde{k}}{2} \left( \tilde{c} \log \left( \frac{TN}{\delta} \right) + 72T \left( \widetilde{\Gamma}(1+C_K) \right)^{2T} \left( n^{\frac{3}{2}} + m^{\frac{3}{2}} \right) \max\{\|x_0\|^2, 1\} \widetilde{\Gamma}^2 \log \left( \frac{TN}{\delta} \right) \right).$$

We next bound the regret in the remaining episodes as follows:

$$\sum_{k=\tilde{k}+1}^{N} \left( J^{\pi^k}(x_0^k) - J^{\star}(x_0^k) \right)$$

$$\leq \sum_{k=\tilde{k}+1}^{N} \left[ \frac{n}{2} \sum_{t=1}^{T-1} \left( \alpha_t \mathcal{V}^2 \epsilon_k^2 + 5\widetilde{\Gamma}^5 \beta_t \sigma_k^2 + 5\widetilde{\Gamma}^5 \sigma_k^2 + o(\epsilon_k^2) \right) + \frac{m}{2} \sum_{t=0}^{T-1} \left( 2\widetilde{\Gamma}^3 \sigma_k^2 + o(\epsilon_k^2) \right) \right.$$

$$\left. + \frac{1}{2} \left( \alpha_0 \mathcal{V}^2 \epsilon_k^2 + 5\widetilde{\Gamma}^5 \beta_0 \sigma_k^2 + 5\widetilde{\Gamma}^5 \sigma_k^2 + o(\epsilon_k^2) \right) \|x_0\|^2 \right]$$

$$\leq \sum_{k=\tilde{k}+1}^{N} \left[ \frac{n}{2} \sum_{t=1}^{T-1} \left( \frac{\alpha_t \mathcal{V}^2 C_N}{\sqrt{k}} + \frac{5\widetilde{\Gamma}^5 \beta_t}{\sqrt{k}} + \frac{5\widetilde{\Gamma}^5}{\sqrt{k}} + o(\epsilon_k^2) \right) + \frac{m}{2} \sum_{t=0}^{T-1} \left( \frac{2\widetilde{\Gamma}^3}{\sqrt{k}} + o(\epsilon_k^2) \right) \right.$$

$$\left. + \frac{1}{2} \left( \frac{\alpha_0 \mathcal{V}^2 C_N}{\sqrt{k}} + \frac{5\widetilde{\Gamma}^5 \beta_0}{\sqrt{k}} + \frac{5\widetilde{\Gamma}^5}{\sqrt{k}} + o(\epsilon_k^2) \right) \|x_0\|^2 \right]$$

$$\leq \left[ n \sum_{t=1}^{T-1} \left( \alpha_t \mathcal{V}^2 C_N + 5\widetilde{\Gamma}^5 \beta_t + 5\widetilde{\Gamma}^5 \right) + 2mT\widetilde{\Gamma}^3 + \left( \alpha_0 \mathcal{V}^2 C_N + 5\widetilde{\Gamma}^5 \beta_0 + 5\widetilde{\Gamma}^5 \right) \|x_0\|^2 \right] \sqrt{N} + o\left(\sqrt{N}\right), \tag{79}$$

where the first inequality follows from (74), the second inequality follows from (75) and $C_N$ is given in (76). On combining (78) with (79), we can obtain

$$\mathrm{Regret}(N) \leq \widetilde{\mathcal{C}} \sum_{t=0}^{T-1} \left( \alpha_t C_N + \beta_t \right) \sqrt{N},$$

where $\widetilde{\mathcal{C}} := \mathrm{Polynomial}\left( n, m, \tilde{c}, \mathcal{V}, \widetilde{\Gamma}, T, \tilde{k}, \|x_0\|, \epsilon_1 \right) \cdot \left( \widetilde{\Gamma}(1+C_K) \right)^{2T}.$

## C DEPENDENCY OF THE REGRET BOUNDS ON OTHER PARAMETERS

In this section, we provide some further discussions on the dependency of the regret bounds on other problem parameters, including the horizon length $T$, and the risk parameter $\gamma$ of the LEQR

model. Because the coefficient terms of regret bounds in Theorem 1 and Theorem 2 share the similar recursive structure, we focus on the regret bound in Theorem 1 and the regret bound in Theorem 2 can be analysed similarly. Spelling out the explicit dependency is generally difficult, due to the implicit dependency of $\widetilde{\Gamma}$ and constant $\mathcal{C}$ on the model parameters. Hence, in the following we focus our discussion on the term $\sum_{t=0}^{T-1} \psi_t$ in view of the bound (12).

Since $\{\psi_t\}_{t=0}^{T-1}$ is defined in a recursive manner, one can directly verify that

$$2\widetilde{\Gamma}^3 \left(10\mathcal{V}^2 \mathcal{L}\widetilde{\Gamma}^4\right)^{2(T-1)} \leq \sum_{t=0}^{T-1} \psi_t \leq 2\widetilde{\Gamma}^3 T^2 \left(10\mathcal{V}^2 \mathcal{L}\widetilde{\Gamma}^4\right)^{2(T-1)}. \tag{80}$$

The formula (80) implies that the term $\sum_{t=0}^{T-1} \psi_t$ has exponential dependence on the horizon length $T$. When $\gamma\widetilde{\Gamma} > 0$ is small, according to Taylor's Theorem, we have

$$\frac{1}{1 - \gamma\widetilde{\Gamma}} = 1 + \gamma\widetilde{\Gamma} + o\left(\gamma\widetilde{\Gamma}\right) \approx \exp\left(\gamma\widetilde{\Gamma}\right). \tag{81}$$

Using the formula of $\mathcal{L}$ in (13) and plugging (81) into (80), we find that the dependence of the term $\sum_{t=0}^{T-1} \psi_t$ on $\gamma$ is on the order of $\exp\left(12\gamma\widetilde{\Gamma}(T-1)\right)$. This also suggests that the regret bound in Theorem 1 has exponential dependence on $\gamma$ (ignoring the possible dependency of the constants $\mathcal{C}$ and $\widetilde{\Gamma}$ on these parameters).

Note that Basei et al. (2022) proved a regret bound that is logarithmic in the number of episodes $N$ for continuous-time risk neutral LQR problem, also in the finite-horizon episodic setting. They also mentioned (see Remark 2.2 in their paper) that the regret bound of their algorithm in general depends exponentially on the time horizon $T$. So our previous discussion is consistent with their findings. Note that they did not make explicit of the dependency of their regret bound on the horizon length $T$.

We also compare our results with Fei & Xu (2022), which proved gap-dependent logarithmic regret bounds for tabular MDPs under the entropic risk criteria. In particular, they showed their algorithms can achieve the regret of $\frac{(\exp(|\beta|H)-1)^2}{|\beta|^2 \Delta_{\min}} \cdot \text{poly}(H, S, A) \cdot \log\left(\frac{HSAK}{\delta}\right)$ with probability at least $1 - \delta$, where $\text{poly}(\cdot)$ represents the polynomial function, $H$ is the length of the episode, $S$ is the size of the state space, $A$ is the size of the action space, $\beta$ is the risk coefficient and $\Delta_{\min}$ is the minimum value of the sub-optimality gap of the value functions. Their regret bound also has exponential dependency on the risk coefficient $\beta$ and the length of the episode $H$, which is similar as our regret bound. While there are some similarities, it is also important to emphasize we consider LEQR which has continuous state and action spaces, which are different from tabular MDPs with finite state and action spaces.

## D  SIMULATION RESULTS IN SYSTEM 2 AND SYSTEM 3 IN SECTION 5

In this section, we present the simulation results of Algorithms 1 and 2 for System 2 and System 3 that are defined in Section 5.

### D.1  SYSTEM 2

Figures 2a–2c show the average regret of Algorithm 1 in System 2 using 150 independent runs and Figures 2d–2f show the average regret of Algorithm 2 in the same system. The two blue dotted lines in Figures 2a and 2d represent the 95% confidence interval of the regret when $\gamma = 0.1$ and $T = 3$. In Figures 2b and 2e, we set the true risk aversion value $\gamma = 0.1$ and plot the regret of our algorithms with the true risk aversion value and the regret of the algorithms with misspecified risk aversion values. The results show that applying the algorithms with a wrong risk aversion value, e.g., applying the algorithm suitable for risk-neutral learning agents to a risk averse agent can lead to greater regret. In Figures 2c and 2f, we set $\gamma = 0.001$ and study the dependence of the regret on the time horizon $T$. As expected, a longer time horizon implies greater regret.

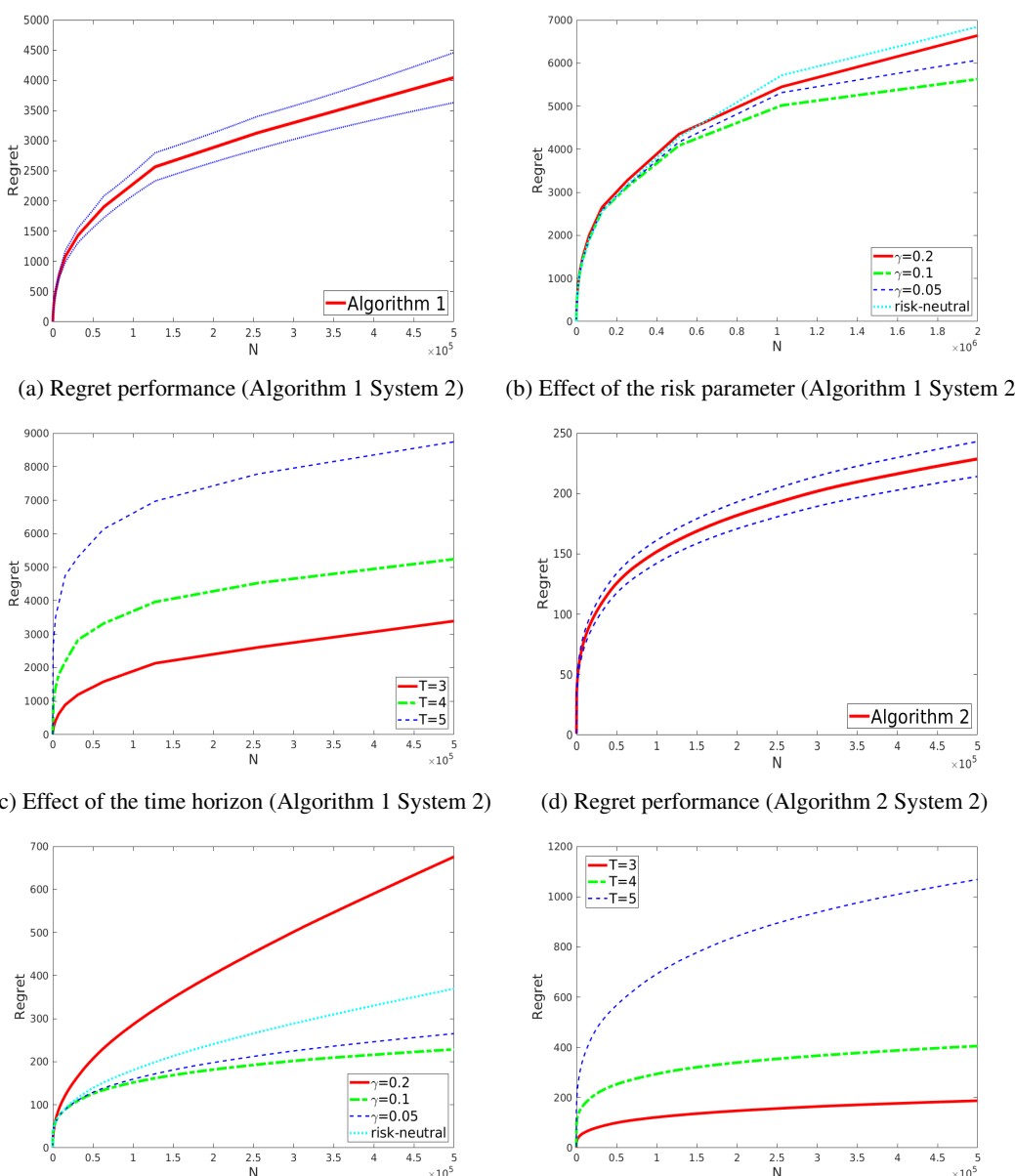

(a) Regret performance (Algorithm 1 System 2)  (b) Effect of the risk parameter (Algorithm 1 System 2)

(c) Effect of the time horizon (Algorithm 1 System 2)  (d) Regret performance (Algorithm 2 System 2)

(e) Effect of the risk parameter (Algorithm 2 System 2)  (f) Effect of the time horizon (Algorithm 2 System 2)

Figure 2: Simulation results in System 2

## D.2 SYSTEM 3

Figures 3a-3c show the average regret of Algorithm 1 in System 3 using 150 independent runs and Figures 3d–3f show the average regret of Algorithm 2 in the same system. The two blue dotted lines in Figures 3a and 3d depict the 95% confidence interval of the regret when $\gamma = 0.1$ and $T = 3$. Setting the true risk aversion value $\gamma = 0.1$, Figures 3b and 3e show the regret of the two algorithms with the true risk aversion value and the misspecified risk aversion values, which illustrates that applying the algorithms with an incorrect risk aversion value can cause poor performance of the algorithms. Setting $\gamma = 0.005$, Figures 3c and 3f illustrates the dependence of the regret on the time horizon $T$. Similar to the results in the previous two systems, the regret of the algorithms can increase when the time horizon is longer.

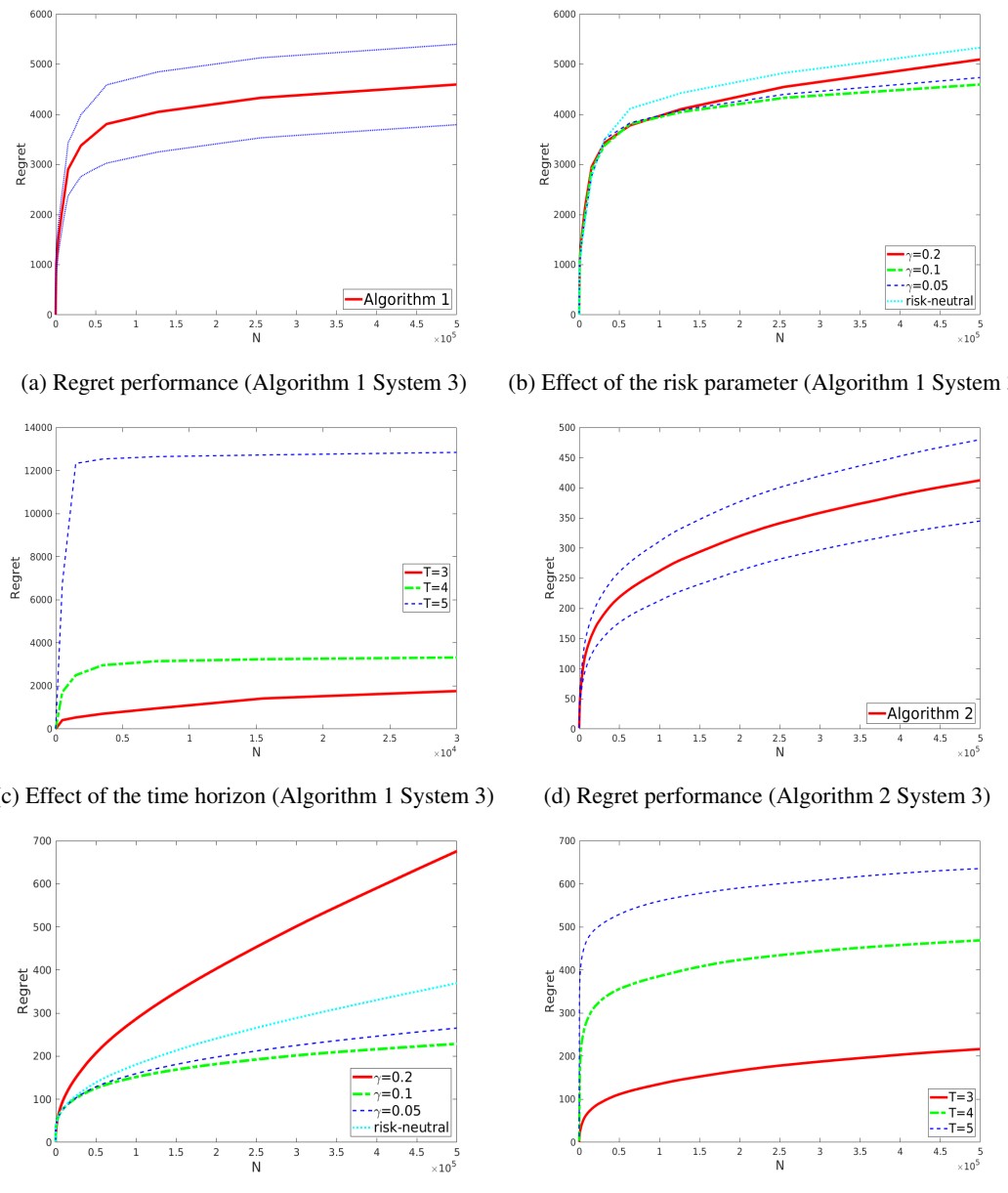

(a) Regret performance (Algorithm 1 System 3)  (b) Effect of the risk parameter (Algorithm 1 System 3)

(c) Effect of the time horizon (Algorithm 1 System 3)  (d) Regret performance (Algorithm 2 System 3)

(e) Effect of the risk parameter (Algorithm 2 System 3)  (f) Effect of the time horizon (Algorithm 2 System 3)

Figure 3: Simulation results in System 3

