# OpenReview forum: "Regret Bounds for Episodic Risk-Sensitive Linear Quadratic Regulator"
_ICLR.cc/2025/Conference — ICLR 2025 Poster_

### Official Review · Reviewer_VSSt · 2024-10-28

**Soundness:** 3
**Presentation:** 3
**Contribution:** 3
**Rating:** 8
**Confidence:** 3

**Summary:**

This paper proposes a least-squares greedy algorithm for the online adaptive control of the risk-sensitive linear quadratic regulator (LQR) in a finite-horizon, episodic setting, which achieves a regret bound of $O(\log n)$ under identifiability assumption. It also proposes adding exploration noise to the least-squares algorithm when the identifiability assumption does not hold, resulting in a modified approach with a regret bound of $O(\sqrt{n})$.

**Strengths:**

- The authors introduce two algorithms based on least-squares methods tailored for LEQR. The first least-squares greedy algorithm without exploration noise achieves a logarithmic regret bound $O(\log n)$ under an identifiability condition, and the second algorithm, designed for scenarios where the identifiability assumption does not hold, achieves a $O(\sqrt{n})$ regret bound.
- This work provides the first regret bounds for finite-horizon episodic LEQR, showing the existence of logarithmic regret under identifiability assumption.
- The theoretical analysis relies on perturbation analysis of special Riccati equations for the risk-sensitive LEQR setting, where additional matrices ($\widetilde P$) are present in the Riccati equations due to the risk sensitivity. The authors show nontrivial analysis techniques addressing the challenges in the analysis.
- The authors also introduce novel analysis on risk-sensitive loss.

**Weaknesses:**

- It would be beneficial to establish matching lower bounds for both algorithms to provide a more comprehensive understanding of their optimality.
- Typo/Inconsistency of the plot descriptions: Regret performance of Algorithm 1 (System 1) vs Effect of the risk parameter (Algorithm 1 System 1)

**Questions:**

- As the authors noted, studying regret bounds for LEQR in the infinite-horizon average-reward (non-episodic) setting would be interesting. What challenges arise in extending the current proof methods to this setting?

---

> ### Author Response · Authors · 2024-11-21
>
> Thank you for your great efforts in reviewing our paper and your positive feedback. We will change the plot descriptions as suggested. The following are the response to your comments and questions.
>
> **W1. Establish matching lower bounds for both algorithms.**
>
> Establishing regret lower bounds for online learning of LEQG is currently an open problem. For learning risk-neutral linear quadratic control in the infinite-horizon average reward setting, it has been shown in [1] that the regret lower bound is $\Omega(\sqrt{T})$, where $T$ is the number of steps the agent interacts with the system. While we consider LEQG in the finite-horizon episodic setting with a risk-sensitive objective, it might be possible to adapt the argument in [1] (albeit with significant modifications) to our setting to prove a regret lower bound $\operatorname{Regret}(N) = \Omega (\sqrt{N})$ (without any extra assumption such as identifiability).
>
> Specifically, we fix a nominal instance $(A\_t, B\_t)\_{t=0, \ldots, T-1}$ with the optimal controller $(K\_t)$. Consider also its purtubation $(\tilde A\_t, \tilde B\_t)\_{t=0, \ldots, T-1} = (A\_t - \Delta K\_t, B\_t + \Delta)\_{t=0, \ldots, T-1}$ with $\Delta \in\mathbb{R}^{n\times m}$ and the corresponding optimal controller $(\tilde K_t)$.
>  If the RL agent plays optimally with $u_t = K_t x_t$ for the nominal instance, then the system dynamics become
> \begin{align}
> x_{t+1}=(A_t +B_t K_t ) x_t+w_t, \quad t=0,1,\cdots,T-1.
> \end{align}
> Now the purturbed system has the same dynamics under the policy $(u_t)$ and it becomes indistinguishable from the nominal instance. But the optimal controller $(\tilde K_t)$ for the purturbed system is different from $(K_t)$. Hence if we can use purturbation analysis of the modified Reccati equation of the LEQR model to show that $(\tilde K_t)$ and $(K_t)$ must be at least $||\Delta||$ apart, then it follows that any low regret algorithm on the nominal instance will incur poor regret on the alternative/purturbed system. This argument provides the intuition for proving a (minimax) lower bound.  Nevertheless, due to time constraints, we are unable to present a thorough and rigorous argument at this moment, leaving the analysis of the regret lower bound for future exploration.
>
> [1] Simchowitz, M., and Foster, D. (2020). Naive exploration is optimal for online lqr. In International Conference on Machine Learning (pp. 8937-8948). PMLR.
>
> **W2. Typo/Inconsistency of the plot descriptions.**
>
> Thank you for pointing out the inconsistency of the plot descriptions. We have fixed this issue in the revision.
>
> **Q1. Challenges of extending the current proof methods to the infinite-horizon average-reward (non-episodic) setting.**
>
> For LEQR in the infinite-horizon average-reward (non-episodic) setting, the optimal value function and controller is characterized by a generalized algebraic Riccati equation, see Section 2.1 of [2].
> The main challenge of regret analysis in this setting lies in establishing explicit perturbation bounds for the generalized algebraic Riccati equation for average-reward LEQR, i.e., how the solutions to Riccati equation change when we perturb the system matrices. In the finite-horizon setting considered in this paper, the Riccati equation satisfies a recursion. This allows us to bound the purturbation error in a recursive manner. However, this approach does not work in the infinite-horizon average-reward setting due to the lack of recursive structure of the corresponding Riccati equation. In the risk-neutral non-episodic LQR setting, existing studies such as [3] use the certainty equivalent property of the risk-neutral LQR to address this issue,  where the certainty equivalent property here means that the risk-neutral LQR problem has the same Riccati equation and optimal controller regardless of the existence of system noise. However, this approach does not work for LEQR, because the solution to Riccati equation of LEQR depends on the covariance matrix of the system noise. Therefore, there are siginificant challenges in extending our current proof methods to the non-episodic LEQR setting.
> Finally, from the perspective of algorithm design, the non-episodic setting would require one to input an initial stabilizing controller, and check/test whether the controllers are stabilizing (i.e., with a finite average reward in infinite horizon) when executing the online algorithm for the infinite-horizon LEQR models. Such strategies (to handle stability issues) are not required for learning a finite-horizon LEQR model.
>
> [2] Cui, L., Basar, T., and Jiang, Z. P. (2023). A reinforcement learning look at risk-sensitive linear quadratic gaussian control. In Learning for Dynamics and Control Conference (pp. 534-546). PMLR.
>
> [3] Mania, H., Tu, S., and Recht, B. (2019). Certainty equivalence is efficient for linear quadratic control. Advances in Neural Information Processing Systems, 32.

---

> > ### Comment · Reviewer_VSSt · 2024-11-28
> >
> > Thank the authors for the detailed response! I'm going to keep my original score of 8.

---

### Official Review · Reviewer_y14b · 2024-11-04

**Soundness:** 3
**Presentation:** 3
**Contribution:** 2
**Rating:** 6
**Confidence:** 3

**Summary:**

The paper studies an online linear quadratic regulator (LQR) problem with risk-sensitive cost functions. In particular, when the environment can provide enough exploration noise, a least-square model fitting algorithm is provided and it achieves log N regret. When the environment is not "noisy" enough, the authors present a gaussian exploration method that achieves square root N regret. Numerical experiments are provided to validate the proposed algorithms.

**Strengths:**

The paper is well-written and easy to follow. All the assumptions are well-motivated and standard in the literature. The proposed algorithms are simple and can be easily implemented by practitioners.

----

**Weaknesses:**

*1.* I think the authors should provide an explicit dependence of gamma in the two theorems, i.e., the big O should contain gamma orders, as gamma is the key in the studied risk-sensitive LQR problems. With gamma dependence in the results, the authors could compare the regret order with risk-neural MDP/LQR problems. Discussions/guidance on gamma selections could be included.

*2.* Literature/discussions on risk-averse RL/LQR problems are limited, in particular, the choice of risk-averse metrics and the reason for studying the LEQR problem.

*3.* The authors could provide a more interesting real-world control problem in the simulation.

----

**Questions:**

*1.* Do the two theorems hold when gamma goes to 0, i.e., the limit is right-continuous in gamma?

*2.* Typo: line 218-219, "We" play.

---

> ### Author Response · Authors · 2024-11-21
>
> We are grateful for your efforts in reviewing our paper. We will address the minor comments and correct the typos as suggested. Below are the responses to your questions.
>
> **W1. The explicit dependence of $\gamma$ and compare the regret order with risk-neural MDP/LQR problems.**
>
> We have added a detailed discussion on the dependence of the regret bounds given in Theorem 1 and 2 on the risk-sensitivity parameter $\gamma$ in the appendix of the revised version of our paper, see p. 45-46. The dependency on $\gamma$ is exponential, for example, it is on the order of  $\exp\left(12\gamma\widetilde{\Gamma} T\right)$ for the regret bound in Theorem 1.
>
> The exponential dependency on the risk sensitivity paramter $\gamma$ is also common in the recent literature on regret bounds for risk-sensitive tabular MDPs. For example, [1] obtain logarithmic regret bounds for tabular MDPs with exponential utility function (similar as our work though we consider LQR), and their regret bound also has exponential dependency on the risk coefficient and the episode length $T$.
>
> [1] Fei, Y., and Xu, R. (2022). Cascaded gaps: Towards logarithmic regret for risk-sensitive reinforcement learning. In International Conference on Machine Learning (pp. 6392-6417). PMLR.
>
> **W2. The choice of risk-averse metrics and the reason for studying the LEQR problem.**
>
> The exponential utility (or entropic risk) is one of the most important risk measures and it is widely used in economics, finance and other areas. There are abundant real-world motivating examples (especially in finance) for this choice, such as  risk-sensitive investment management [2] and optimal stock liquidations [3]. In addition, LEQR is one of the most fundamental problems in risk-sensitive optimal control with continuous state and action spaces. There is extensive literature on LEQR and there is also a growing interest recently in RL for LEQR (see the references at lines 118 - 128 on p.3 of our paper). Moreover, the first paper [4] that studies regret bounds for risk-sensitive tabular MDPs (Markov Decision Processes) also considers the exponential utility function.  Based on these considerations, in this paper we focus on the exponential risk-sensitive cost. If one considers applying other risk measures (e.g. CVAR) directly to the total cost, it is well known that the resulting control problem might not be time-consistent and the dynamic programming principle may not hold. Hence, how to formulate the regret analysis problem and design efficient algorithms for learning LQR with other risk measures is a significant open problem and we leave it to the future.
>
> [2] Risk-Sensitive Investment Management: A Guide for Quants
>
> [3] Schied, A., Schöneborn, T., and Tehranchi, M. (2010). Optimal basket liquidation for CARA investors is deterministic. Applied Mathematical Finance, 17(6), 471-489.
>
> [4] Fei Y, Yang Z, Chen Y, et al. Risk-sensitive reinforcement learning: Near-optimal risk-sample tradeoff in regret. Advances in Neural Information Processing Systems, 2020, 33: 22384-22395.
>
> **W3. Real-world control problem.**
>
> We have provided a flying robot problem from Section IV of [5] in the simulation experiments. Due to space limitations, the detailed simulation results are provided in Appendix D.2.
>
> [5] Tsiamis, A., Kalogerias, D. S., Chamon, L. F., Ribeiro, A., and Pappas, G. J. (2020, December). Risk-constrained linear-quadratic regulators. In 2020 59th IEEE Conference on Decision and Control (CDC) (pp. 3040-3047). IEEE.
>
> **Q1. Do the two theorems hold when gamma goes to 0?**
>
> Yes, the two theorems still hold when $\gamma$ goes to $0$. When $\gamma\to 0$, the modified Riccati equation in equation (5) converge to the standard Riccati equation in the risk-neutral setting, and in the two theorems $\mathcal{L}$ converges to $1$ and $\mathcal{V}$ converges to $4\widetilde{\Gamma}^3$. So every term that depends on $\gamma$ in our regret bounds  is well-defined when $\gamma\to 0$.
>
> **Q2. Typo: line 218-219, "We" play.**
>
> Thank you for pointing out the typo. We have fixed this issue in the revision.

---

> > ### Comment · Reviewer_y14b · 2024-11-25
> >
> > Thanks for your responses. All the concerns have been addressed and I will maintain my positive score.

---

### Official Review · Reviewer_Z5vG · 2024-11-04

**Soundness:** 3
**Presentation:** 3
**Contribution:** 3
**Rating:** 8
**Confidence:** 3

**Summary:**

This paper studies the online adaptive control of risk-sensitive Linear Quadratic Regulator (LQR) problem in a finite-horizon episodic setting, which is referred to as the Linear Exponential-of-Quadratic Regulator (LEQR) problem. The goal is to design the online algorithm that selects control that minimizes the total regret, i.e., the difference between the sum of minimum exponential risk-sensitive cost (defined in Eq. 2) and the algorithm's cost in each episode.

The authors introduce two algorithms: a least-squares greedy algorithm (Algorithm 1) that achieves $\tilde{O}(\log N)$ regret under identifiability condition (Assumption 1) and a second algorithm (Algorithm 2: Least-Squares-Based Algorithm with Exploration Noise) that incorporates exploration noise to achieve $\tilde{O}(\sqrt{N​})$ regret without the identifiability assumption. Finally, the authors have shown the different performance aspects of the proposed algorithms on two synthetic LEQR systems.

**Strengths:**

**The following are the strengths of the paper:**
1. This paper is the first to provide regret bounds for the episodic finite-horizon LEQR problem, which has applications in risk-sensitive control problems in areas like finance and healthcare.

2. The authors proposed two algorithms with sub-linear regret bounds guarantees. The regret bounds are derived using perturbation analysis of modified Riccati equations, which incorporate exponential risk-sensitive cost (defined in Eq. 2).

3. Finally, the authors have demonstrated the different performance aspects (sub-linear regret and effect of risk parameter and horizon on regret) of the proposed algorithms on two synthetic LEQR systems.

**Weaknesses:**

**The following are the weaknesses of the paper:**
1. Since verifying the identifiability assumption (Assumption 1) for a given problem may not be possible, the first algorithm may not be useful in practice.

2. Both proposed algorithms are restricted to fixed finite-horizon settings and linear dynamics, which limits their real-world application, where horizon length can vary across episodes and problems with non-linear dynamics.

3. The following parts of the paper are not clear enough:
    - Why are the $N$ episodes in Algorithm 1 divided into epochs of increasing lengths to estimate the system matrices?
    - Why does the paper only consider the exponential risk-sensitive cost? Are there any real-world motivating examples for this choice?

**Questions:**

Please address the weaknesses of the paper. I have a few more questions/comments:
1. Instead of using exponential noise, can a UCB (upper confidence bound)-based exploration strategy in Algorithm 2?

2. How to extend these results for other risk measures, e.g., coherent risk measures (VaR, CVaR, EVaR, etc.) as studies in the following paper:
	1. Lam et al., [Risk-Aware Reinforcement Learning with Coherent Risk Measures and Non-linear Function Approximation](https://openreview.net/pdf?id=-RwZOVybbj)

3. What is the overall dependence of regret bounds given in Theorem 1 and 2 on horizon length ($T$) and risk-sensitivity parameter ($\gamma$)?


**Minor comment:**

Adding the layman's explanation and consequences of Assumption 1 will help readers identify when this assumption can hold in practice. Also, details (in the appendix and refer to the main paper) can added on how the standard Riccati equations are modified to incorporate exponential risk-sensitive costs.

I am open to changing my score based on the authors' responses.

**Details Of Ethics Concerns:**

Since this work is a theoretical paper, I do not find any ethical concerns.

---

> ### Author Response · Authors · 2024-11-21
>
> We are grateful for your time and efforts in reviewing our paper. We provide responses to your comments and questions below.
>
> **W1. Verifying Assumption 1 may not be possible and Algorithm 1 may not be useful in practice.**
>
> Indeed, the first algorithm is primarily of theoretical significance. In the learning theory community, there has been a significant interest in the questions of whether logarithmic regret is possible for what type of linear systems and under what assumptions. See the references on p.2 (lines 77 - 79) of our paper.  Our first algorithm and its logarithmic regret provide an answer to these questions in the setting of episodic risk-sensitive LEQR models.
>
> In practice, one possible heuristic strategy is to devote a fixed number of episodes as a warm-up period for system identification by applying i.i.d Gaussian noises as control inputs (this only incurs a constant regret). Based on the collected data, one can perform a hypothesis testing about whether the underlying true system parameters satisfy Assumption 1 or not.
>
> **W2. Fixed finite-horizon settings, linear dynamics and real-world applications.**
>
> The fixed horizon setting is standard in the study of regret bounds for episodic reinforcement learning. Most papers assume a fixed horizon in the literature of regret bounds for episodic MDPs (Markov decision processes), see e.g. [1, 2]. We follow this standard assumption in our study of episodic risk-sensitive LQR problems.
>
> The linear dynamics in the LEQR problem are certainly special, and we consider it for three reasons: 1) LEQR is one of the most fundamental problems in risk-sensitive optimal control. There is extensive literature on LEQR and there is also a growing interest recently in RL for LEQR (see lines 118 - 128 on p.3 of our paper); 2) Some applications, especially in finance, naturally fall into our setting with a fixed-horizon and linear dynamics. For example, a common task faced by a financial institution is to liquidate a large position of assets, e.g., a stock, in a finite amount of time, e.g., in one day. With a linear price impact and exponential utility, the optimal liquidation problem becomes an LEQR problem with linear dynamics and fixed horizon (e.g., one day); See e.g. [3,4]; 3) One can use LEQR as a local approximation model and solve risk-sensitive control problems by iteratively solving LEQR problems. Hence our analysis in this paper could provide useful tools and serve as a first step in better understanding regret bounds for risk-sensitive nonlinear optimal control problems with continuous state and action spaces.
>
> [1] Azar, M. G., Osband, I., and Munos, R. (2017, July). Minimax regret bounds for reinforcement learning. In International conference on machine learning (pp. 263-272). PMLR.
>
> [2] Jin, C., Allen-Zhu, Z., Bubeck, S., and Jordan, M. I. (2018). Is Q-learning provably efficient?. Advances in neural information processing systems, 31.
>
> [3] Almgren, R., and Chriss, N. (2001). Optimal execution of portfolio transactions. Journal of Risk, 3, 5-40
>
> [4] Schied, A., Schöneborn, T., and Tehranchi, M. (2010). Optimal basket liquidation for CARA investors is deterministic. Applied Mathematical Finance, 17(6), 471-489.
>
> **W3(1). Reasons for the increasing epoch length in Algorithm 1.**
>
> We use this technique in order to obtain logarithmic regret bounds. If we use epochs of equal lengths to estimate the system matrices, then the regret will be linear in the total number of episodes $N$. This can be readily seen from the proof sketch of theorem 1 in Section 3.3.  The regret is bounded by $\sum_{l=1}^L m_l\epsilon_l^2$, where $\epsilon_l = O(\frac{1}{\sqrt{m_{l-1}}})$ is the estimation error of the system matrices, and $L$ is the number of epochs satisfying $\sum_{l=1}^L m_l=N$. If the epoch length $m_l$ is a constant, $L$ is linear in $N$ and so is the regret bound.

---

> ### Author Response · Authors · 2024-11-21
>
> **W3(2). Reasons for considering the exponential risk-sensitive cost and real-world examples.**
>
> The exponential utility (or entropic risk) is widely used in economics, finance and other areas. There are abundant real-world motivating examples (especially in finance) for this choice, such as risk-sensitive investment management [6] and optimal stock liquidations [7]. In addition, LEQR is one of the most fundamental problems in risk-sensitive optimal control with continuous state and action spaces. There is extensive literature on LEQR with exponential risk-sensitive cost and there is also a growing interest recently in RL for LEQR (see the references at lines 118 - 128 on p.3 of our paper). Moreover, the first paper [5] that studies regret bounds for risk-sensitive tabular MDPs (Markov Decision Processes) also considers the exponential utility function.  Based on these considerations, in this paper we focus on the exponential risk-sensitive cost. If one considers applying other risk measures (e.g. CVAR) directly to the total cost, it is well known that the resulting control problem might not be time-consistent and the dynamic programming principle may not hold. Hence, how to formulate the regret analysis problem and design efficient algorithms for learning LQR with other risk measures is a significant open problem and we leave it to the future.
>
> [5] Fei Y, Yang Z, Chen Y, et al. Risk-sensitive reinforcement learning: Near-optimal risk-sample tradeoff in regret. Advances in Neural Information Processing Systems, 2020, 33: 22384-22395.
>
> [6] Risk-Sensitive Investment Management: A Guide for Quants
>
> [7] Schied, A., Schöneborn, T., and Tehranchi, M. (2010). Optimal basket liquidation for CARA investors is deterministic. Applied Mathematical Finance, 17(6), 471-489.
>
> **Q1. Can a UCB-based exploration strategy in Algorithm 2?**
>
> The exploration strategy in Algorithm 2 is very simple and is often referred to as $\epsilon-$greedy (or naive random search) strategy in context of continuous control and LQR literature, see e.g. [8]. Such naive exploration is optimal for online LQR in the risk-neutral average-reward setting [9]. In this paper we adopt such $\epsilon-$greedy in Algorithm 2, and establish $\sqrt{N}$ regret bound for the episodic LEQR problem. For learning more complex nonlinear dynamical systems, one would expect the naive exploration strategy is not sufficient, and strategic exploration such as UCB is required for sample efficient learning. Note that implementing UCB for RL problems such as LQR with unbounded continuous state and action spaces is nontrivial and might be computationally inefficient [10].
>
> [8] Mania, H., Tu, S., and Recht, B. (2019). Certainty equivalence is efficient for linear quadratic control. Advances in Neural Information Processing Systems, 32.
>
> [9] Simchowitz, M., and Foster, D. (2020). Naive exploration is optimal for online lqr. In International Conference on Machine Learning (pp. 8937-8948). PMLR.
>
> [10] Abbasi-Yadkori, Y., and Szepesvári, C. (2011). Regret bounds for the adaptive control of linear quadratic systems. In Proceedings of the 24th Annual Conference on Learning Theory (pp. 1-26).
>
> **Q2. Extend the results for other risk measures.**
>
> Thanks for bringing to our attention this interesting paper which considers risk-aware RL with possibly continuous state space and finite action space. We have cited it in our revision. Extending our results to linear quadratic regulators (LQR) with other risk measures (e.g. coherent risk measures) is a significant open problem, because LQR features continuous state and action spaces and our regret analysis hinges on certain properties of the entropic risk measure (which is not coherent) we consider. Indeed, how to formulate the regret analysis problem for LQR with other risk measures requires care because the control problem might not be time-consistent and dynamic programming might fail.  Lam et al. 2023 considered the recursive application of coherent risk measures at each step (instead of applying risk measures to the total cost) to address this issue so that the resulting control problem is time-consistent. This formulation appears to be less studied in the (risk-sensitive) LQR literature.
> Therefore, we leave the extensions of our results to other risk measures to the future.

---

> > ### Author Response · Authors · 2024-11-21
> >
> > **Q3. The dependence of regret bounds on $T$ and $\gamma$.**
> >
> > We have added a detailed discussion on the dependence of the regret bounds given in Theorem 1 and 2 on horizon length $T$ and risk-sensitivity parameter $\gamma$ in the appendix of the revised version of our paper, see p. 45-46. The dependency on $T$ and $\gamma$ are both exponential.
> >
> > The exponential dependence of the regret bound on the time horizon $T$ might be fundamental to the risk-sensitive setting. Indeed, the lower bound in [11] shows that such exponential dependency is unavoidable for any algorithm with $\tilde O(\sqrt{N})$ regret in episodic tabular MDPs with exponential utility and $N$ is the number of episodes.
> >
> > The exponential dependency on the risk sensitivity paramter $\gamma$ is also common in the recent literature on regret bounds for risk-sensitive tabular MDPs. For example, [12] obtain logarithmic regret bounds for tabular MDPs with exponential utility function (similar as our work though we consider LQR), and their regret bound also has exponential dependency on the risk coefficient and the horizon length.
> >
> > [11] Fei Y, Yang Z, Chen Y, et al. Risk-sensitive reinforcement learning: Near-optimal risk-sample tradeoff in regret. Advances in Neural Information Processing Systems, 2020, 33: 22384-22395.
> >
> > [12] Fei, Y., and Xu, R. (2022). Cascaded gaps: Towards logarithmic regret for risk-sensitive reinforcement learning. In International Conference on Machine Learning (pp. 6392-6417). PMLR.
> >
> > **Minor comment: Layman's explanation and consequences of Assumption 1. Details for deriving the modified Riccati equation.**
> >
> > Assumption 1 is equivalent to equation (11) in our paper, which stipulates that the covariance matrix of the state-action pairs should be positively definite. This assumption essentially guarantees  the identifiability of the true system matrices when the time-dependent optimal control in (5) is executed.
> >
> > The derivation of the modified Riccati equations of the LEQR problem is known and the details can be found in Page 8 of Jacobson (1973) [13].
> >
> > [13] Jacobson, D. (1973). Optimal stochastic linear systems with exponential performance criteria and their relation to deterministic differential games. IEEE Transactions on Automatic control, 18(2), 124-131.

---

> > > ### Comment · Reviewer_Z5vG · 2024-11-26
> > >
> > > Dear authors, thank you for addressing my concerns and answering my questions. I have increased my rating to 8. Good luck!

---

### Official Review · Reviewer_pfYi · 2024-11-07

**Soundness:** 3
**Presentation:** 2
**Contribution:** 2
**Rating:** 6
**Confidence:** 4

**Summary:**

This paper considers the risk-sensitive LQR problem with unknown system matrices. The authors consider the standard LEQR formulation to capture the risk-aware scenario in the control problem and study the episodic problem setting where the system needs to be reset to the initial condition at the end of each episode. The authors propose online learning algorithms to solve the problem and show that the regret of the algorithm is bounded as \sqrt{N}, where N is the number of episodes in the problem. Under some additional assumption, the authors show that the regret bound becomes \logN.

**Strengths:**

1.	This paper initiate the consideration of risk-sensitive formulation of LQR in an online episodic setting.
2.	The paper rigorously characterize the regret bounds of the proposed algorithms.

**Weaknesses:**

1. Although the authors mentioned a previous work Basei et al. 2022 that considers an episodic control problem, it would be good to further explain why it is important/of practical interest to consider the episodic setting in the control problem. In addition, it would be good to discuss what the major challenge is if we move to the non-episodic setting which is the most standard setting in control problems (i.e., the system involves continuously and does not reset).
2. It would be good to explain more how the LEQR formulation captures the risk-aware scenario in Section 2.1. The authors also mention that their analysis extends to \gamma<0. What does \gamma<0 stand for in the risk-aware scenario? In general, it would be good to provide an intuitive explanation of how different values of γ (positive, negative, or zero) correspond to different risk attitudes. This would help readers better understand the practical implications of the LEQR formulation.
3. The authors should justify their choice to use only data from epoch l-1 for estimation in Algorithm 1, and discuss whether using data from multiple previous epochs could improve accuracy. Additionally, they should clarify whether the i.i.d. assumption on initial states is necessary for their results to hold.
4. The optimal controller based on the initial estimate \theta^1 needs to satisfy Assumption~1 as well in Theorem 1. Could you justify this assumption on \theta^1, i.e., how do you choose \theta^1 so that the assumption is satisfied?
5. Regarding the regret bounds provided in Theorems~1 and 2, they depend exponentially on the horizon length T for each episode. However, in episodic MDP problems, the regret bound depends on T only polynomially (e.g., https://proceedings.mlr.press/v70/azar17a/azar17a.pdf). Could the authors comment on this issue:  whether this is fundamental to the risk-sensitive setting or if it's an artifact of the regret analysis? Moreover, would it be possible to improve this dependence to polynomial in T?
6. Could you explain why the least squares approach in Algoirthm~2 needs a regularization term?
7. Please explicitly specify how the initial estimate \theta^1 is chosen in Algorithm~2.

**Questions:**

Please refer to Weakness.

---

> ### Author Response · Authors · 2024-11-21
>
> Thank you for the many constructive comments and suggestions! We provide responses to your comments and questions below.
>
> **1. Motivation for studying the episodic setting and the challenges for moving to the non-episodic setting.**
>
> **Motivation for the episodic setting:** Some applications, especially in finance, naturally fall into the episodic setting. For example, a common task faced by a financial institution is to liquidate a large position of assets, e.g., a stock, in a finite amount of time, e.g., in one day. With a linear price impact, such problems can be formulated as a stochastic control problem with linear dynamics and quadratic cost functions; see the classical reference [1]. One can consider optimizing the expected utility of the total cost of trading, and with an exponential utility function (see, e.g., [2, 3]), then the problem becomes an episodic LEQR problem.
> In different days, the institution may need to liquidate different assets, so the initial state of this problem, which represents the initial position of the asset that the institution needs to liquidate during the day, resets at the beginning of each day, resembling the episodic setting.
>
> **Major challenges for the non-episodic setting:**
> For LEQR in the infinite-horizon average-reward (non-episodic) setting, the optimal value function and controller is characterized by a generalized algebraic Riccati equation, see Section 2.1 of [4].
> The main challenge of regret analysis in this setting lies in establishing explicit perturbation bounds for the generalized algebraic Riccati equation for average-reward LEQR, i.e., how the solutions to Riccati equation change when we perturb the system matrices. In the finite-horizon setting considered in this paper, the Riccati equation satisfies a recursion. This allows us to bound the purturbation error in a recursive manner. However, this approach does not work in the infinite-horizon average-reward setting due to the lack of recursive structure of the corresponding Riccati equation. In the risk-neutral non-episodic LQR setting, existing studies such as [5] use the certainty equivalent property of the risk-neutral LQR to address this issue,  where the certainty equivalent property here means that the risk-neutral LQR problem has the same Riccati equation and optimal controller regardless of the existence of system noise. However, this approach does not work for LEQR, because the solution to Riccati equation of LEQR depends on the covariance matrix of the system noise. Therefore, there are significant challenges in extending our current proof methods to the non-episodic LEQR setting.
> Finally, from the perspective of algorithm design, the non-episodic setting would require one to input an initial stabilizing controller, and check/test whether the controllers are stabilizing (i.e., with a finite average reward in infinite horizon) when executing the online algorithm for the infinite-horizon LEQR models. Such strategies (to handle stability issues) are not required for learning a finite-horizon LEQR model.
>
> [1] Almgren, R., and Chriss, N. (2001). Optimal execution of portfolio transactions. Journal of Risk, 3, 5-40
>
> [2] Schied, A., Schöneborn, T., and Tehranchi, M. (2010). Optimal basket liquidation for CARA investors is deterministic. Applied Mathematical Finance, 17(6), 471-489.
>
> [3] Cartea, A, Jaimungal, S., and Penalva, J. (2015). Algorithmic and high-frequency trading. Cambridge University Press.
>
> [4] Cui, L., Basar, T., and Jiang, Z. P. (2023). A reinforcement learning look at risk-sensitive linear quadratic gaussian control. In Learning for Dynamics and Control Conference (pp. 534-546). PMLR.
>
> [5] Mania, H., Tu, S., and Recht, B. (2019). Certainty equivalence is efficient for linear quadratic control. Advances in Neural Information Processing Systems, 32.

---

> ### Author Response · Authors · 2024-11-21
>
> **2. Explanation of the risk aversion and the LEQR formulation.**
>
> It is well understood in the economics literature that $\gamma$ measures the risk aversion degree; see for instance [6]. For your convenience, we explain it in detail below. An individual's certainty equivalent (CE) of a random cost $Z$ is a deterministic amount of cost $c(Z)$ such that she is indifferent between $Z$ and $c(Z)$. We can then define comparative risk aversion in terms of the CE: Individual $A$ is more risk averse than individual $B$ if, for any random cost $Z$, $A$'s CE of $Z$ is larger than or equal to $B$'s CE. In other words, the same random cost $Z$ appears to be more costly, in terms of the CE, from the perspective of the more risk averse individual. Suppose that both individuals have expected utility preferences with exponential utility functions, i.e., that individual $i$ evaluates cost $Z$ by $\mathbb{E}[\exp(\gamma_i Z)]$, $i\in {A,B}$. Then, $A$ is more risk averse than $B$ if and only if $\gamma_A\ge \gamma_B$. Indeed, supposing $\gamma_A\ge \gamma_B$ and denoting by $c_i(Z)$ individual $i$'s CE of $Z$, we have
> \begin{align*}
>     &c_A(Z) = \frac{1}{\gamma_A}\log\mathbb{E}[\exp(\gamma_AZ)] = \frac{1}{\gamma_A}\log\mathbb{E}\left[\left(\exp(\gamma_BZ)\right)^{\gamma_A/\gamma_B}\right]\\\\
>     &\ge \frac{1}{\gamma_A}\log\left(\mathbb{E}\left[\exp(\gamma_BZ)\right]\right)^{\gamma_A/\gamma_B} = \frac{1}{\gamma_B}\log\mathbb{E}[\exp(\gamma_BZ)] = c_B(Z),
> \end{align*}
> where the inequality is due to Jensen's inequality. An individual is risk neutral if she is indifferent between any random cost $Z$ and its mean $\mathbb{E}[Z]$. An individual is called risk averse (risk seeking, respectively) if she is more (less, respectively) risk averse than the risk-neutral preferences. Therefore, we can see that in the LEQR problem, where the cost $Z$ is evaluated by $\mathbb{E}[\exp(\gamma Z)]$, $\gamma$ measures the risk aversion degree of the controller and a positive (negatively, respectively) $\gamma$ stands for risk-averse (risk-seeking, respectively) attitude.
>
> [6] Pratt, J. W. (1964). Risk aversion in the small and in the large. Econometrica, Vol. 32.
>
> **3. Reasons for the data selection in ALgorithm1 and the necessity of the i.i.d assumption on the initial states.**
>
> While using data from multiple previous epochs might improve estimation accuracy empirically, we can not use this approach in our theoretical analysis. This is because the control in each episode is computed based on the estimated system matrices, and it is different across different epochs. Hence, the trajectories/data are not i.i.d. across different epochs, and it is difficult to analyze the finite-sample error bound of the estimated system matrices if one uses non-i.i.d data from multiple previous epochs.
>
> The i.i.d assumption on the initial states is commonly used in the literature on regret analysis of the (risk-neutral) LQR problems, {see e.g. [7], [8] and [9]}. We need this assumption in the analysis of Algorithm 1 to ensure that the data trajectories within each epoch are i.i.d. for deriving estimation error bounds of system matrices.
>
> [7] Dean, S., Mania, H., Matni, N., Recht, B., and Tu, S. (2020). On the sample complexity of the linear quadratic regulator. Foundations of Computational Mathematics, 20(4), 633-679.
>
> [8] Mania, H., Tu, S., and Recht, B. (2019). Certainty equivalence is efficient for linear quadratic control. Advances in Neural Information Processing Systems, 32.
>
> [9] Basei, M., Guo, X., Hu, A., and Zhang, Y. (2022). Logarithmic regret for episodic continuous-time linear-quadratic reinforcement learning over a finite-time horizon. Journal of Machine Learning Research, 23(178), 1-34.
>
> **4. The methodology for selecting $\theta^1$ in Algorithm 1.**
>
> In Proposition 4 in the appendix of our submitted manuscript (see page 20), we provide a sufficient condition for Assumption 1 to hold: If the system matrices satisfy a) $A\in \mathbb{R}^{n\times n}$ has full rank; b) $B\in \mathbb{R}^{n\times m}$ has full column rank; c) $Q\succ 0$ and $Q_T=0$, then Assumption 1 is satisfied. Therefore, we can easily choose $\theta^1= (A^1, B^1)$ so that it satisfies Assumption 1.

---

> > ### Author Response · Authors · 2024-11-21
> >
> > **5. The exponential dependence of the regret bounds on $T$.**
> >
> > The exponential dependence of regret on the time horizon $T$ might be fundamental to the risk-sensitive setting. Indeed, the lower bound in [10] shows that such exponential dependency is unavoidable for any algorithm with $\tilde O(\sqrt{N})$ regret in tabular MDPs with exponential utility and $N$ is the number of episodes.
> >
> > [10] Fei Y, Yang Z, Chen Y, et al. Risk-sensitive reinforcement learning: Near-optimal risk-sample tradeoff in regret. Advances in Neural Information Processing Systems, 2020, 33: 22384-22395.
> >
> > **6. The necessity of the regularization term in Algorithm 2.**
> >
> > The regularization term in Algorithm 2 is used to guarantee that $\bar{V}^k$ is positive definite so that equation (15) is well-defined. The sample covariance matrix $\sum_{i=1}^k\sum_{t=0}^{T-1}z_t^iz_t^{i\top}$ can have zero eigenvalue due to the lack of sufficient amount of data in the first few episodes, so we need to add the regularization term to guarantee the positive definiteness of $\bar{V}^k$.
> >
> > **7. The methodology for selecting $\theta^1$ in Algorithm 2.**
> >
> > In our experiments implementing Algorithm 2, the initial estimate $\theta^1$ is randomly generated, where every element of the initial estimate is drawn from i.i.d. uniform distribution $U(0,1)$. We do not impose additional assumptions on the initial estimate.

---

> > > ### Comment · Reviewer_pfYi · 2024-11-26
> > >
> > > Thanks for your response. I have raised my score to 6. The authors are encouraged to study the non-episodic setting in the future.

---

### Meta-Review · Area_Chair_5HZm · 2024-12-21

**Metareview:**

This paper introduces a problem setting for online learning in risk-sensitive episodic LQR optimal control, proposes an algorithm, and establishes its regret upper bound. While the paper has weaknesses, such as the absence of a regret lower bound and insufficient explanation of its motivation and applications, the authors have adequately addressed these issues raised by the reviewers. The reviewers have reached a consensus with positive opinions, and therefore, I support the acceptance of this paper.

**Additional Comments On Reviewer Discussion:**

The reviewers expressed concerns about the limitations and practicality of restricting the problem to an episodic setting with a fixed episode length, as well as theoretical aspects such as the lack of a regret lower bound and the necessity of the assumptions. However, the authors' responses appear to have been sufficient to alleviate these concerns.

---

### Decision · Program_Chairs · 2025-01-22

Accept (Poster)